# Mahalanobis++: Improving OOD Detection via Feature Normalization

**Maximilian Müller** [1]   **Matthias Hein** [1]

## Abstract

Detecting out-of-distribution (OOD) examples is an important task for deploying reliable machine learning models in safety-critial applications. While post-hoc methods based on the Mahalanobis distance applied to pre-logit features are among the most effective for ImageNet-scale OOD detection, their performance varies significantly across models. We connect this inconsistency to strong variations in feature norms, indicating severe violations of the Gaussian assumption underlying the Mahalanobis distance estimation. We show that simple $\ell_2$-normalization of the features mitigates this problem effectively, aligning better with the premise of normally distributed data with shared covariance matrix. Extensive experiments on 44 models across diverse architectures and pretraining schemes show that $\ell_2$-normalization improves the conventional Mahalanobis distance-based approaches significantly and consistently, and outperforms other recently proposed OOD detection methods. Code is available at github.com/mueller-mp/maha-norm.

## 1. Introduction

Deep neural networks have demonstrated remarkable performance across a variety of real-world tasks. However, when faced with inputs that fall outside their training distribution, they can behave unpredictably and even result in high-confidence predictions (Hendrycks & Gimpel, 2017; Hein et al., 2019). These so-called out-of-distribution (OOD) inputs are often misclassified with high confidence as belonging to the in-distribution (ID) classes, creating significant risks for real-world deployments. OOD detectors aim to identify and reject such anomalous inputs — potentially prompting human intervention, transitioning to a safe state, or declining to provide a prediction — while still allowing

[1]University of Tübingen and Tübingen AI Center. Correspondence to: Maximilian Müller <maximilian.mueller@wsii.uni-tuebingen.de>.

*Proceedings of the 42nd International Conference on Machine Learning*, Vancouver, Canada. PMLR 267, 2025. Copyright 2025 by the author(s).

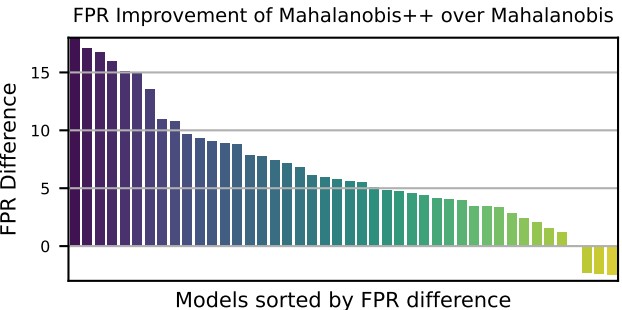

FPR Improvement of Mahalanobis++ over Mahalanobis

*Figure 1.* **Normalizing features improves OOD detection with the Mahalanobis distance consistently.** Shown is the difference in false-positive rate at true positive rate of 95% between unnormalized and normalized features for 44 ImageNet models, averaged over five OOD datasets of the OpenOOD benchmark.

genuine ID samples to pass through normally. OOD detection methods are commonly divided into methods that require modifications to the training process and so-called post-hoc detection methods that can be applied to any pretrained network. For many downstream tasks (not only OOD detection), the best results are achieved by models that have been pretrained on large datasets, some of which might not be publicly available. Since adjusting the training scheme for these networks is usually not feasible, simple post-hoc OOD detection is most often used in practice.

Common post-hoc OOD detection methods are based on a scoring function that typically inputs either the logit/softmax outputs of a model (Hendrycks & Gimpel, 2017; Hendrycks et al., 2022; Liu et al., 2020), or the pre-logit features (Lee et al., 2018b; Ren et al., 2021; Sun et al., 2022), or both (Sun et al., 2021; Wang et al., 2022). VisionTransformers have shown particular success in this area (Koner et al., 2021). For large-scale settings where, e.g., ImageNet is the ID dataset, they perform particularly well (Galil et al., 2023), especially when paired with feature-based methods (Bitterwolf et al., 2023). Among those, the Mahalanobis distance (Lee et al., 2018b; Ren et al., 2021) stands out as a particularly effective and simple scoring function. However, despite leading for some models to state-of-the-art OOD performance, it fails for others and shows high performance variation across different models and pretraining schemes, and brittleness when confronted with supposedly easy noise distributions as OOD data (Bitterwolf et al., 2023).

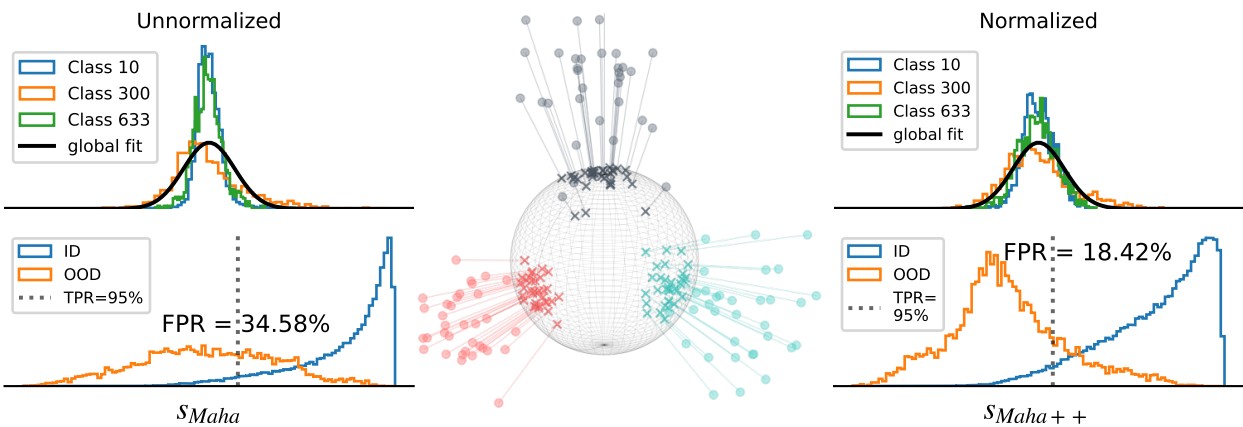

*Figure 2.* **Mahalanobis++:** We illustrate how to improve Mahalanobis-based OOD detection. **Left:** For unnormalized features, assuming a shared covariance matrix for all classes leads to suboptimal OOD detection (bottom) with the Mahalanobis score. **Center:** Normalizing the features, i.e. projecting them onto the unit sphere mitigates this problem effectively. **Right:** After normalization, the fit of the shared covariance matrix is tighter for all classes, leading to improved OOD detection as in- and out-distribution are better separated. Shown are the *Mahalanobis++* scores for a pretrained ConvNextV2-L on NINCO, which achieves a new state-of-the-art FPR of 18.4% (see Tab. 6).

In this work, we observe that for models where the Mahalanobis distance does not work well as OOD detector, the assumptions underlying the method are often not well satisfied. In particular, the feature norms vary much more than expected when assuming a Gaussian model with a shared covariance matrix. To mitigate this problem, we provide a simple solution, called *Mahalanobis++*, which we visualize in Figure 2: By projecting the features onto the unit sphere before estimating the Mahalanobis distance, we significantly reduce the class-dependent feature variability and obtain a better fit of the covariance matrix, which ultimately leads to consistent improvements in OOD detection, as demonstrated in Fig. 1 or Tab. 4.

In summary, our contributions are the following:

- We observe that the assumptions underlying the Mahalanobis distance as OOD detection method, in particular that the features are normally distributed with a shared covariance matrix, are often not well satisfied

- We relate this to variations in the feature norm, which can vary strongly across and within classes, and correlates with the Mahalanobis distance

- We provide an easy solution, which we call *Mahalanobis++*: Normalizing the features by their $\ell_2$-norm before computing the Mahalanobis distance

- We evaluate *Mahalanobis++* across a large range of models with different pretraining schemes and architectures on ImageNet and Cifar datasets and find that it consistently outperforms the conventional Mahalanobis distance and other baseline methods, and improves the detection of far-OOD noise distributions

## 2. Related Work

**Mahalanobis distance.** Most closely related to our work are the well-established OOD detection methods based on the Mahalanobis distance. Lee et al. (2018b) proposed to estimate a class-conditional Gaussian distribution with a shared covariance matrix "*with respect to (low- and upper-level) features*", and to use the minimal Mahalanobis distance to the respective mean vectors as OOD score. Since then, the community has transitioned to using only the pre-logit features. Ren et al. (2021) proposed to additionally estimate a class-agnostic mean and covariance matrix and use the difference between the two resulting scores as OOD score, called relative Mahalanobis distance. These methods have demonstrated broad applicability, spanning domains such as medical imaging (Anthony & Kamnitsas, 2023) and self-supervised OOD detection (Sehwag et al., 2021). Gaussian mixture models (GMMs) represent a more comprehensive framework for modelling feature distributions. They have been applied to small-scale setups but require tweaks to the training process (e.g. spectral normalization) (Mukhoti et al., 2023). Adapting them to ImageNet-scale setups as post-hoc OOD detectors has so far not been successful.

**Feature norm.** The role of the feature norm for OOD detection has been investigated in several works (Yu et al., 2020). Park et al. (2023b) underline that the norm of pre-logit features are equivalent to confidence scores and that the feature norms of OOD samples are typically smaller than those of ID samples. Their observations are mostly based on results obtained with strong over-training and simple networks. We will show that this observation does not hold generally. Gia & Ahn (2023) investigate the role of the $\ell_2$ norm in contrastive learning and OOD detection. Regmi et al. (2024)

and Haas et al. (2024) try to leverage the feature norm to discriminate between ID and OOD samples. In particular, they concurrently suggested training with $\ell_2$-normalized features and then using the norm of the unnormalized features as OOD score at inference time, similar to Yu et al. (2020) and Wei et al. (2022).

**Spherical embeddings.** Spherical embeddings have been investigated and leveraged across several fields (Liu et al., 2018; Zhou et al., 2022; Sablayrolles et al., 2018; Yaras et al., 2022), also within the OOD detection literature (Zheng et al., 2022). Ming et al. (2023) proposed CIDER, a contrastive training scheme that creates well-separated hyperspherical embeddings via a dispersion loss and applies KNN as detection method at inference time. Sehwag et al. (2021) also train with a contrastive loss, and apply the Mahalanobis distance as OOD detection method on the normalized features at inference time. Haas et al. (2023) observe that normalizing features during train and inference time improves performance on the DDU benchmark (Mukhoti et al., 2023). They hypothesize that their training scheme induces early neural collapse, which might benefit out-of-distribution detection capabilities of networks. Importantly, all those methods are *train-time* methods, i.e. require modifications to the training process, including feature normalization - either explicitly in the case of Haas et al. (2023), or implicitly through the contrastive loss in Ming et al. (2023) and Mukhoti et al. (2023). They then apply normalization at inference time, because they also normalized at train time. In contrast, we highlight the benefits of feature normalization when applying the Mahalanobis distance as *post-hoc* OOD detection method in this work - which is non-obvious for generic pretraining schemes.

**Cosine-based detection scores.** Many previous works have suggested using the angle, or more specifically, the cosine, for OOD detection, but those mostly require modifications to training or architecture (Techapanurak et al., 2020; Tack et al., 2020), or are used for unsupervised setups (Radford et al., 2021; Ming et al., 2022). Park et al. (2023a) and Sun et al. (2022) use nearest neighbour search in the normalized feature-space, which amounts to a nearest neighbour search in the cosine space. We show that *Mahalanobis++* outperforms cosine-based OOD detection methods.

## 3. Variations in feature norm degrade the performance of Mahalanobis-based OOD detectors

In this Section, we investigate the assumptions underlying the Mahalanobis distance as OOD detection method. We report results for NINCO (Bitterwolf et al., 2023) as OOD dataset. For all experiments, we use a pretrained ImageNet SwinV2-B-In21k model (Liu et al., 2022) with 87.1% ImageNet accuracy. This strong model is a prototypical example

where OOD detection on NINCO with Mahalanobis score performs significantly worse (FPR of 58.2%) than for other similar models like the ViT-B16-In21k-augreg with 84.5% accuracy (Steiner et al., 2022) but low FPR of 31.3% using the Mahalanobis score.

### 3.1. Mahalanobis Distance

The Mahalanobis distance is a simple, hyperparameter-free post-hoc OOD detector that has been suggested by Lee et al. (2018b). Given the training set $(x_i, y_i)_{i=1}^n$ with input $x_i$ and class labels $y_i$ one estimates: i) the class-wise means $\hat{\mu}_c$ and ii) a shared covariance matrix $\hat{\Sigma}$:

$$\hat{\mu}_c = \frac{1}{N_c} \sum_{i:y_i=c} \phi(x_i) \tag{1}$$

$$\hat{\Sigma} = \frac{1}{N} \sum_c^C \sum_{i:y_i=c} (\phi(x_i) - \hat{\mu}_c)(\phi(x_i) - \hat{\mu}_c)^T \tag{2}$$

where $\phi(x_i)$ are the pre-logit features of $x_i$, $N_c$ the number of train samples in class $c$, $N$ the total number of train samples, and $C$ the total number of classes. The Mahalanobis distance of a test sample $x_t$ to a class mean $\hat{\mu}_c$ is then

$$d_{Maha}(x_t, \hat{\mu}_c) = (\phi(x_t) - \hat{\mu}_c)^T \hat{\Sigma}^{-1} (\phi(x_t) - \hat{\mu}_c) \tag{3}$$

and the final OOD-score $s_{\text{Maha}}(x_t)$ of $x_t$ is the negative smallest distance to one of the class means:

$$s_{\text{Maha}}(x_t) = -\min_c d_{Maha}(x_t, \hat{\mu}_c) \tag{4}$$

If $s_{\text{Maha}}(x_t) \leq T$ then the sample is rejected as OOD, where for evaluation purposes $T$ is typically determined by fixing a TPR of 95% on the in-distribution. The core assumption of Lee et al. (2018a) is that "the pre-trained features of the softmax neural classifier might also follow the class-conditional Gaussian distribution". Indeed, one implicitly uses a probabilistic model where each class is modelled as a Gaussian $\mathcal{N}(\hat{\mu}_c, \hat{\Sigma})$ with a shared covariance matrix $\hat{\Sigma}$, which can be seen as a weighted average of the covariance matrices of the features of each class: $\hat{\Sigma} = \sum_{c=1}^C \frac{N_c}{N} \hat{\Sigma}_c$ with $\hat{\Sigma}_c = \frac{1}{N_c} \sum_{i:y_i=c} (\phi(x_i) - \hat{\mu}_c)(\phi(x_i) - \hat{\mu}_c)^T$, with the weight $N_c \backslash N$ being an estimate of $P(Y = c)$.

The Mahalanobis score is a strong baseline for OOD detection as noted in Bitterwolf et al. (2023) where they report for a particular Vision Transformer (ViT) trained with *augreg* (a carefully selected combination of augmentation and regularization techniques) by Steiner et al. (2022) state-of-the-art results on their NINCO benchmark comparing several models and OOD detection methods. On the other hand other ViTs like DeiT or Swin that are equally strong in terms of classification performance showed degraded OOD detection results. Moreover, Bitterwolf et al. (2023) report that the

Mahalanobis-based OOD detector performs worse on their "unit tests" of simple far-OOD test sets than other methods.

In the remainder of this section, we will try to identify the reasons for the varying performance of Mahalanobis-based OOD detection. Our main hypothesis is that it is due to violations of its core assumptions:

- **Assumption I:** the class-wise features $(\phi(x_i))_{y_i=c}$ follow a multivariate normal distribution $\mathcal{N}(\mu_c, \Sigma)$,

- **Assumption II:** the covariance matrix $\hat{\Sigma}$ is the same for all classes.

Below, we will show that these assumptions do not hold for some models, as indicated in Fig. 2. One strong indicator of this violation is the norm of the features, which turns out to be a strong confounder, ultimately degrading the OOD detection performance with Mahalanobis-based detectors.

For completeness, we mention the Relative Mahalanobis score here, proposed by Ren et al. (2021), also suggested as a fix to the Mahalanobis score. They argue that for the detection of near-OOD, one should use a likelihood ratio of two generative models compared to the likelihood used in the Mahalanobis method. Thus they fit a global Gaussian distribution with mean $\hat{\mu}_{\text{global}}$ and covariance matrix $\hat{\Sigma}_{\text{global}}$, and use the difference between the class-conditional and the global Mahalanobis score as OOD score.

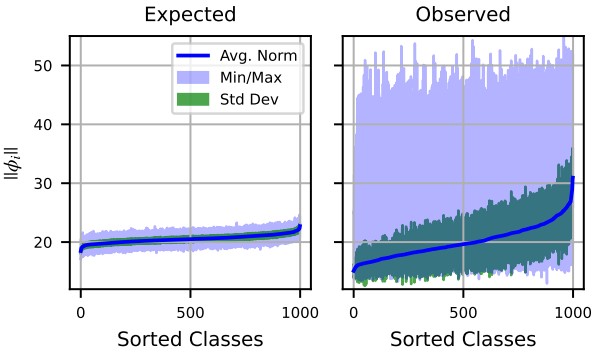

*Figure 3.* **The feature norms vary strongly across and within classes. Left:** We simulate how the feature norms per class would be distributed if they were sampled from Gaussians with the means and covariance matrix used for the Mahalanobis distance estimation. **Right:** The actual feature norm distribution observed in practice. Both the average norms across classes and the norms within each class vary much stronger than expected.

### 3.2. Is the Gaussian fit in feature space justified?

As the features $\phi(x_i) \in \mathbb{R}^d$ for input $x$ are high-dimensional, e.g. $d = 1024$ for a SwinV2-B, we expect some concentration of measure phenomenon if the features $\phi(x)$ of a

particular class are Gaussian distributed. In particular, the feature norm would be concentrated, as the following lemma shows.

**Lemma 3.1.** *Let $\Phi(X) \sim \mathcal{N}(\mu, \Sigma)$. Then*

$$P\left(|\,\|\Phi(X)\|_2^2 - (\text{tr}(\Sigma) + \|\mu\|_2^2)| \geq \epsilon\right) \leq \frac{\text{Var}\left(\|\Phi(X)\|_2^2\right)}{\epsilon^2},$$

*where $\text{Var}(\|\Phi(X)\|_2^2) := \sum_{i=1}^{d}(3\lambda_i^2 + 6\mu_i^2\lambda_i + \mu_i^4) - (\lambda_i + \mu_i^2)^2$ and $(\lambda_i)_{i=1}^{d}$ are the eigenvalues of $\Sigma$.*

This implies that $\|\Phi(X)\|_2$ should be concentrated around $\sqrt{\text{tr}(\Sigma) + \|\mu\|_2^2}$. In the right part of Fig. 3, we show the distribution of the norms of the training features across classes for the SwinV2-B model, i.e. the feature norms of those samples that were used for estimating class means and covariance. In the left part of Fig. 3 we show the distribution of feature norms when sampling from $\mathcal{N}(\hat{\mu}_c, \hat{\Sigma})$ for every class $c$. As expected from the derived Lemma, the sampled norms vary little around their mean value. It is evident by the differences of the left and right part of Fig. 3, that the fit with class-conditional means and shared covariance matrix does not represent the structure of the data well as the observed feature norms of SwinV2-B show heavy tails (right) which would not be present if the data was Gaussian (left). In Figure 8 we show that similar heavy-tailed feature norm distributions but with different skewness can be found even for the same ViT-architecture where the Mahalanobis score does not work well. This shows that Assumption I of the Mahalanobis score is not fulfilled across models, and models can deviate heavily from it. In contrast, for the ViT-augreg (Steiner et al., 2022), which has been shown to have very good OOD detection performance with the Mahalanobis score (Bitterwolf et al., 2023), the feature norms behave roughly as expected under the Gaussian assumption (right plot in Figure 8).

To further evaluate the adherence to Assumption I, we center training features of the SwinV2-B by their class means: $\phi^{\text{center}}(x_i) = \phi(x_i) - \mu_{c[i]}$. These centered features, used for covariance estimation, should ideally follow a zero-mean multivariate normal distribution. To quantify deviations from normality, we use Quantile-Quantile (QQ) plots, a standard approach in statistics (see, e.g. Wilk & Gnanadesikan (1968)) which compares sample quantiles against those of a theoretical distribution (here, the standard normal). A straight diagonal line indicates agreement with the theoretical distribution; deviations highlight mismatches. To enable direct comparison between models (and later between normalized and unnormalized features), we standardize $\phi^{\text{center}}(x_i)$ by its empirical standard deviation. While standardization technically alters the distribution (as the empirical variance is sample-dependent), we expect this to be

*Table 1.* **Variance alignment.** We measure how much the class-variances deviate from the global variance via the deviation score (see Eq. 5). Lower values indicate better alignment. Normalization aligns the features of SwinV2 and DeiT3, but not ViT-augreg.

|  | unnormalized | normalized |
|---|---|---|
| SwinV2-B-In21k | 0.26 | 0.12 |
| DeiT3-B16-In21k | 0.24 | 0.15 |
| ViT-B16-In21k-augreg | 0.05 | 0.05 |

negligible due to the large dataset size ($> 10^6$ samples). We report QQ-plots for three directions for a SwinV2-B and a DeiT3 (blue lines in Figure 4), and observe strong deviations from the ideal diagonal line, indicating that the centered features have much stronger tails than expected if the features followed a Gaussian distribution, further refuting Assumption I. We observe similar heavy tails in QQ-plots of other models where the Mahalanobis score is not working well for OOD detection (see Fig. 9 in App. D). Only the ViT with augreg training has a QQ plot close to the expected one.

To assess the validity of Assumption II, we measure how strongly the individual class variances deviate from the global variance. To this end, we compute the expected relative deviation over all directions:

$$\mathbb{E}_u[(u^T A u)^2] = \frac{2\mathrm{tr}(A^2) + \mathrm{tr}(A)^2}{d(d+2)}, \qquad (5)$$

where $u$ has a uniform distribution on the unit sphere and $A = \hat{\Sigma}^{-\frac{1}{2}}(\hat{\Sigma}_i - \hat{\Sigma})\hat{\Sigma}^{-\frac{1}{2}}$ (see App. C for a derivation). We average over all classes $i$ and report the results for a SwinV2-B, a DeiT3-B and a ViT-augreg in Table 1. We observe that the SwinV2 and DeiT3 show significantly larger deviations than the ViT-augreg, indicating that the class-specific variances differ more. More models in Tab. 7 in the Appendix.

### 3.3. Correlation of feature norm and $s_{\textbf{Maha}}$-score

The strong variations within and across classes we observed in Figure 3 indicate that the feature norm might impact the Mahalanobis estimation. To investigate this, we plot the feature norm against the Mahalanobis score $s_{Maha}$ assigned by the SwinV2-B model for ID and OOD test samples (i.e. samples that were not used for estimating means and covariance) in Figure 5. We observe a clear correlation: Samples with large feature norms consistently receive a large OOD score, and vice versa for samples with small feature norms - irrespective of whether they belong to the in or out distribution. Ideally, a detector should be able to distinguish ID from OOD samples irrespective of the norm of the OOD samples. Since a large fraction of the OOD samples have a comparably small feature norm, the resulting OOD detection performance is, however, poor. The reason for this strong correlation is the strongly different average feature

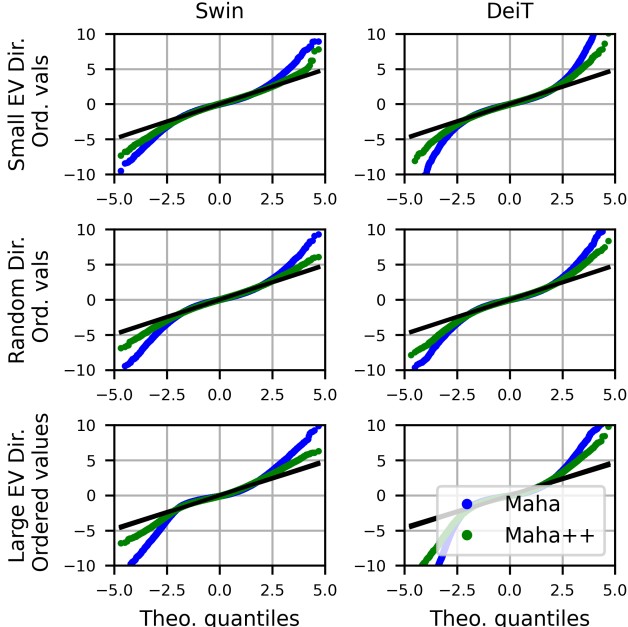

*Figure 4.* **QQ-plot:** $\ell_2-$**normalization helps transform the features to be more aligned with a normal distribution.** For a SwinV2 and DeiT3 model (where the feature norms vary strongly across and within classes) normalization shifts the distribution towards a Gaussian (black line).

norm across classes observed above. In Fig. 7, we observe the same correlation for other models (again, the ViT-B-augreg being an exception). In Fig. 6 in the Appendix, we substantiate this observation by artificially scaling the feature norm of OOD samples, leading to improved detection when the feature norm is increased and worse detection when the feature norm is decreased.

The heavy correlation between feature norm and OOD score implies that images yielding small feature norm are not detected as OOD (see Fig. 6 for a discussion). This also explains why the simple OOD unit tests in Bitterwolf et al. (2023), using synthetic images of little variation, e.g. black or uni-colour images, often fail. These synthetic images contain little variation in color, which often results in small activations in the network, and thus small pre-logit features, see Figure 10 for an analysis.

## 4. Mahalanobis++: Normalize your features

A challenge with Mahalanobis distance-based OOD detection is its sensitivity to feature norms, which can strongly correlate with Mahalanobis scores. We further find the feature distribution to strongly contradict the theoretical Gaussian assumption (with shared covariance), as empirical feature norms vary much more in practice than expected. To address this mismatch, we propose a simple but effective fix: Discarding the feature norm and leveraging only directional information in the features by $\ell_2$-normalization.

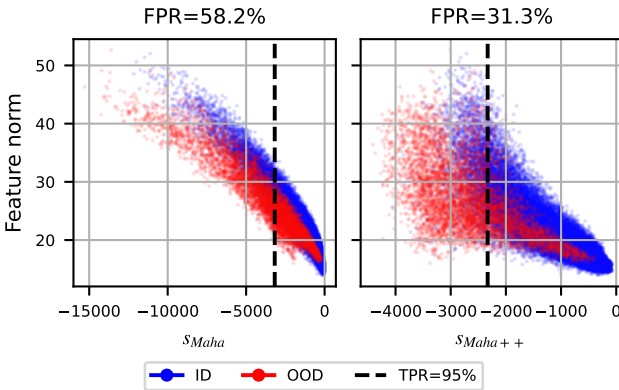

*Figure 5.* **The feature norm correlates with the Mahalanobis score for SwinV2-B: Left:** The smaller the feature norm, the smaller the Mahalanobis OOD score $s_{Maha}$, irrespective of whether a sample is ID or not. OOD samples with small feature norms are systematically classified as ID. **Right:** After normalization, OOD samples with small feature norms can be detected, and OOD detection is significantly improved.

**Method.** Instead of the original features $\phi(x)$, we use $\ell_2$-normalized ones for computing the Mahalanobis score:

$$\hat{\phi}(x_i) = \phi(x_i)/\|\phi(x_i)\|_2, \qquad (6)$$

The class-means and covariance matrix of the Mahalanobis score are estimated using the normalized features, and also test features are normalized when computing their score. We denote this simple modification as *Mahalanobis++*.

We note that $\ell_2$-normalization has been used with non-parametric post-hoc OOD detection methods like KNN (Sun et al., 2022; Park et al., 2023a) or cosine similarity (Techapanurak et al., 2020). With the Mahalanobis score, however, $\ell_2$-normalization has - to the best of our knowledge - only been investigated for train-time methods like SSD+ (Sehwag et al., 2021) or CIDER (Ming et al., 2023). Those methods normalize their features for OOD detection because they also normalize during training. This is orthogonal to our work: The standard Mahalanobis method for OOD detection is a post-hoc method, where adjusting the pretraining scheme is not feasible. We show below that *Mahalanobis++* outperforms KNN and cosine similarity in all considered cases and, in particular, improves OOD detection consistently across tasks, architectures, training methods and OOD datasets as *post-hoc* method.

**Improved normality.** To evaluate how *Mahalanobis++* improves the adherence to the assumption of a Gaussian model with a shared covariance matrix, we compare the resulting feature distributions via QQ-plots to the unnormalized features. Like for the unnormalized features, we center $\hat{\phi}^{center}(x_i) = \hat{\phi}(x_i) - \hat{\mu}_{c[i]}$, divide by the empirical standard deviation, and plot the resulting quantiles against the quantiles of a standard normal. We observe that across all

directions, normalization (green line in Figure 4) shifts the feature quantiles closer to the diagonal line, confirming that *Mahalanobis++* better satisfies the Gaussian assumption of Mahalanobis-based detection. We validate this for more models in Figure 9 in the Appendix.

**Variance alignment.** In Table 1, we observe lower variance deviation scores for normalized features of the SwinV2 model compared to unnormalized features, indicating that normalization aligns the class variances in *Mahalanobis++*. We illustrate this effect in Figure 2, which visualizes centered training features for three selected classes along a random direction. Without normalization, class feature variances differ substantially, and the shared covariance matrix fails to jointly capture their distributions. After normalization, class variances become more consistent, making the shared variance assumption more appropriate. To further validate this, we examine which in-distribution test samples are flagged as OOD at a 95% true-positive rate: unnormalized Mahalanobis rejects samples from 634 classes, while *Mahalanobis++* rejects samples from 728 classes. In an ideal setting with a perfect covariance fit, one would expect samples to be drawn uniformly from all 1,000 classes. The increase from 634 to 728 classes suggests that normalization reduces bias in the covariance estimation, better aligning with the shared variance assumption. We substantiate our observations in Figure 11 and Table 7 in the Appendix for more models. We find that class variances are more similar to the global variances after normalization for all models - except the ViT-augreg.

**Decoupling of feature norm and OOD score.** In Figure 5 on the right, we plot the feature norm of ID and OOD samples against their OOD scores obtained via *Mahalanobis++*. In contrast to the conventional Mahalanobis score, the correlation between OOD score and feature norm (before normalization) is much weaker. In particular, OOD samples with small feature norm are now also detected as OOD, which was not the case for unnormalized Mahalanobis.

## 5. Experiments

**ImageNet.** Our main goal is to investigate the effectiveness of *Mahalanobis++* across a large pool of architectures, model sizes and training schemes for ImageNet-scale OOD detection, as this is where the conventional Mahalanobis distance showed the most varied results in previous studies (Bitterwolf et al., 2023; Mueller & Hein, 2024). To this end, we use 44 publicly available model checkpoints from timm (Wightman, 2019) and `huggingface.co`. Following the OpenOOD setup (Yang et al., 2022), we report results on Ninco (Bitterwolf et al., 2023), iNaturalist (Van Horn et al., 2018), SSB-hard (Vaze et al., 2022), OpenImages-O (Krasin et al., 2017) and Texture (Cimpoi et al., 2014). We report the false positive rate at a true positive rate of 95% (FPR)

*Table 2.* **ImageNet.** FPR (lower is better) on five OpenOOD datasets. Green indicates that normalization improves over unnormalized features, **bold** indicates the best and underlined the second best method. *Mahalanobis++* consistently improves over Maha and baselines.

| model | ConvNeXtV2-B-In21k | | | | | | SwinV2-B-In21k | | | | | | DeiT3-B16-In21k | | | | | |
| dataset | NIN | SSB | TxT | OpO | iNat | Avg | NIN | SSB | TxT | OpO | iNat | Avg | NIN | SSB | TxT | OpO | iNat | Avg |
|---|---|---|---|---|---|---|---|---|---|---|---|---|---|---|---|---|---|---|
| MSP (Hendrycks & Gimpel, 2017) | 41.4 | 60.1 | 47.4 | 24.6 | 8.7 | 36.5 | 48.2 | 63.8 | 51.7 | 32.5 | 21.1 | 43.4 | 61.0 | 73.2 | 66.0 | 46.5 | 32.9 | 55.9 |
| MaxLogit (Hendrycks et al., 2022) | 31.9 | 51.1 | 40.7 | 16.5 | 4.9 | 29.0 | 38.6 | 52.6 | 47.7 | 24.6 | 13.0 | 35.3 | 55.2 | 67.2 | 61.9 | 41.4 | 34.3 | 52.0 |
| Energy (Liu et al., 2020) | 30.1 | 47.8 | 39.5 | 14.6 | 4.2 | 27.2 | 38.3 | **47.8** | 50.9 | 26.3 | 13.9 | 35.5 | 55.9 | 65.2 | 63.2 | 43.3 | 45.6 | 54.7 |
| GEN (Liu et al., 2023) | 29.7 | 52.3 | 35.9 | 13.8 | 3.5 | 27.1 | 37.0 | 57.0 | 38.7 | 17.6 | 8.9 | 31.8 | 45.2 | **61.5** | 50.1 | 24.8 | 13.7 | 39.0 |
| Energy+React (Sun et al., 2021) | 29.5 | 48.0 | 38.6 | 13.9 | 3.7 | 26.7 | 35.1 | 48.8 | 44.8 | 18.5 | 7.2 | 30.9 | 50.9 | 63.8 | 55.2 | 32.1 | 27.3 | 45.9 |
| fDBD (Liu & Qin, 2024) | 37.0 | 60.4 | 37.9 | 15.4 | 3.8 | 30.9 | 50.5 | 74.5 | 41.9 | 19.1 | 6.1 | 38.4 | 53.5 | 70.9 | 50.7 | 24.8 | 11.8 | 42.4 |
| ViM (Wang et al., 2022) | 26.9 | 47.7 | 28.1 | 7.9 | **1.1** | 22.4 | 50.4 | 75.7 | 35.4 | 15.4 | 1.7 | 35.7 | 55.3 | 75.7 | 48.8 | 21.1 | 4.8 | 41.1 |
| KNN (Sun et al., 2022) | 40.9 | 59.0 | 32.9 | 16.5 | 6.5 | 31.2 | 57.2 | 82.3 | 35.4 | 18.5 | 6.8 | 40.0 | 52.6 | 73.7 | 43.7 | 21.1 | 9.7 | 40.2 |
| Neco (Ammar et al., 2024) | 27.7 | **45.9** | 35.0 | 12.5 | 2.8 | 24.8 | 32.6 | 48.7 | 39.8 | 17.6 | 5.8 | 28.9 | 51.5 | 64.8 | 57.4 | 34.1 | 24.2 | 46.4 |
| NNguide (Park et al., 2023a) | 31.7 | 53.5 | 31.6 | 12.7 | 3.3 | 26.6 | 42.7 | 72.5 | 33.0 | 12.3 | 3.5 | 32.8 | 46.4 | 68.4 | 44.3 | 19.3 | 9.3 | 37.5 |
| Rel.-Mahalanobis (Ren et al., 2021) | 28.1 | 54.6 | 33.2 | 11.7 | 2.0 | 25.9 | 48.2 | 74.0 | 39.7 | 19.3 | 3.5 | 36.9 | 47.4 | 69.8 | 46.2 | 20.1 | 6.0 | 37.9 |
| Rel.-Mahalanobis++ | 24.7 | 51.6 | 32.1 | 11.3 | 2.2 | 24.4 | 34.4 | 62.5 | 36.8 | 15.7 | 3.9 | 30.6 | **38.3** | 61.6 | 42.5 | 17.1 | 4.0 | 32.7 |
| Mahalanobis (Lee et al., 2018b) | 30.3 | 53.8 | 30.4 | 9.4 | 1.4 | 25.0 | 58.2 | 81.4 | 41.5 | 23.2 | 3.5 | 41.6 | 52.5 | 72.8 | 47.0 | 21.4 | 5.5 | 39.8 |
| Mahalanobis++ | **22.4** | 46.9 | **26.5** | **7.8** | 1.3 | **21.0** | **31.3** | 62.0 | **28.7** | **9.7** | **1.6** | **26.7** | 38.8 | 62.8 | **42.0** | **15.6** | **3.1** | **32.5** |

as the OOD detection metric and refer to the appendix for other metrics, such as AUC, details on the model checkpoints, baseline methods, and extended results. In addition to *Mahalanobis++*, we also report *relative Mahalanobis++*, i.e the relative Mahalanobis distance with $\ell_2$ normalization.

We report results on the five OOD datasets in Table 2 using three pretrained base-size models: ConvNextV2 (Woo et al., 2023), SwinV2 (Liu et al., 2022) and DeiT3 (Touvron et al., 2022). For all models, *Mahalanobis++* outperforms the conventional Mahalanobis distance consistently across datasets, and is the best-performing method on average, and in most cases also per dataset. Also the *relative Mahalanobis++* outperforms its counterpart across models and datasets, but is slightly worse on average. In Table 4, we show the results averaged over the five datasets for 44 models with different training schemes, model sizes and network types. With the exception of three models (two of which are trained with *augreg*), *Mahalanobis++* outperforms its counterparts in *all* cases. *relative Mahalanobis++* outperforms its counterpart in 39/44 cases. In 30/44 cases, the best performing method is *Mahalanobis++* (in 6/44 cases it is *relative Mahalanobis++*) and the differences to the baseline methods are often large. Averaged over models, *Mahalanobis++* is the best method, followed by *relative Mahalanobis++* and outperforming the previously best method ViM by 7 FPR points. We note that *Mahalanobis++* is particularly effective for the best-performing models, as it is the best method for 4 of the top-5 models.

We further note that NNguide (Park et al., 2023a) and KNN (Sun et al., 2022), both of which operate in a normalized feature space, are consistently outperformed by *Mahalanobis++*. The most competitive baseline method that is not based on the Mahalanobis distance is ViM (Wang et al., 2022), which for certain models shows similar or slightly better performance than *Mahalanobis++* (e.g. for

*Table 3.* **CIFAR100.** Green indicates that normalization improves the baseline, **bold** and underlined indicate the best and second best method. We report FPR averaged over OpenOOD datasets. Maha++ is the best method. The best FPR is achieved by Maha++ for ViT-S16-21k highlighted in blue.

| Model | MSP | Ash | ML | KNN | ViM | MD | MD++ |
|---|---|---|---|---|---|---|---|
| SwinV2-S-1k | 47.28 | 92.66 | 40.96 | 36.27 | 34.02 | 40.10 | **26.01** |
| Deit3-S-21k | 48.92 | 94.47 | 42.37 | 36.81 | 39.99 | 41.99 | **31.72** |
| ConvN-T-21k | 60.60 | 92.11 | 57.44 | 51.16 | 51.18 | 52.48 | **42.69** |
| ViT-B32-21k | 48.02 | 93.98 | 31.28 | 26.49 | 27.14 | 26.28 | **18.94** |
| ViT-S16-21k | 52.17 | 80.45 | 37.63 | 31.91 | 24.90 | 25.51 | **18.58** |
| RN18 | 80.59 | 78.98 | 79.87 | 76.61 | 79.61 | 79.48 | **72.92** |
| RN34 | 76.93 | 78.27 | 75.33 | **74.44** | 77.17 | 76.63 | 74.51 |
| RNxt29-32 | 82.31 | 72.59 | 82.30 | 73.17 | 76.40 | 77.67 | **67.71** |
| Average | 62.10 | 85.44 | 55.90 | 50.86 | 51.30 | 52.52 | **44.13** |

EVA and DeiT networks). For several other networks (e.g., ConvNexts, Mixer, ResNets, EfficientNets, Swins,...), differences are, however, larger and often in the range of 8-15% FPR. We note that most of the OOD datasets in OpenOOD show contamination with ID samples, as reported in Bitterwolf et al. (2023). Therefore, we report results on Ninco, which has been cleaned from ID data, separately in Table 6, and find even clearer improvements of *Mahalanobis++*.

Two of the three models for which *Mahalanobis++* does not bring an improvement are ViTs trained with *augreg* by Steiner et al. (2022). Those are the models that showed state-of-the-art performance in Bitterwolf et al. (2023). We extend our observations from the previous Sections regarding these models in Appendix D, where we show that their feature norms are already well-behaved; therefore, $\ell_2$ normalization does not improve the normality assumptions.

Bitterwolf et al. (2023) reported that Mahalanobis-based detectors sometimes fail to detect supposedly easy-to-detect noise distributions (called *"unit tests"*). In Section 3, we connected this to the small feature norm those samples ob-

*Table 4.* FPR on OpenOOD datasets, Green indicates that a normalized method is better than its unnormalized counterpart, **bold** indicates the best and underlined the second best method. Maha++ improves over Maha on average by 7.6% in FPR over all models. Similarly, rMaha++ is, on average, 2.9% better in FPR than rMaha. In total, Maha++ improves the SOTA compared to the strongest competitor rMaha among all OOD methods by 6.9%, which is a significant improvement. The lowest FPR is achieved by Maha++ for the EVA02-L14-M38m-In21k highlighted in blue.

| Model | Val Acc | MSP | E | E+R | ML | ViM | AshS | KNN | NNG | NEC | GMN | GEN | fDBD | Maha | Maha++ | rMaha | rMaha++ |
|---|---|---|---|---|---|---|---|---|---|---|---|---|---|---|---|---|---|
| ConvNeXt-B-In21k | 86.3 | 41.7 | 40.1 | 36.0 | 37.3 | 29.5 | 88.5 | 37.2 | 31.8 | 31.4 | 54.2 | 32.6 | 37.9 | 33.6 | **24.3** | 31.7 | 29.5 |
| ConvNeXt-B | 84.4 | 61.4 | 90.9 | 86.9 | 70.2 | 52.8 | 99.5 | 58.7 | 51.2 | 66.5 | 73.9 | 60.1 | 60.3 | 54.2 | **44.6** | 50.0 | 45.4 |
| ConvNeXtV2-T-In21k | 85.1 | 44.7 | 37.3 | 37.1 | 38.6 | **27.0** | 96.7 | 41.6 | 36.4 | 33.2 | 47.2 | 36.5 | 42.3 | 32.5 | 28.6 | 34.6 | 33.4 |
| ConvNeXtV2-B-In21k | 87.6 | 36.5 | 27.2 | 26.7 | 29.0 | 22.4 | 95.3 | 31.2 | 26.6 | 24.8 | 38.9 | 27.1 | 30.9 | 25.0 | **21.0** | 25.9 | 24.4 |
| ConvNeXtV2-L-In21k | 88.2 | 35.0 | 27.0 | 26.5 | 28.5 | 28.7 | 95.6 | 30.8 | 24.1 | 32.9 | 26.4 | 31.6 | 27.8 | **18.8** | 25.8 | 23.0 |
| ConvNeXtV2-T | 83.5 | 60.5 | 66.1 | 58.6 | 58.9 | 49.9 | 99.2 | 72.1 | 62.8 | 54.1 | 73.9 | 53.6 | 61.8 | 55.4 | **44.4** | 48.9 | 44.6 |
| ConvNeXtV2-B | 85.5 | 58.8 | 70.8 | 64.1 | 59.5 | 46.8 | 99.6 | 53.4 | 47.9 | 55.9 | 71.2 | 48.6 | 53.7 | 46.3 | **37.5** | 43.0 | 39.1 |
| ConvNeXtV2-L | 86.1 | 58.6 | 68.0 | 60.1 | 58.3 | 48.4 | 99.1 | 48.9 | 44.7 | 55.9 | 63.8 | 46.3 | 48.5 | 41.7 | **36.2** | 39.0 | 38.0 |
| DeiT3-S16-In21k | 84.8 | 60.5 | 53.3 | 50.4 | 54.4 | 47.6 | 99.2 | 49.8 | 47.5 | 51.6 | 52.0 | 47.8 | 54.4 | 50.2 | **42.4** | 48.9 | 43.6 |
| DeiT3-B16-In21k | 86.7 | 55.9 | 54.7 | 45.9 | 52.0 | 41.1 | 99.2 | 40.2 | 37.5 | 46.4 | 46.2 | 39.0 | 42.4 | 39.8 | **32.5** | 37.9 | 32.7 |
| DeiT3-L16-In21k | 87.7 | 55.0 | 45.5 | 38.3 | 46.9 | 36.4 | 98.1 | 35.0 | 32.1 | 38.5 | 36.8 | 34.6 | 38.2 | 34.9 | 30.1 | 33.4 | **29.9** |
| DeiT3-S16 | 83.4 | 56.9 | 54.0 | 58.1 | 52.2 | **43.3** | 85.6 | 69.8 | 48.2 | 52.2 | 64.1 | 46.6 | 54.2 | 52.4 | 46.5 | 49.1 | 44.9 |
| DeiT3-B16 | 85.1 | 59.7 | 82.3 | 88.4 | 64.4 | **44.7** | 99.2 | 66.1 | 71.3 | 63.5 | 46.0 | 54.8 | 51.5 | 49.7 | 48.3 | 45.0 |
| DeiT3-L16 | 85.8 | 60.3 | 80.5 | 89.3 | 64.0 | 46.1 | 78.4 | 54.0 | 72.5 | 64.3 | 56.9 | 45.1 | 52.4 | 45.3 | 39.7 | 42.7 | **38.6** |
| EVA02-B14-In21k | 88.7 | 32.4 | 26.8 | 26.2 | 28.8 | 22.0 | 87.9 | 29.6 | 25.8 | 25.0 | 36.0 | 24.3 | 28.6 | 25.5 | **21.0** | 26.2 | 23.8 |
| EVA02-L14-M38m-In21k | 90.1 | 27.0 | 22.6 | 22.4 | 24.3 | 18.0 | 91.0 | 25.8 | 22.8 | 21.8 | 39.9 | 20.3 | 23.9 | 19.7 | **17.7** | 21.1 | 20.4 |
| EVA02-T14 | 80.6 | 64.8 | 66.2 | 66.8 | 63.1 | 49.3 | 98.4 | 60.8 | 57.1 | 55.4 | 54.1 | 57.9 | 66.4 | 51.0 | **48.1** | 52.6 | 50.7 |
| EVA02-S14 | 85.7 | 52.2 | 53.3 | 53.1 | 49.5 | **34.8** | 99.1 | 44.1 | 40.3 | 42.9 | 43.0 | 41.7 | 48.9 | 36.6 | 35.4 | 38.1 | 36.8 |
| EffNetV2-S | 83.9 | 59.3 | 71.0 | 58.7 | 61.1 | 52.2 | 99.4 | 45.6 | 45.2 | 59.3 | 77.9 | 49.7 | 54.0 | 47.3 | **40.2** | 43.6 | 40.4 |
| EffNetV2-L | 85.7 | 57.1 | 74.2 | 57.4 | 58.8 | 48.9 | 99.2 | 48.9 | 47.1 | 56.0 | 58.1 | 44.7 | 49.2 | 41.3 | **34.6** | 38.0 | 34.8 |
| EffNetV2-M | 85.2 | 57.0 | 69.3 | 56.7 | 57.3 | 54.7 | 99.5 | 51.3 | 48.6 | 54.9 | 66.8 | 45.2 | 52.6 | 46.0 | 37.1 | 41.1 | **36.8** |
| Mixer-B16-In21k | 76.6 | 71.5 | 83.0 | 83.5 | 75.0 | 71.8 | 95.8 | 77.8 | 83.9 | 75.5 | 61.2 | 67.7 | 71.8 | 63.3 | **52.5** | 60.0 | 52.9 |
| SwinV2-B-In21k | 87.1 | 43.4 | 35.5 | 30.9 | 35.3 | 35.7 | 77.0 | 40.0 | 32.8 | 28.9 | 57.7 | 31.8 | 38.4 | 41.6 | **26.7** | 36.9 | 30.6 |
| SwinV2-L-In21k | 87.5 | 40.4 | 35.9 | 31.2 | 34.5 | 39.0 | 85.1 | 38.9 | 32.9 | 29.0 | 48.8 | 31.5 | 37.5 | 41.8 | **24.7** | 36.2 | 28.7 |
| SwinV2-S | 84.2 | 61.2 | 68.1 | 62.1 | 60.9 | 51.1 | 99.9 | 58.6 | 52.9 | 56.0 | 61.2 | 52.8 | 60.7 | 52.4 | **38.9** | 48.7 | 39.3 |
| SwinV2-B | 84.6 | 62.4 | 66.2 | 58.2 | 60.5 | 49.9 | 99.1 | 55.0 | 51.1 | 55.4 | 56.1 | 49.9 | 56.9 | 47.9 | **40.1** | 45.2 | 39.7 |
| ResNet101 | 81.9 | 67.7 | 82.8 | 99.6 | 70.7 | 50.5 | 80.2 | 53.6 | 51.4 | 70.6 | 82.3 | 62.5 | 71.3 | 45.9 | 43.5 | 55.6 | 66.8 |
| ResNet152 | 82.3 | 66.4 | 82.1 | 99.5 | 70.0 | 49.7 | 80.0 | 52.0 | 46.8 | 69.1 | 77.2 | 60.3 | 69.3 | 44.4 | **38.3** | 51.8 | 64.7 |
| ResNet50 | 80.9 | 72.0 | 95.9 | 99.4 | 75.8 | 53.1 | 80.3 | 67.8 | 64.1 | 76.6 | 89.5 | 65.4 | 74.8 | 49.5 | 52.0 | 62.5 | 70.4 |
| ResNet50-supcon | 78.7 | 54.0 | 47.3 | 42.1 | 48.4 | 72.0 | **40.6** | 47.0 | 40.2 | 47.8 | 78.8 | 53.5 | 48.0 | 95.5 | 44.5 | 90.2 | 63.7 |
| ViT-T16-In21k-augreg | 75.5 | 70.7 | 55.3 | 48.4 | 58.3 | 51.1 | 94.9 | 76.2 | 71.0 | 52.8 | 58.2 | 64.7 | 58.5 | 55.5 | **48.0** | 59.2 | 57.7 |
| ViT-S16-In21k-augreg | 81.4 | 57.0 | 38.9 | 42.5 | 41.7 | 33.4 | 76.7 | 55.6 | 48.9 | 38.1 | 44.3 | 46.5 | 44.0 | 36.7 | **31.7** | 43.0 | 40.6 |
| ViT-B16-In21k-augreg2 | 85.1 | 55.3 | 45.9 | 41.1 | 47.5 | 53.9 | 98.6 | 47.5 | 42.2 | 43.7 | 60.1 | 42.9 | 51.4 | 54.2 | **38.2** | 47.0 | 39.1 |
| ViT-B16-In21k-augreg | 84.5 | 46.5 | 33.7 | 36.0 | 34.6 | **26.9** | 94.9 | 54.3 | 45.5 | 32.4 | 38.6 | 36.5 | 36.4 | 25.7 | 28.3 | 30.8 | 31.5 |
| ViT-B16-In21k-orig | 81.8 | 44.6 | 30.7 | 30.9 | 33.1 | 29.0 | 62.6 | 38.6 | 35.4 | 30.5 | 48.8 | 38.4 | 35.7 | 30.9 | **27.5** | 35.4 | 33.9 |
| ViT-B16-In21k-miil | 84.3 | 48.0 | 35.0 | 34.6 | 38.8 | 37.8 | 96.9 | 45.0 | 38.5 | 33.9 | 57.1 | 38.3 | 44.6 | 47.1 | **30.4** | 43.6 | 36.7 |
| ViT-L16-In21k-augreg | 85.8 | 40.2 | 29.4 | 25.0 | 30.0 | 23.6 | 94.5 | 50.6 | 41.2 | 28.0 | 41.6 | 30.7 | 30.4 | **21.0** | 23.9 | 25.2 | 25.8 |
| ViT-L16-In21k-orig | 81.5 | 40.8 | 29.3 | 29.2 | 31.1 | 30.4 | 49.3 | 34.0 | 31.6 | 29.5 | 47.9 | 35.2 | 33.0 | 30.9 | **26.8** | 33.6 | 32.6 |
| ViT-S16-augreg | 78.8 | 64.8 | 59.0 | 60.6 | 60.0 | 68.1 | 96.9 | 71.5 | 68.9 | 60.2 | 61.8 | 61.8 | 64.8 | 49.3 | 49.2 | 48.4 | **48.2** |
| ViT-B16-augreg | 79.2 | 64.3 | 59.6 | 56.2 | 60.1 | 63.4 | 90.2 | 65.5 | 64.1 | 59.9 | 60.3 | 61.5 | 63.5 | 49.6 | 48.0 | 47.6 | **46.7** |
| ViT-B16-CLIP-L2b-In12k | 86.2 | 42.2 | 37.7 | 35.5 | 37.2 | 35.5 | 99.5 | 35.6 | 31.6 | 34.0 | 41.4 | 33.4 | 38.0 | 43.2 | **28.1** | 38.0 | 32.4 |
| ViT-L14-CLIP-L2b-In12k | 88.2 | 31.5 | 25.2 | 24.6 | 26.5 | 21.5 | 97.6 | 29.9 | 22.3 | 26.3 | 36.3 | 24.3 | 27.3 | 28.2 | **22.4** | 27.1 | 25.4 |
| ViT-H14-CLIP-L2b-In12k | 88.6 | 32.0 | 26.5 | 26.1 | 27.7 | 22.3 | 99.8 | 31.2 | 23.4 | 27.5 | 53.0 | 24.6 | 28.9 | 27.1 | **22.0** | 26.8 | 25.1 |
| ViT-so400M-SigLip | 89.4 | 45.5 | 47.1 | 39.4 | 41.8 | 30.6 | 93.5 | 28.7 | 26.1 | 39.6 | 64.3 | 28.3 | 29.9 | 28.8 | **24.5** | 27.3 | 25.5 |
| **Average** | 84.4 | 52.7 | 53.0 | 51.0 | 49.0 | 41.9 | 90.7 | 48.9 | 44.8 | 46.0 | 56.3 | 43.6 | 47.8 | 42.5 | **34.9** | 41.8 | 38.9 |

tain. In Table 5 we report the number of "failed" unit tests (a unit test counts as failed when a detector shows FPR values above 10%) and observe that normalization, in particular *Mahalanobis++* remedies this effectively. For results on all models, we refer to Table 17 Appendix.

*Table 5.* **Normalization improves robustness against noise distributions.** We report the number of failed unit tests (noise distributions with FPR values $\geq 10\%$) from Bitterwolf et al. (2023). Normalization remedies the brittleness of Mahalanobis-based detectors. Full Table in Appendix E.

| model | ConvNeXtV2 | SwinV2 | ViT-CLIP |
|---|---|---|---|
| Maha | 5/17 | 10/17 | 14/17 |
| Maha++ | **0/17** | **0/17** | **0/17** |

**CIFAR** We investigate *Mahalanobis++* on CIFAR100 (Krizhevsky, 2009), following the OpenOOD setup with tiny

ImageNet (Le & Yang, 2015), Mnist (LeCun et al., 1998), SVHN (Netzer et al., 2011), Texture (Cimpoi et al., 2014), Places (Zhou et al., 2017) and Cifar10 as OOD datasets for a range of architectures and training schemes.

We report results averaged across the OOD datasets in Table 3 for the most competitive methods and standard baselines (full results in Appendix E). We observe that *Mahalanobis++* consistently outperforms the conventional Mahalanobis distance, but the differences are smaller compared to the ImageNet setup. We hypothesize that this is because the problems of the Mahalanobis distance are less drastic at a smaller scale, and therefore the conventional Mahalanobis distance is already fairly effective for OOD detection. ViM and KNN are the most competitive baseline methods, but *Mahalanobis++* remains the most consistent and effective method across models.

*Table 6.* FPR on NINCO, Green indicates that normalized method is better than its unnormalized counterpart, **bold** indicates the best method, and underlined indicates second best method. Maha++ improves over Maha on average by 10.9% in FPR over all models. Similarly, rMaha++ is 6.0% better in FPR than rMaha. In total, Maha++ improves the SOTA compared to the strongest competitor rMaha among all OOD models by 6.3% which is significant. The lowest FPR is achieved by Maha++ for the ConvNeXtV2-L-In21k highlighted in blue.

| Model | Val Acc | MSP | E | E+R | ML | ViM | AshS | KNN | NNG | NEC | GMN | GEN | fDBD | Maha | Maha++ | rMaha | rMaha++ |
|---|---|---|---|---|---|---|---|---|---|---|---|---|---|---|---|---|---|
| ConvNeXt-B-In21k | 86.3 | 46.2 | 43.0 | 40.0 | 40.5 | 41.2 | 92.7 | 51.6 | 41.2 | 35.2 | 53.8 | 38.0 | 48.3 | 48.7 | **28.8** | 40.2 | 32.5 |
| ConvNeXt-B | 84.4 | 64.1 | 89.4 | 86.2 | 71.4 | 64.7 | 99.6 | 70.1 | 62.2 | 68.2 | 68.3 | 65.9 | 68.6 | 64.6 | 50.5 | 59.5 | **49.7** |
| ConvNeXtV2-T-In21k | 85.1 | 51.6 | 42.4 | 42.4 | 44.1 | 34.5 | 97.5 | 54.1 | 45.0 | 38.7 | 52.5 | 44.2 | 51.4 | 40.9 | **32.8** | 40.0 | 36.8 |
| ConvNeXtV2-B-In21k | 87.6 | 41.4 | 30.1 | 29.5 | 31.9 | 26.9 | 95.8 | 40.9 | 31.7 | 27.7 | 40.6 | 29.7 | 37.0 | 30.3 | **22.4** | 28.1 | 24.7 |
| ConvNeXtV2-L-In21k | 88.2 | 38.7 | 30.7 | 29.8 | 31.5 | 37.2 | 96.0 | 38.7 | 29.9 | 27.0 | 43.4 | 29.0 | 36.6 | 34.6 | **18.4** | 27.7 | 21.4 |
| ConvNeXtV2-T | 83.5 | 66.1 | 73.3 | 68.4 | 65.7 | 64.4 | 99.2 | 82.3 | 73.9 | 61.7 | 72.3 | 64.1 | 71.6 | 66.1 | 52.3 | 58.6 | **49.7** |
| ConvNeXtV2-B | 85.5 | 62.9 | 73.9 | 69.8 | 63.5 | 61.1 | 99.4 | 67.3 | 60.3 | 60.7 | 69.2 | 57.1 | 65.1 | 58.9 | 44.7 | 52.4 | **44.1** |
| ConvNeXtV2-L | 86.1 | 63.8 | 72.3 | 66.8 | 63.6 | 62.4 | 99.5 | 62.4 | 56.9 | 61.9 | 71.2 | 55.6 | 59.7 | 53.8 | 43.0 | 47.1 | **42.1** |
| DeiT3-S16-In21k | 84.8 | 68.7 | 61.8 | 59.8 | 62.6 | 60.6 | 99.7 | 62.9 | 59.3 | 60.1 | 58.5 | 58.7 | 65.8 | 62.4 | 50.8 | 59.8 | 50.9 |
| DeiT3-B16-In21k | 86.7 | 61.0 | 55.9 | 50.9 | 55.2 | 55.3 | 99.5 | 52.6 | 46.4 | 51.5 | 52.2 | 45.2 | 53.5 | 52.5 | 38.8 | 47.4 | **38.3** |
| DeiT3-L16-In21k | 87.7 | 59.7 | 46.2 | 41.8 | 48.9 | 45.7 | 98.4 | 43.8 | 37.8 | 42.3 | 43.7 | 38.2 | 46.6 | 42.0 | 33.9 | 38.1 | **32.8** |
| DeiT3-S16 | 83.4 | 64.3 | 63.0 | 63.8 | 60.6 | 54.3 | 84.1 | 75.6 | 57.8 | 60.6 | 66.4 | 57.4 | 65.0 | 61.1 | 53.5 | 56.3 | 50.5 |
| DeiT3-B16 | 85.1 | 66.7 | 87.8 | 89.9 | 72.6 | 59.7 | 99.1 | 74.5 | 80.1 | 71.9 | 66.7 | 57.3 | 67.1 | 63.7 | 57.2 | 58.9 | 53.2 |
| DeiT3-L16 | 85.8 | 67.8 | 82.3 | 86.6 | 70.5 | 57.9 | 81.1 | 67.2 | 77.9 | 70.8 | 62.9 | 58.4 | 64.4 | 57.0 | 50.4 | 52.0 | **46.6** |
| EVA02-B14-In21k | 88.7 | 35.8 | 28.2 | 27.4 | 30.9 | 28.7 | 92.7 | 37.6 | 30.0 | 27.3 | 39.0 | 25.8 | 32.3 | 31.7 | 23.8 | 30.3 | 25.9 |
| EVA02-L14-M38m-In21k | 90.1 | 29.0 | 24.3 | 24.1 | 25.7 | 21.4 | 94.8 | 30.3 | 26.1 | 22.9 | 39.5 | 20.3 | 26.0 | 22.2 | 18.6 | 22.1 | 20.1 |
| EVA02-T14 | 80.6 | 72.7 | 74.5 | 75.2 | 72.1 | 67.7 | 98.8 | 74.5 | 71.0 | 68.4 | 65.6 | 70.5 | 73.5 | 65.9 | 64.0 | 65.9 | 64.4 |
| EVA02-S14 | 85.7 | 61.2 | 61.4 | 61.5 | 57.8 | 51.6 | 98.9 | 60.0 | 54.0 | 53.2 | 51.9 | 53.1 | 60.0 | 49.3 | 48.0 | 49.1 | **47.8** |
| EffNetV2-S | 83.9 | 67.7 | 77.5 | 73.3 | 69.5 | 74.0 | 99.7 | 60.9 | 59.6 | 69.4 | 79.9 | 62.9 | 67.9 | 67.5 | 59.9 | 59.2 | 52.1 |
| EffNetV2-L | 85.7 | 63.7 | 77.2 | 68.8 | 64.3 | 69.4 | 98.9 | 62.5 | 60.1 | 63.5 | 64.4 | 56.3 | 62.4 | 58.4 | 47.8 | 50.8 | 44.3 |
| EffNetV2-M | 85.2 | 63.4 | 75.1 | 69.1 | 63.8 | 72.3 | 99.6 | 63.1 | 60.6 | 63.3 | 67.5 | 56.3 | 64.5 | 61.7 | 50.0 | 52.2 | 45.3 |
| Mixer-B16-In21k | 76.6 | 77.4 | 83.4 | 83.5 | 79.5 | 78.0 | 94.8 | 85.8 | 83.7 | 79.8 | 66.7 | 75.9 | 80.3 | 73.4 | 65.4 | 70.3 | **63.1** |
| SwinV2-B-In21k | 87.1 | 48.2 | 38.3 | 35.1 | 38.6 | 50.4 | 86.0 | 57.2 | 42.7 | 32.6 | 56.3 | 37.0 | 50.5 | 58.2 | 31.3 | 48.2 | 34.4 |
| SwinV2-L-In21k | 87.5 | 45.9 | 38.7 | 35.4 | 38.6 | 55.3 | 89.9 | 55.1 | 41.7 | 32.3 | 63.1 | 36.5 | 50.5 | 57.4 | **28.3** | 47.6 | 32.2 |
| SwinV2-S | 84.2 | 67.6 | 71.7 | 70.1 | 66.7 | 66.8 | 99.8 | 73.1 | 66.8 | 62.7 | 61.0 | 63.8 | 73.4 | 68.0 | 49.8 | 63.7 | 48.5 |
| SwinV2-B | 84.6 | 69.5 | 72.6 | 69.3 | 67.4 | 66.6 | 97.8 | 69.4 | 65.2 | 64.2 | 60.7 | 62.0 | 70.5 | 63.3 | 52.2 | 59.1 | 50.2 |
| ResNet101 | 81.9 | 73.4 | 85.2 | 100.0 | 76.1 | 75.8 | 89.9 | 74.9 | 66.4 | 77.2 | 83.5 | 72.5 | 84.5 | 66.8 | 50.4 | 55.8 | 53.5 |
| ResNet152 | 82.3 | 71.2 | 83.2 | 100.0 | 74.4 | 74.6 | 88.1 | 72.0 | 61.6 | 75.0 | 79.5 | 69.9 | 82.4 | 64.9 | 46.5 | 52.7 | 52.2 |
| ResNet50 | 80.9 | 76.0 | 94.7 | 99.9 | 78.6 | 79.6 | 89.9 | 83.7 | 75.0 | 80.0 | 89.1 | 75.0 | 85.7 | 69.9 | 61.0 | 58.2 | 56.9 |
| ResNet50-supcon | 78.7 | 60.6 | 57.0 | 56.1 | 56.8 | 84.6 | 59.1 | 65.8 | 58.4 | 56.9 | 80.3 | 60.0 | 63.9 | 98.3 | 59.6 | 90.7 | 61.8 |
| ViT-T16-In21k-augreg | 75.5 | 79.0 | 72.9 | 69.6 | 74.0 | 66.4 | 90.1 | 81.7 | 81.9 | 71.1 | 69.4 | 78.4 | 72.3 | **61.6** | 63.2 | 67.6 | 68.0 |
| ViT-S16-In21k-augreg | 81.4 | 67.0 | 53.3 | 55.4 | 54.7 | 47.4 | 85.0 | 70.9 | 64.0 | 51.9 | 58.3 | 61.4 | 57.6 | 44.8 | **44.6** | 51.1 | 50.5 |
| ViT-B16-In21k-augreg2 | 85.1 | 62.1 | 52.1 | 49.4 | 54.6 | 71.0 | 98.7 | 64.9 | 57.0 | 51.3 | 65.2 | 53.4 | 64.4 | 69.8 | 45.9 | 58.5 | 44.5 |
| ViT-B16-In21k-augreg | 84.5 | 56.8 | 45.2 | 48.9 | 45.4 | 38.1 | 94.1 | 67.7 | 59.0 | 43.1 | 50.1 | 48.2 | 49.8 | 31.3 | 35.7 | 35.2 | 37.1 |
| ViT-B16-In21k-orig | 81.8 | 52.2 | 39.2 | 39.0 | 41.0 | 35.6 | 71.4 | 52.7 | 47.6 | 38.3 | 56.9 | 48.2 | 44.5 | 35.6 | **31.6** | 38.3 | 36.5 |
| ViT-B16-In21k-miil | 84.3 | 57.2 | 46.4 | 46.5 | 49.3 | 46.1 | 98.0 | 59.6 | 51.7 | 43.6 | 59.4 | 50.0 | 58.1 | 56.2 | 35.4 | 48.6 | 40.2 |
| ViT-L16-In21k-augreg | 85.8 | 47.0 | 39.0 | 27.3 | 37.7 | 31.7 | 95.2 | 68.6 | 58.9 | 35.4 | 52.7 | 37.6 | 40.3 | 24.2 | 28.9 | 26.5 | 28.1 |
| ViT-L16-In21k-orig | 81.5 | 46.2 | 37.3 | 37.1 | 37.7 | 42.2 | 58.5 | 45.8 | 40.7 | 36.4 | 55.8 | 42.1 | 40.1 | 39.4 | 32.4 | 37.6 | 36.1 |
| ViT-S16-augreg | 78.8 | 72.8 | 72.8 | 73.6 | 72.5 | 80.7 | 97.0 | 82.1 | 80.0 | 72.7 | 71.0 | 72.8 | 75.4 | 63.2 | 63.1 | 59.2 | **58.9** |
| ViT-B16-augreg | 79.2 | 72.2 | 71.7 | 69.6 | 71.1 | 73.5 | 90.9 | 77.6 | 75.9 | 70.9 | 65.5 | 72.0 | 73.8 | 62.9 | 61.3 | 58.4 | **57.4** |
| ViT-B16-CLIP-L2b-In12k | 86.2 | 49.7 | 44.7 | 42.6 | 44.0 | 49.6 | 99.4 | 49.4 | 42.3 | 40.9 | 50.5 | 41.4 | 48.4 | 57.2 | 35.8 | 49.0 | 38.9 |
| ViT-L14-CLIP-L2b-In12k | 88.2 | 35.5 | 28.8 | 28.1 | 29.9 | 24.5 | 97.9 | 39.5 | 25.4 | 29.7 | 41.5 | 26.8 | 31.4 | 35.3 | 25.4 | 30.3 | 27.2 |
| ViT-H14-CLIP-L2b-In12k | 88.6 | 36.4 | 31.1 | 30.8 | 31.6 | 24.9 | 97.0 | 41.7 | 27.4 | 31.5 | 53.4 | 27.4 | 33.6 | 33.5 | 23.7 | 29.5 | 26.1 |
| ViT-so400M-SigLip | 89.4 | 50.3 | 47.4 | 42.1 | 44.2 | 40.2 | 95.5 | 36.3 | 30.0 | 42.6 | 65.1 | 30.9 | 36.1 | 36.4 | 27.4 | 31.3 | **26.1** |
| **Average** | 84.4 | 58.9 | 58.6 | 57.6 | 55.2 | 54.9 | 92.9 | 61.5 | 55.1 | 52.9 | 61.0 | 52.0 | 58.1 | 53.8 | 42.9 | 49.2 | 43.2 |

## 6. Conclusion

We showed that the frequently occurring failure cases of the Mahalanobis distance as an OOD detection method are related to violations of the method's basic assumptions. We showed that the feature norms vary much stronger than expected under a Gaussian model, that the feature distributions are strongly heavy-tailed and that feature norms correlate with the Mahalanobis score - irrespective of whether a sample is ID or OOD. These insights explain why certain models - despite impressive ID classification performance - showed strongly degraded OOD detection results with the Mahalanobis score in previous studies (Bitterwolf et al., 2023). We introduced *Mahalanobis++*, a simple remedy consisting of $\ell_2$ normalization that effectively mitigates those problems. In particular, the resulting feature distributions are more aligned with a normal distribution, less heavy-tailed, and the class variances are more similar, leading to improved OOD detection results across a wide range of models. *Mahalanobis++* outperforms the conventional Mahalanobis distance in 41/44 cases, rendering it clearly the most effective method across models. It outperforms the previously best baseline ViM by 7 FPR points on average on the OpenOOD datasets, and is the best method for 4 of the 5 top models.

## Impact Statement

This paper presents work whose goal is to advance the field of Machine Learning. There are many potential societal consequences of our work, none of which we feel must be specifically highlighted here.

## Acknowledgements

We thank Yannic Neuhaus for insightful discussions about distances on unit spheres. Further, we acknowledge support from the DFG (EXC number 2064/1, Project number 390727645) and the Carl Zeiss Foundation in the project "Certification and Foundations of Safe Machine Learning Systems in Healthcare". Finally, we acknowledge support from the German Federal Ministry of Education and Research (BMBF) through the Tübingen AI Center (FKZ: 01IS18039A) and the European Laboratory for Learning and Intelligent Systems (ELLIS).

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

## A. Overview

The Appendix is structured as follows:
In Section B we provide the proof for Lemma 3.1. In Section D we provide extended analysis on feature norm and normalization. In particular,

- we show the feature norm distribution for more models in Figure 8
- we provide QQ-plots for more models in Figure 9
- We report the feature norm distribution for ID and OOD data in Figure 10, showing that OOD features can be larger than ID features for off-the-shelf pretrained models
- we highlight that the class variances become more similar to the global variance after normalization in Figure 11
- we plot the correlation between feature norm and OOD-score in Figure 7

In Section E we report extended results. In particular,

- we show additional ImageNet numbers (AUC for NINCO in Table 11, OpenOOD near and far in Table 8 and Table 9, OpenOOD averaged AUC in Table 10)
- we compare cosine-based methods on ImageNet explicitly in Table 12
- we show robustness to noise distributions (unit tests) in Table 17
- we show additional CIFAR numbers (Cifar10 AUC in Table 13 and FPR in Table 14, Cifar100 AUC in Table 15 and FPR in Table 16
- we compare *Mahalanobis++* to SSD+ in Table 18 to highlight the benefits of post-hoc OOD detection methods

In Section F we report details on the model checkpoints used throughout the experiments (ImageNet models in Table 19 and Cifar models in Table 20). In Section G we provide details on the OOD detection methods evaluated in the main paper.

## B. Proof of Lemma 3.1

*Proof.* Let $\Sigma = U\Lambda U^T$ be the eigendecomposition of the covariance matrix with $U$ being an orthogonal matrix containing the eigenvectors of $\Sigma$ and $\Lambda$ the diagonal matrix containing the eigenvalues of $\sigma$. Let $X$ be a random variable with distribution $\mathcal{N}(\mu, \Sigma)$ (in the main paper, we denoted the features as $\Phi(X)$, here we write them as $X$ for notational simplicity). Then it holds $Z = U^T X$ has distribution $\mathcal{N}(U^T\mu, \Lambda)$ and since $U^T$ is an orthogonal matrix: $\|X\|_2^2 = \|Z\|_2^2$. We have

$$\mathbb{E}[\|X\|_2^2] = \mathbb{E}[\|Z\|_2^2] = \sum_{i=1}^d \mathbb{E}[Z_i^2] = \sum_{i=1}^d \mathrm{Var}(Z_i) + \mathbb{E}[Z_i]^2 = \sum_{i=1}^d \lambda_i + \|U^T\mu\|_2^2 = \mathrm{tr}(\Sigma) + \|\mu\|_2^2$$

We note that

$$\mathrm{Var}(\|Z\|_2^2) = \mathbb{E}[\|Z\|_2^4] - \mathbb{E}[\|Z\|_2^2]^2 = \mathbb{E}[\|Z\|_2^4] - (\mathrm{tr}(\Sigma) + \|\mu\|_2^2)^2 \tag{7}$$

and it remains to compute $\mathbb{E}[\|Z\|_2^4]$. We note that

$$\mathbb{E}[\|Z\|_2^4] = \sum_{i=1}^d \mathbb{E}[Z_i^4] + \sum_{i \neq j}^d \mathbb{E}[Z_i^2]\mathbb{E}[Z_j^2] = \sum_{i=1}^d (3\lambda_i^2 + 6\mu_i^2\lambda_i + \mu_i^4) + \left(\sum_{i=1}^d (\lambda_i + \mu_i^2)\right)^2 - \sum_{i=1}^d (\lambda_i + \mu_i^2)^2,$$

where we have used the following calculations:

$$0 = \mathbb{E}[(Z_i - \mu_i)^3] = \mathbb{E}[Z_i^3] - 3\mu_i\lambda_i - \mu_i^3$$
$$3\lambda_i^2 = \mathbb{E}[(Z_i - \mu_i)^4] = \mathbb{E}[Z_i^4] - 4\mu_i\mathbb{E}[Z_i^3] + 6\mu_i^2\mathbb{E}[Z_i^2] - 3\mu_i^4$$

and thus

$$\mathbb{E}[Z_i^3] = 3\mu_i\lambda_i + \mu_i^3$$
$$\mathbb{E}[Z_i^4] = 3\lambda_i^2 + 6\mu_i^2\lambda_i + \mu_i^4$$

This yields

$$\mathrm{Var}(\|Z\|_2^2) = \sum_{i=1}^d (3\lambda_i^2 + 6\mu_i^2\lambda_i + \mu_i^4) - \sum_{i=1}^d (\lambda_i + \mu_i^2)^2$$

Applying Chebychev's inequality yields the result. $\square$

## C. Derivation of expected squared relative variance deviation

Here we want to derive the statement about the expected squared relative variance (denoting the covariance matrix as $C$ instead of $\Sigma$):

$$\mathbb{E}_u[(u^T C^{-\frac{1}{2}}(C_i - C)C^{-\frac{1}{2}}u)^2] = \frac{2\mathrm{trace}(A^2) + \mathrm{trace}(A)^2}{d(d+2)}, \tag{8}$$

where $u$ has a uniform distribution on the unit sphere and $A = C^{-\frac{1}{2}}(C_i - C)C^{-\frac{1}{2}}$. We note that $A$ is symmetric and thus has an eigendecomposition $A = U\Lambda U^T$. We have

$$\mathbb{E}_u[(u^T A u)^2] = \mathbb{E}_u[((U^T u)^T \Lambda (U^T u))^2] = \mathbb{E}_u[(u^T \Lambda u)^2] = \sum_{i=1}^d \lambda_i^2 \mathbb{E}_u[u_i^4] + \sum_{i \neq j} \lambda_i \lambda_j \mathbb{E}_u[u_i^2 u_j^2]$$

It remains to compute these moments on the unit sphere. For this purpose we note that $\|u\|_2^2 = \sum_{i=1}^d u_i^2 = 1$ and thus

$$1 = \|u\|_2^4 = \left(\sum_{i=1}^d u_i^2\right)^2 = \sum_{i=1}^d u_i^4 + \sum_{i \neq j} u_i^2 u_j^2$$

We note that $u_i^4$ for $i = 1, \ldots, d$ and $u_i^2 u_j^2$ for $i \neq j$ are all equally distributed and thus for $i \neq j$

$$1 = d\mathbb{E}[u_i^4] + d(d-1)\mathbb{E}[u_i^2 u_j^2] \tag{9}$$

Moreover, we note that rotations do not change the distribution for a unifom distribution on the sphere and thus $(u_i, u_j)$ and $\left(\frac{u_i - u_j}{\sqrt{2}}, \frac{u_i + u_j}{\sqrt{2}}\right)$ have the same distribution and

$$\mathbb{E}[u_i^2 u_j^2] = \mathbb{E}\left[\left(\frac{u_i - u_j}{\sqrt{2}}\right)^2 \left(\frac{u_i + u_j}{\sqrt{2}}\right)^2\right] = \frac{1}{2}\mathbb{E}[u_i^4] - \frac{1}{2}\mathbb{E}[u_i^2 u_j^2].$$

This yields $\mathbb{E}[u_i^4] = 3\mathbb{E}[u_i^2 u_j^2]$. Plugging this into (9) yields

$$\mathbb{E}[u_i^4] = \frac{3}{d(d+2)}, \quad \mathbb{E}[u_i^2 u_j^2] = \frac{1}{d(d+2)}$$

Thus

$$\mathbb{E}_u[(u^T A u)^2] = \frac{1}{d(d+2)}\left(3\sum_{i=1}^d \lambda_i^2 + \sum_{i \neq j} \lambda_i \lambda_j\right) = \frac{1}{d(d+2)}\left(2\sum_{i=1}^d \lambda_i^2 + \sum_{i,j=1}^d \lambda_i \lambda_j\right)$$

Using that $\mathrm{trace}(A) = \sum_{i=1}^d \lambda_i$ finishes the derivation.

## D. Extended Analysis

Here, we report extended results on the experiments of Section 3 of the main paper. In particular, we show that the observations made hold beyond the SwinV2 model. If not stated differently, all experiments are with ImageNet as ID dataset and NINCO as OOD dataset.

**Feature Norm Correlation.** In the main paper, we showed that for a SwinV2 model, the feature norm of a sample correlates strongly with the OOD score received via the Mahalanobis distance. Here, we show this phenomenon for more models. In Figure 7, we plot the feature norm against the OOD score assigned by Mahalanobis and *Mahalanobis++* for four models. For SwinV2, ConvNext and ViT-clip, the feature norms correlate strongly with the OOD score. Normalizing the features (bottom) mitigates this dependency, as OOD samples with small feature norms are detected as OOD, and thus improves OOD detection strongly. A notable exception is the augreg-ViT, for which there is no correlation between feature norms and OOD score. In Figure 6, we further investigate the dependency of the Mahalanobis score on the feature norm by artificially scaling the feature norm of the OOD features with a prefactor $\alpha$. That is, for a SwinV2 model, we use $\alpha * \phi_i$ for each OOD feature and leave the ID validation features unchanged. We report the FPR values against $\alpha$ in Figure 6, and again observe a clear correlation between the scaling factor and the FPR: Perhaps unexpectedly, upscaling the features reduces the false-positive rate, up to a scaling factor of 2, where zero false positives are achieved. When scaling down the feature norm, the FPR increases, and for $\alpha \approx 0.5$, i.e. at half the original feature norm, all OOD samples are identified as ID. Notably, this does not change for smaller $\alpha$ values, not even for $\alpha = 0$, where all OOD samples collapse to the zero vector. In other words, everything in the vicinity of the origin is identified as in-distribution, which contradicts the intuition of tight Gaussian clusters centered around class means. In the main paper, we hypothesized that this might explain why the Mahalanobis distance sometimes fails to detect the unit tests since those might receive a small feature norm. In Figure 10, we plot the feature norms of different datasets for a range of models with (top) and without (bottom) pretraining. We find that the feature norms of natural OOD images like those from NINCO tend to be even larger than the ImageNet feature norms. This violates basic assumptions in feature-norm-based OOD detection methods like the negative-aware-norm (Park et al., 2023b), indicating that special training schemes might be necessary for those methods. However, noise distributions like the unit tests from Bitterwolf et al. (2023) can lead to fairly small feature norms for most models. Since we showed that small feature norms lead to small Mahalanobis distances for many models, this highlights why these supposedly easy-to-detect images were not detected with the Mahalanobis distance in previous studies.

**Feature norm distribution.** In Figure 8, we plot the feature norms for four ViTs of exactly the same architecture (ViT-B16). In order to make the plots comparable, we normalize by the average feature norm per model. We observe that, like for the SwinV2 in the main paper, the norms vary strongly across and within classes - except for the augreg-ViT. This model is one of the models that performed well with Mahalanobis "out-of-the-box", i.e., not requiring normalization.

**QQ plots.** In Figure 9 we show QQ plots for four models along three directions in feature space. We observe that the normalized features (green) more closely resemble a normal distribution compared to the unnormalized features (blue), which is best visible via the long tail. The only exception is the augreg-ViT, for which normalized and unnormalized features are similarly close to a Gaussian distribution.

**Variance alignment.** We report extended results on the expected relative deviation scores (see Eq. 5) for more models in Table 7. We observe that for all models - except the ViT-augreg - normalization lowers the deviations, indicating a better alignment of the global variance with the individual class variances. In Figure 11, we illustrate this further: Instead of the score reported in Table 7, which computes an expectation over all directions, we pick three specific directions: 1) a random direction, 2) an eigendirection with a large eigenvalue, and 3) an eigendirection with a small eigenvalue. Ideally, along each direction, the 1000 class variances would coincide with the globally estimated, shared variance. For each direction, we divide the 1000 class variances by the global variance and plot the resulting distribution. Distributions peaked around 1 indicate that the global variance can capture the class variances well. We observe that the distributions of the variances after feature normalization peak more towards one for all models, except the ViT-augreg.

**Augreg ViTs.** The ViTs that showed the best performance with Mahalanobis distance in previous studies were base-size ViTs pretrained on ImageNet21k and fine-tuned on ImageNet1k by Steiner et al. (2022). The training scheme is called *augreg*, a carefully tuned combination of augmentation and regularization methods. In this paper, we made several observations regarding those models (applies for both base-size and large-size models with pretraining on ImageNet21k). In particular, they

- show strong OOD detection performance with Mahalanobis distance without normalization, and normalization does

*Table 7.* **Deviations from global variance.** We report the mean squared relative variance deviation as defined in Equation (5) for multiple models. In all cases, except for the ViT-augreg, normalization significantly improves the fit of the global covariance matrix to the covariance structure of the individual classes. As noted in the text for the ViT-augreg the features already follow very well the assumptions of the Mahalanobis score and normalization leads to no improvements.

| model | unnormalized | normalized |
|---|---|---|
| ViT-B16-In21k-augreg | 0.05 | 0.05 |
| SwinV2-B-In21k | 0.26 | 0.12 |
| ViT-B16-CLIP-L2b-In12k | 0.17 | 0.08 |
| ViT-B16-In21k-augreg2 | 0.14 | 0.07 |
| ViT-B16-In21k-miil | 0.12 | 0.09 |
| DeiT3-B16-In21k | 0.24 | 0.15 |
| ConvNeXt-B-In21k | 0.17 | 0.11 |
| EVA02-B14-In21k | 0.21 | 0.14 |
| ConvNeXtV2-B-In21k | 0.23 | 0.18 |
| ConvNeXtV2-B | 0.22 | 0.14 |
| ConvNeXt-B-In21k | 0.17 | 0.11 |
| ConvNeXt-B | 0.22 | 0.12 |

not improve Mahalanobis-based OOD detection

- show little variations in feature norm compared to all other investigated models

- show no correlation between feature norm and Mahalanobis score (in contrast to all other investigated models)

- show much weaker heavy tails than the other models

- show low values for the variance deviation metric

- loose their advantage for unnormalized Mahalanobis-based detection when the fine-tuning scheme is changed (the augreg2 model is fine-tuned from a 21k-augreg-checkpoint, but the fine-tuning scheme differs in learning rate and augmentations)

In short, the augreg models omit all the points that we identified as problematic for Mahalanobis-based OOD detection. This indicates that the augreg training scheme induces a feature space that lends itself naturally towards a normal distribution, aligning well with the assumptions of the Mahalanobis distance as OOD detection method. Understanding the exact reason why the augreg scheme induces those features is beyond the scope of this paper. The connection of training hyperparameters and OOD detection performance was, however, investigated by Mueller & Hein (2024). It should be stressed that for post-hoc OOD detection, we ideally want a method that works well with *all* models, not only those obtained via a certain training scheme. We provide such a method with *Mahalanobis++*.

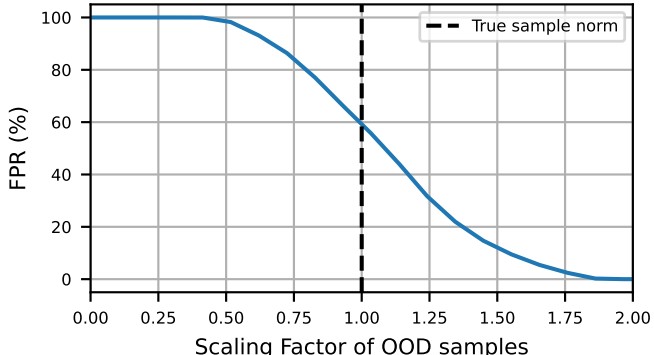

*Figure 6.* **Impact of the feature norm of OOD samples on their Mahalanobis score.** When scaling down the norm of the features while leaving the feature direction unchanged, OOD samples receive a smaller Mahalanobis score and are incorrectly classified as ID samples. When the feature norm is artificially increased, the opposite happens.

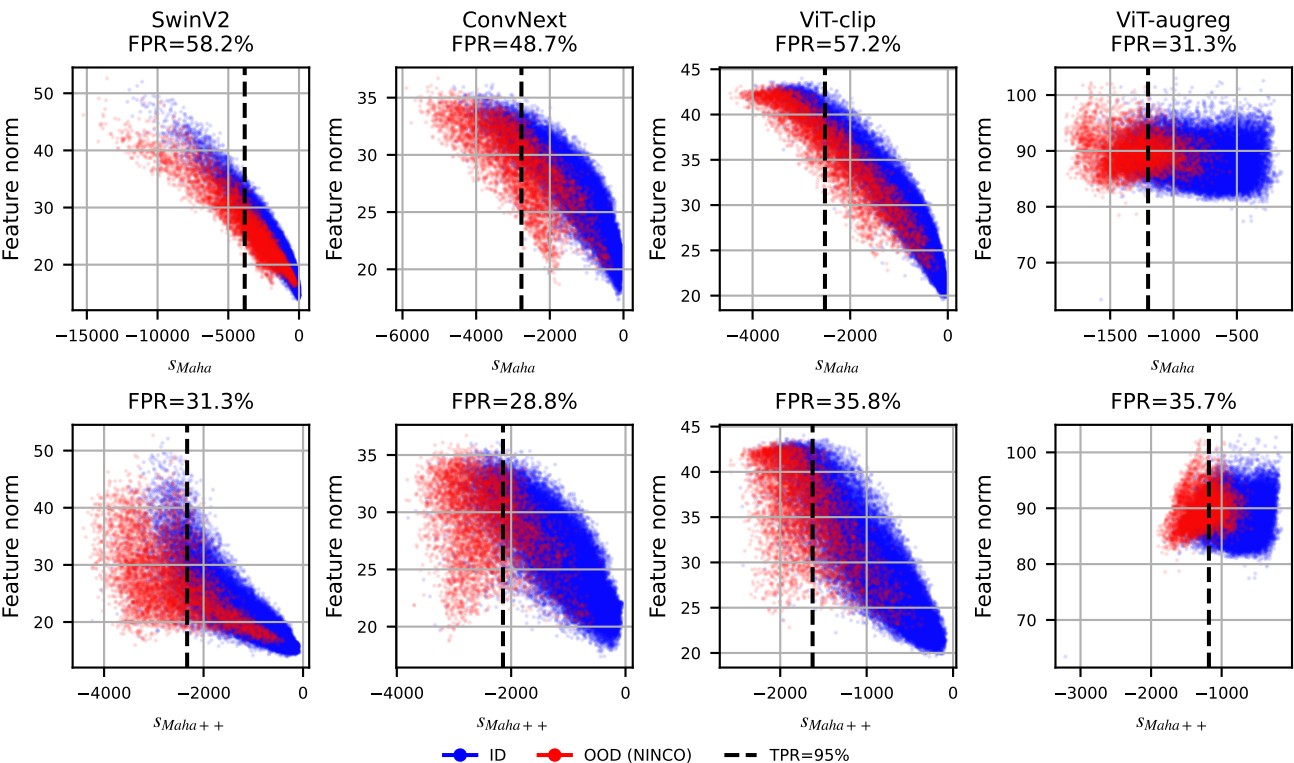

*Figure 7.* **Mahalanobis++ resolves feature-norm dependency of Mahalanobis score.** With unnormalized features, OOD samples with small pre-logit feature norm were systematically identified as ID, but after normalization, OOD samples with small feature norm are rightfully detected as OOD, resulting in significantly improved OOD detection with *Mahalanobis++*. The only exception is an *augreg* ViT, which does not show a correlation between feature norm and Mahalanobis score, even without normalization.

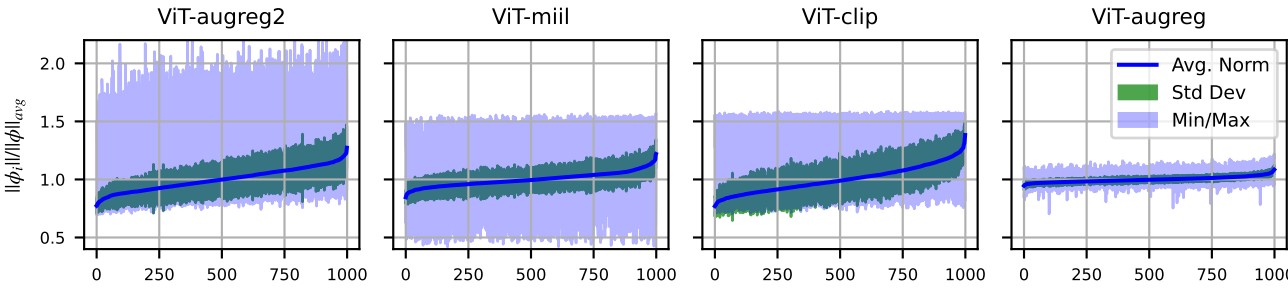

*Figure 8.* **For most models, the feature norms vary strongly across and within classes.** The same plot as the *"observed"* part of Figure 5 in the main paper, but normalized by the global average feature norm to make the scales of different models comparable. We thus show how strongly the feature norms vary relative to their scale. We report results for ViT-B16 models with different pretraining schemes. Only the *augreg* ViT shows little variation in feature norm and is the only model that does not benefit from normalization. Interestingly, the *augreg2* model was finetuned on ImageNet-1k from the *same* 21k-checkpoint as the *augreg* model and even achieves higher classification accuracy, but shows a very different feature norm distribution - which reflects in the OOD detection performance with Mahalanobis and *Mahalanobis++*: All models except for the *augreg* model benefit from normalization.

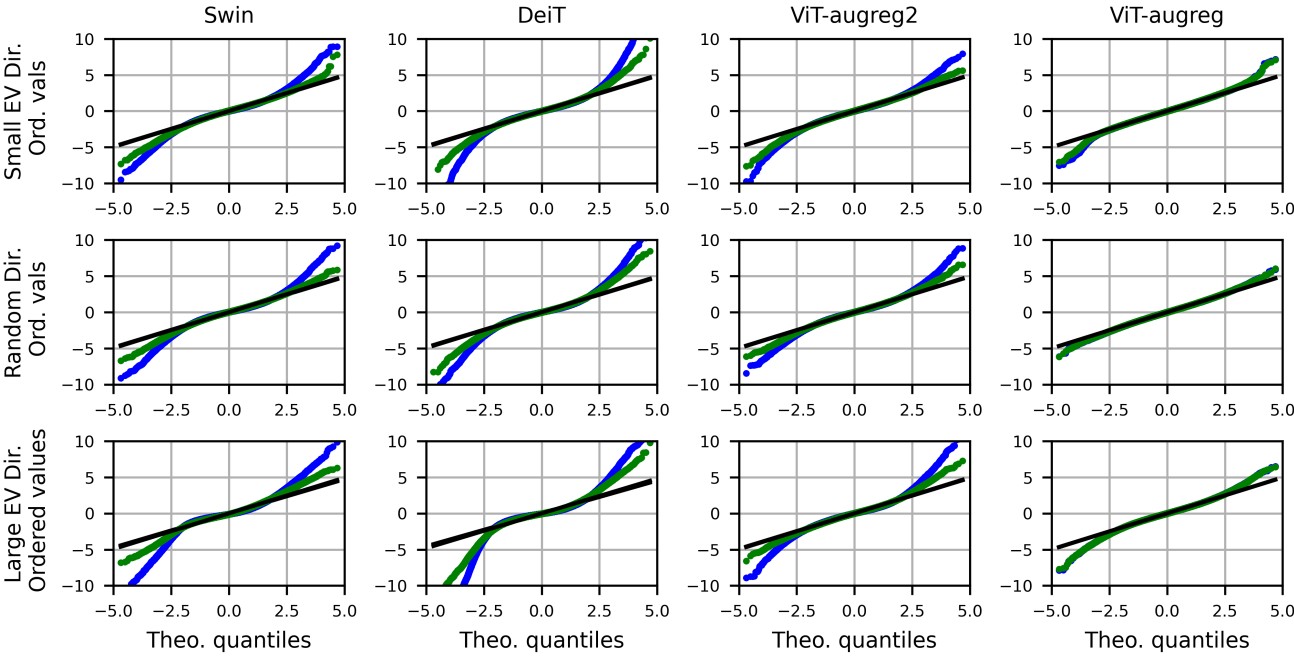

*Figure 9.* **QQ-plot: $\ell_2-$normalization helps transform the features to be more aligned with a normal distribution.** Normalized features in green, unnormalized features in blue. For a SwinV2, DeiT3 and ViT-augreg2, the feature norms vary strongly across classes (see e.g. Fig. 3 and Fig. 8) and normalization shifts the distribution towards a Gaussian. For a ViT-B-augreg the feature norms are similar across classes (see Fig 8) and the feature norms are already fairly normal, so $\ell_2$-normalization has almost no effect.

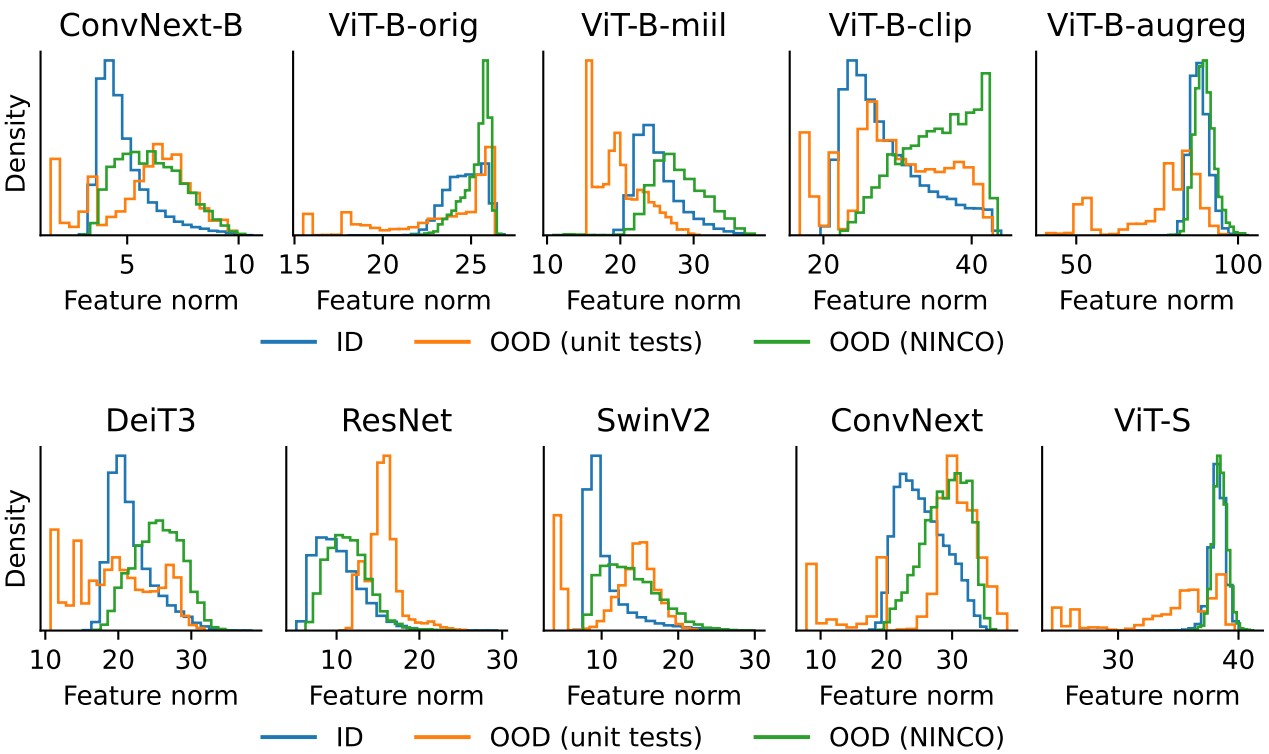

Figure 10. **Feature norm distribution.** In contrast to previous work (e.g., (Park et al., 2023b)), we find that the feature norm of natural OOD samples (NINCO in green) is often larger than that of ID samples (orange). Far-OOD data, like noise distributions, tend to have lower feature norms. This holds for models with (top) and without (bottom) pretraining.

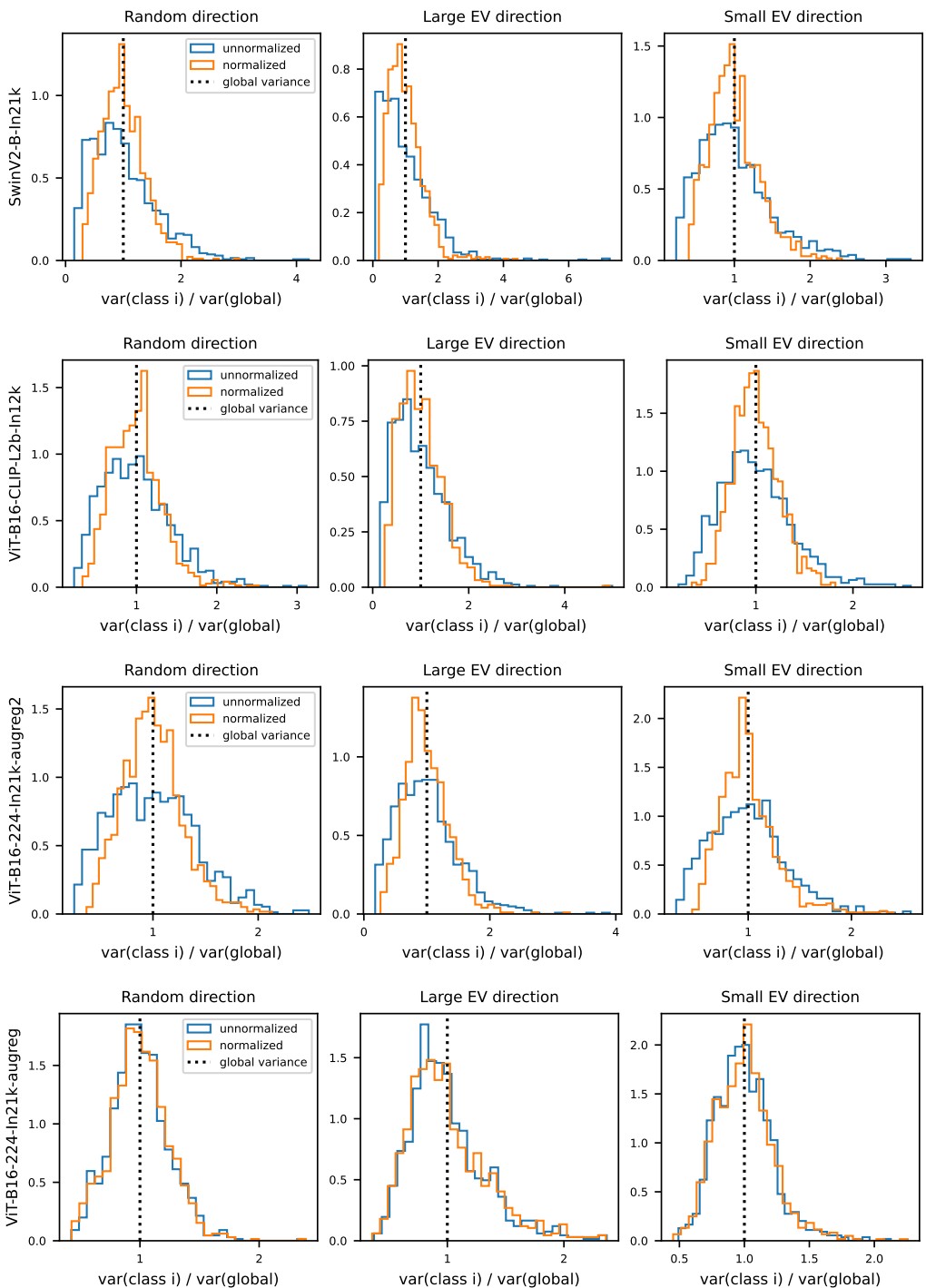

*Figure 11.* ***Mahalanobis++ aligns class-variances.*** We report the distribution of the variances of the train features for each class along three directions: 1) a random direction, 2) a large eigendirection, 3) a small eigendirection. For each class, we compute the variance divided by the global variance, and plot the resulting distributions. Larger deviations from one indicate larger deviations of the class variance from the global variance. For all directions the distribution of variances is more peaked around 1 after normalization, indicating that after normalization the shared variance assumption is more appropriate - except for the ViT-augreg.

# E. Extended results

*Table 8.* FPR on OpenOOD-near datasets, Green indicates that normalized method is better than its unnormalized counterpart, **bold** indicates the best method, and underlined indicates second best method. Maha++ improves over Maha on average by 9.6% in FPR over all models. Similarly, rMaha++ is 5.5% better in FPR than rMaha. The lowest FPR is achieved by ViM for the EVA02-L14-M38m-In21k highlighted in blue, closely followed by Maha++ for the same model.

| Model | Val Acc | MSP | E | E+R | ML | ViM | AshS | KNN | NNG | NEC | GMN | GEN | fDBD | Maha | Maha++ | rMaha | rMaha++ |
|---|---|---|---|---|---|---|---|---|---|---|---|---|---|---|---|---|---|
| ConvNeXt-B-In21k | 86.3 | 53.9 | 46.1 | 44.2 | 46.7 | 53.8 | 91.3 | 63.1 | 52.7 | 42.2 | 53.8 | 46.7 | 58.3 | 59.6 | **41.7** | 51.8 | 44.7 |
| ConvNeXt-B | 84.4 | 70.0 | 89.0 | 86.8 | 75.1 | 72.1 | 99.0 | 77.5 | 72.1 | 72.1 | 72.9 | 73.3 | 76.1 | 72.0 | 59.1 | 67.3 | **58.1** |
| ConvNeXtV2-T-In21k | 85.1 | 59.7 | 51.2 | 51.5 | 53.3 | 45.4 | 95.7 | 62.7 | 55.6 | 48.3 | 56.2 | 54.4 | 61.7 | 52.7 | **44.9** | 52.1 | 48.8 |
| ConvNeXtV2-B-In21k | 87.6 | 50.8 | 39.0 | 38.8 | 41.5 | 37.3 | 95.5 | 49.9 | 42.6 | 36.8 | 46.1 | 41.0 | 48.7 | 42.0 | **34.7** | 41.3 | 38.2 |
| ConvNeXtV2-L-In21k | 88.2 | 48.4 | 38.5 | 38.6 | 40.6 | 48.8 | 95.6 | 49.5 | 41.4 | 35.9 | 54.9 | 39.7 | 49.5 | 46.6 | **31.2** | 41.2 | 35.6 |
| ConvNeXtV2-T | 83.5 | 72.7 | 79.1 | 75.8 | 72.9 | 72.6 | 98.6 | 86.7 | 81.0 | 69.7 | 74.6 | 72.4 | 78.4 | 73.3 | 60.5 | 66.6 | **58.2** |
| ConvNeXtV2-B | 85.5 | 69.6 | 79.2 | 76.3 | 70.5 | 69.0 | 99.0 | 74.9 | 70.3 | 68.2 | 71.1 | 65.8 | 72.6 | 66.9 | 55.0 | 61.5 | **54.1** |
| ConvNeXtV2-L | 86.1 | 69.8 | 77.4 | 73.7 | 69.9 | 70.6 | 98.4 | 70.3 | 66.8 | 68.8 | 77.8 | 64.1 | 68.2 | 62.4 | 53.6 | 56.4 | **51.9** |
| DeiT3-S16-In21k | 84.8 | 73.1 | 65.8 | 64.8 | 67.4 | 68.7 | 98.8 | 70.8 | 67.1 | 65.4 | 64.5 | 65.5 | 72.6 | 70.7 | **60.4** | 68.4 | 60.6 |
| DeiT3-B16-In21k | 86.7 | 67.1 | 60.6 | 57.4 | 61.2 | 65.5 | 98.9 | 63.2 | 57.4 | 58.1 | 59.1 | 53.4 | 62.2 | 62.7 | 50.8 | 58.6 | **49.9** |
| DeiT3-L16-In21k | 87.7 | 66.7 | 54.3 | 51.2 | 57.1 | 57.3 | 97.2 | 53.9 | 49.4 | 52.2 | 52.9 | 48.1 | 56.9 | 54.5 | 47.3 | 51.6 | **46.3** |
| DeiT3-S16 | 83.4 | 70.8 | 71.3 | 71.3 | 68.5 | 63.8 | 86.7 | 81.1 | 67.7 | 68.5 | 70.5 | 66.7 | 72.8 | 69.7 | 62.4 | 65.4 | **60.0** |
| DeiT3-B16 | 85.1 | 71.8 | 89.5 | 90.1 | 76.8 | 67.3 | 98.7 | 79.6 | 83.6 | 76.1 | 71.9 | 65.4 | 74.7 | 71.8 | 64.8 | 67.4 | **61.7** |
| DeiT3-L16 | 85.8 | 72.5 | 83.3 | 86.7 | 74.4 | 65.9 | 85.0 | 75.3 | 80.0 | 74.7 | 67.9 | 65.8 | 72.3 | 66.4 | 60.7 | 61.9 | **57.2** |
| EVA02-B14-In21k | 88.7 | 45.1 | 37.0 | 36.5 | 40.3 | 36.5 | 93.8 | 46.9 | 40.8 | 36.6 | 45.2 | 36.6 | 43.3 | 41.9 | **34.8** | 42.0 | 38.0 |
| EVA02-L14-M38m-In21k | 90.1 | 38.3 | 31.9 | 31.7 | 34.6 | 28.1 | 95.2 | 40.2 | 35.7 | 31.5 | 45.2 | 30.8 | 36.3 | 31.8 | 28.7 | 33.3 | 32.3 |
| EVA02-T14 | 80.6 | 77.6 | 76.9 | 77.3 | 76.5 | 74.6 | 97.8 | 80.6 | 77.3 | 73.5 | **67.5** | 76.4 | 79.2 | 73.0 | 71.0 | 72.7 | 71.2 |
| EVA02-S14 | 85.7 | 67.7 | 67.0 | 67.1 | 64.8 | 58.1 | 98.2 | 67.7 | 62.9 | 60.8 | 56.4 | 61.5 | 68.1 | 58.3 | 57.0 | 58.7 | 57.3 |
| EffNetV2-S | 83.9 | 72.2 | 78.4 | 76.9 | 72.9 | 79.7 | 98.7 | 70.3 | 68.9 | 72.7 | 79.9 | 69.1 | 74.9 | 75.1 | 64.6 | 67.7 | **60.2** |
| EffNetV2-L | 85.7 | 69.0 | 81.4 | 75.6 | 70.1 | 74.7 | 98.6 | 70.0 | 68.6 | 69.4 | 70.5 | 63.5 | 69.9 | 65.8 | 56.4 | 59.5 | **53.8** |
| EffNetV2-M | 85.2 | 68.9 | 78.6 | 75.2 | 69.3 | 77.7 | 99.0 | 71.2 | 69.3 | 68.8 | 70.2 | 64.0 | 72.1 | 69.5 | 58.1 | 61.5 | **54.7** |
| Mixer-B16-In21k | 76.6 | 82.4 | 87.7 | 87.8 | 84.6 | 83.3 | 94.9 | 89.4 | 87.9 | 84.8 | 69.4 | 82.4 | 85.6 | 78.4 | 71.8 | 74.9 | **68.8** |
| SwinV2-B-In21k | 87.1 | 56.0 | 43.1 | 41.9 | 45.6 | 63.1 | 84.5 | 69.8 | 57.6 | 40.7 | 60.9 | 47.0 | 62.5 | 69.8 | 46.7 | 61.1 | 48.4 |
| SwinV2-L-In21k | 87.5 | 54.0 | 45.4 | 44.4 | 46.4 | 66.6 | 87.2 | 67.9 | 57.3 | 41.6 | 72.6 | 47.4 | 62.2 | 69.3 | 44.0 | 60.6 | 46.1 |
| SwinV2-S | 84.2 | 73.4 | 77.5 | 77.0 | 73.0 | 75.4 | 99.7 | 79.8 | 75.7 | 70.2 | 65.5 | 72.0 | 79.7 | 75.3 | 59.2 | 71.1 | **58.0** |
| SwinV2-B | 84.6 | 73.9 | 77.8 | 75.3 | 73.1 | 73.6 | 98.4 | 76.0 | 73.4 | 70.6 | 66.3 | 69.6 | 76.4 | 69.5 | 59.6 | 67.6 | **57.8** |
| ResNet101 | 81.9 | 78.4 | 87.0 | 99.7 | 80.4 | 82.1 | 91.5 | 82.6 | 75.0 | 81.1 | 87.4 | 78.8 | 87.6 | 73.1 | **58.7** | 62.8 | 59.9 |
| ResNet152 | 82.3 | 76.9 | 85.9 | 99.7 | 79.2 | 81.4 | 90.7 | 81.0 | 72.2 | 79.8 | 84.4 | 76.8 | 86.3 | 71.3 | **55.5** | 60.6 | 58.6 |
| ResNet50 | 80.9 | 80.7 | 95.0 | 99.6 | 82.5 | 84.1 | 91.5 | 88.3 | 81.9 | 83.4 | 91.0 | 80.7 | 88.5 | 75.5 | 65.4 | 64.6 | **62.4** |
| ResNet50-supcon | 78.7 | 70.5 | 67.5 | 67.7 | 67.7 | 87.4 | 71.5 | 77.3 | 69.4 | 67.8 | 85.5 | 71.2 | 74.1 | 98.2 | 71.3 | 91.3 | 68.5 |
| ViT-T16-In21k-augreg | 75.5 | 83.7 | 78.7 | 75.2 | 79.8 | 73.5 | 92.1 | 86.9 | 86.5 | 77.4 | 73.1 | 83.5 | 78.6 | **69.7** | 71.6 | 74.3 | 75.0 |
| ViT-S16-In21k-augreg | 81.4 | 73.8 | 61.0 | 63.6 | 63.1 | 56.6 | 88.7 | 77.2 | 72.1 | 60.6 | 62.4 | 69.3 | 66.6 | 56.7 | 56.0 | 61.3 | 60.8 |
| ViT-B16-In21k-augreg2 | 85.1 | 69.4 | 59.1 | 57.4 | 62.4 | 77.8 | 97.4 | 73.4 | 67.4 | 59.7 | 68.6 | 63.5 | 73.2 | 77.4 | 56.8 | 68.1 | **56.0** |
| ViT-B16-In21k-augreg | 84.5 | 64.4 | 54.3 | 58.4 | 54.8 | 47.9 | 95.3 | 74.9 | 68.0 | 52.8 | 55.9 | 57.7 | 58.7 | 44.6 | 49.0 | 48.3 | 49.9 |
| ViT-B16-In21k-orig | 81.8 | 60.6 | 44.8 | 44.9 | 48.0 | 41.4 | 62.5 | 60.1 | 54.6 | 44.9 | 61.2 | 57.1 | 52.6 | 45.3 | 40.6 | 49.9 | 47.8 |
| ViT-B16-In21k-miil | 84.3 | 65.6 | 52.6 | 53.3 | 56.8 | 55.3 | 96.2 | 69.2 | 61.5 | 52.1 | 62.4 | 60.1 | 67.3 | 66.3 | 47.5 | 60.1 | 52.1 |
| ViT-L16-In21k-augreg | 85.8 | 56.5 | 49.6 | 38.3 | 48.9 | 42.6 | 96.3 | 73.2 | 65.9 | 46.5 | 54.2 | 48.7 | 50.8 | **36.4** | 41.9 | 39.2 | 40.8 |
| ViT-L16-In21k-orig | 81.5 | 53.1 | 40.2 | 40.1 | 42.2 | 44.4 | 55.9 | 50.7 | 46.0 | 40.7 | 59.3 | 49.2 | 46.0 | 46.4 | **39.7** | 47.5 | 46.1 |
| ViT-S16-augreg | 78.8 | 78.3 | 79.3 | 80.2 | 78.6 | 85.5 | 96.6 | 86.8 | 85.0 | 78.9 | 75.6 | 78.9 | 81.2 | 69.8 | 69.9 | 66.5 | 66.4 |
| ViT-B16-augreg | 79.2 | 77.7 | 78.1 | 75.5 | 77.4 | 79.0 | 93.2 | 83.1 | 81.4 | 77.3 | 69.2 | 78.0 | 79.8 | 69.4 | 68.3 | 65.4 | 64.7 |
| ViT-B16-CLIP-L2b-In12k | 86.2 | 56.2 | 46.9 | 46.0 | 49.1 | 55.4 | 99.3 | 57.9 | 50.6 | 45.9 | 52.5 | 49.1 | 56.6 | 65.0 | **44.7** | 59.2 | 49.2 |
| ViT-L14-CLIP-L2b-In12k | 88.2 | 44.1 | 34.8 | 34.3 | 37.2 | 32.0 | 96.9 | 48.6 | 33.1 | 37.0 | 46.2 | 36.3 | 41.6 | 45.3 | 35.9 | 41.9 | 39.1 |
| ViT-H14-CLIP-L2b-In12k | 88.6 | 44.5 | 36.8 | 36.7 | 38.7 | 33.6 | 97.0 | 50.5 | 35.5 | 38.6 | 55.5 | 36.9 | 44.1 | 44.4 | 35.8 | 41.6 | 38.9 |
| ViT-so400M-SigLip | 89.4 | 58.6 | 56.1 | 52.5 | 53.4 | 49.9 | 95.3 | 48.6 | 43.7 | 51.3 | 63.9 | 42.8 | 48.9 | 47.0 | **38.6** | 43.7 | 39.2 |
| **Average** | 84.4 | 65.6 | 64.0 | 63.6 | 62.0 | 62.7 | 93.0 | 69.5 | 63.9 | 59.9 | 65.3 | 60.5 | 66.3 | 62.5 | **52.9** | 58.8 | 53.3 |

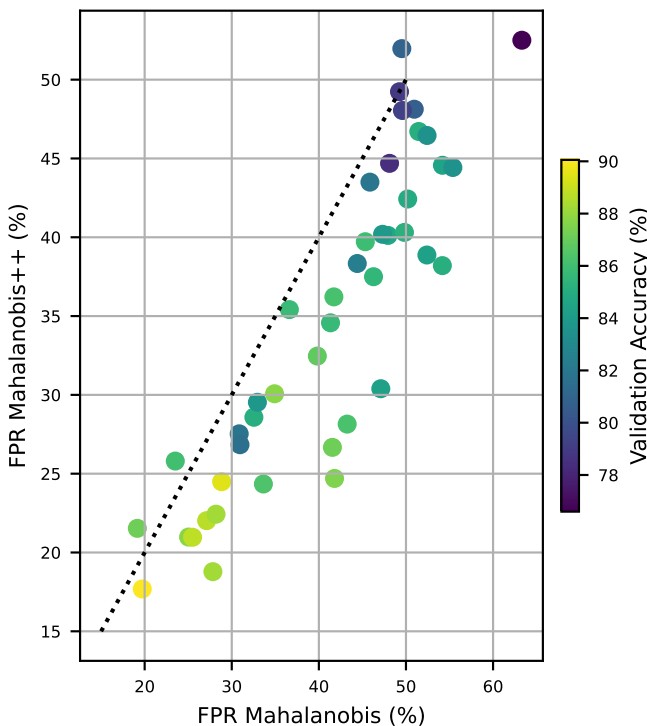

*Figure 12.* We plot the FPR with *Mahalanobis++* against the FPR with the conventional Mahalanobis score averaged over the five OpenOOD datasets. With three minor exceptions, *Mahalanobis++* improves OOD detection performance for all models. In particular, it significantly improves all models with high accuracy.

*Table 9.* FPR on OpenOOD-far datasets, Green indicates that normalized method is better than its unnormalized counterpart, **bold** indicates the best method, and underlined indicates second best method. Maha++ improves over Maha on average by 6.1% in FPR over all models. Similarly, rMaha++ is 1.2% better in FPR than rMaha. In total, Maha++ improves the SOTA compared to the previously strongest method, ViM, by 5.1%, which is significant. The lowest FPR is achieved by Maha++ for the EVA02-L14-M38m-In21k highlighted in blue.

| Model | Val Acc | MSP | E | E+R | ML | ViM | AshS | KNN | NNG | NEC | GMN | GEN | fDBD | Maha | Maha++ | rMaha | rMaha++ |
|---|---|---|---|---|---|---|---|---|---|---|---|---|---|---|---|---|---|
| ConvNeXt-B-In21k | 86.3 | 33.6 | 36.1 | 30.6 | 31.1 | 13.3 | 86.6 | 20.0 | 17.9 | 24.1 | 54.5 | 23.2 | 24.3 | 16.3 | **12.8** | 18.2 | 19.4 |
| ConvNeXt-B | 84.4 | 55.7 | 92.2 | 86.9 | 67.0 | 39.9 | 99.8 | 46.2 | 37.2 | 62.7 | 74.5 | 51.4 | 49.8 | 42.3 | **34.9** | 38.5 | 37.0 |
| ConvNeXtV2-T-In21k | 85.1 | 34.7 | 28.0 | 27.5 | 28.8 | **14.7** | 97.3 | 27.5 | 23.6 | 23.1 | 41.2 | 24.6 | 29.4 | 19.1 | 17.7 | 23.0 | 23.1 |
| ConvNeXtV2-B-In21k | 87.6 | 26.9 | 19.4 | 18.7 | 20.7 | 12.4 | 95.2 | 18.6 | 15.9 | 16.7 | 34.2 | 17.7 | 19.1 | 13.7 | **11.9** | 15.7 | 15.2 |
| ConvNeXtV2-L-In21k | 88.2 | 26.0 | 19.3 | 18.5 | 20.4 | 15.3 | 95.6 | 18.4 | 15.6 | 16.3 | 18.2 | 17.5 | 19.6 | 15.3 | **10.5** | 15.5 | 14.6 |
| ConvNeXtV2-T | 83.5 | 52.4 | 57.4 | 47.1 | 49.6 | 34.8 | 99.6 | 62.4 | 50.6 | 43.6 | 73.4 | 41.1 | 50.7 | 43.5 | **33.7** | 37.2 | 35.5 |
| ConvNeXtV2-B | 85.5 | 51.7 | 65.2 | 56.0 | 52.2 | 32.0 | 99.9 | 39.1 | 32.9 | 47.7 | 71.3 | 37.1 | 41.1 | 32.5 | **25.8** | 30.6 | 29.0 |
| ConvNeXtV2-L | 86.1 | 51.2 | 61.7 | 51.0 | 50.6 | 33.6 | 99.6 | 34.6 | 30.0 | 47.2 | 54.5 | 34.4 | 35.3 | 27.9 | **24.6** | 27.4 | 28.6 |
| DeiT3-S16-In21k | 84.8 | 52.1 | 44.9 | 40.8 | 45.7 | 33.6 | 99.5 | 35.8 | 34.4 | 42.4 | 43.6 | 36.0 | 42.2 | 36.5 | **30.5** | 35.9 | 32.2 |
| DeiT3-B16-In21k | 86.7 | 48.5 | 50.7 | 38.2 | 45.8 | 24.9 | 99.5 | 24.8 | 24.3 | 38.6 | 37.7 | 29.5 | 29.1 | 24.6 | **20.2** | 24.1 | 21.2 |
| DeiT3-L16-In21k | 87.7 | 47.2 | 39.7 | 29.6 | 40.0 | 22.4 | 98.7 | 22.5 | 20.7 | 29.3 | 26.1 | 25.6 | 25.8 | 21.9 | **18.6** | 21.3 | 19.0 |
| DeiT3-S16 | 83.4 | 47.5 | 42.5 | 49.3 | 41.3 | **29.6** | 84.9 | 62.3 | 35.1 | 41.3 | 59.9 | 33.1 | 41.8 | 40.9 | 35.8 | 38.3 | 34.8 |
| DeiT3-B16 | 85.1 | 51.6 | 77.4 | 87.2 | 56.1 | **29.6** | 99.6 | 57.2 | 63.1 | 55.1 | 57.8 | 33.1 | 41.6 | 37.9 | 34.7 | 35.6 | 33.8 |
| DeiT3-L16 | 85.8 | 52.2 | 78.6 | 91.0 | 57.1 | 32.8 | 73.9 | 39.8 | 67.6 | 57.5 | 49.5 | 31.4 | 39.1 | 31.3 | **25.7** | 29.9 | 26.3 |
| EVA02-B14-In21k | 88.7 | 23.9 | 20.0 | 19.3 | 21.0 | 12.4 | 83.9 | 18.0 | 15.8 | 17.3 | 29.8 | 16.2 | 18.8 | 14.6 | **11.8** | 15.6 | 14.2 |
| EVA02-L14-M38m-In21k | 90.1 | 19.5 | 16.5 | 16.2 | 17.5 | 11.2 | 88.2 | 16.1 | 14.2 | 15.3 | 36.4 | 13.2 | 15.6 | 11.7 | **10.3** | 13.0 | 12.4 |
| EVA02-T14 | 80.6 | 56.2 | 59.0 | 59.7 | 54.2 | **32.5** | 98.8 | 47.6 | 43.7 | 43.4 | 45.2 | 45.5 | 57.9 | 36.2 | 32.9 | 39.3 | 37.1 |
| EVA02-S14 | 85.7 | 41.9 | 44.4 | 43.8 | 39.2 | **19.2** | 99.6 | 28.3 | 25.2 | 31.0 | 34.2 | 28.5 | 36.1 | 22.1 | 21.0 | 24.4 | 23.1 |
| EffNetV2-S | 83.9 | 50.8 | 66.1 | 46.6 | 53.3 | 33.8 | 99.8 | 29.2 | 29.4 | 50.3 | 76.6 | 36.7 | 40.1 | 28.8 | **23.9** | 27.6 | 27.2 |
| EffNetV2-L | 85.7 | 49.1 | 69.4 | 45.3 | 51.3 | 31.7 | 99.6 | 34.8 | 32.9 | 47.1 | 49.8 | 32.2 | 35.4 | 25.0 | **20.0** | 23.7 | 22.2 |
| EffNetV2-M | 85.2 | 49.1 | 63.2 | 44.4 | 49.3 | 39.4 | 99.8 | 38.0 | 34.7 | 45.7 | 64.6 | 32.7 | 39.6 | 30.3 | **23.1** | 27.5 | 24.8 |
| Mixer-B16-In21k | 76.6 | 64.2 | 79.8 | 80.7 | 68.5 | 64.1 | 96.4 | 70.1 | 81.2 | 69.3 | 55.8 | 57.8 | 62.6 | 53.3 | **39.7** | 50.0 | 42.3 |
| SwinV2-B-In21k | 87.1 | 35.1 | 30.4 | 23.5 | 28.4 | 17.5 | 72.0 | 20.2 | 16.3 | 21.1 | 55.5 | 21.7 | 22.3 | 22.7 | **13.3** | 20.8 | 18.8 |
| SwinV2-L-In21k | 87.5 | 31.3 | 29.7 | 22.4 | 26.6 | 20.7 | 83.7 | 19.6 | 16.6 | 20.7 | 32.9 | 20.8 | 21.1 | 23.5 | **11.9** | 20.0 | 17.1 |
| SwinV2-S | 84.2 | 53.1 | 61.8 | 52.2 | 52.9 | 34.9 | 99.9 | 44.5 | 37.6 | 46.5 | 58.3 | 39.9 | 48.1 | 37.1 | **25.3** | 33.8 | 26.8 |
| SwinV2-B | 84.6 | 54.6 | 58.5 | 46.9 | 52.1 | 34.1 | 99.6 | 41.0 | 36.2 | 45.2 | 49.3 | 36.8 | 43.9 | 33.5 | **27.1** | 31.6 | 27.6 |
| ResNet101 | 81.9 | 60.5 | 79.9 | 99.5 | 64.3 | 29.4 | 72.7 | 34.2 | 35.6 | 63.6 | 78.9 | 51.6 | 60.4 | 27.7 | **27.7** | 50.8 | 71.4 |
| ResNet152 | 82.3 | 59.3 | 79.5 | 99.4 | 63.8 | 28.5 | 72.8 | 32.8 | 29.9 | 62.0 | 72.3 | 49.3 | 58.0 | 26.5 | 26.9 | 45.9 | 68.7 |
| ResNet50 | 80.9 | 66.2 | 96.5 | 99.2 | 71.3 | 32.4 | 72.8 | 54.2 | 52.2 | 72.0 | 88.6 | 55.2 | 65.7 | 32.2 | **32.2** | 61.2 | 75.7 |
| ResNet50-supcon | 78.7 | 43.0 | 33.9 | 25.0 | 35.5 | 61.7 | **20.0** | 26.8 | 23.5 | 34.4 | 74.4 | 41.7 | 30.7 | 93.7 | 26.6 | 89.5 | 60.5 |
| ViT-T16-In21k-augreg | 75.5 | 62.1 | 39.6 | **30.6** | 44.0 | 36.1 | 96.8 | 69.1 | 60.7 | 36.4 | 48.3 | 52.1 | 45.2 | 46.0 | 32.2 | 49.1 | 46.1 |
| ViT-S16-In21k-augreg | 81.4 | 45.8 | 24.2 | 28.5 | 27.4 | 18.0 | 68.8 | 41.2 | 33.4 | 23.1 | 32.2 | 31.4 | 29.0 | 23.4 | **15.5** | 30.9 | 27.1 |
| ViT-B16-In21k-augreg2 | 85.1 | 46.0 | 37.1 | 30.2 | 37.6 | 38.0 | 99.5 | 30.3 | **25.4** | 33.1 | 54.5 | 29.2 | 36.8 | 38.7 | 25.8 | 33.0 | 27.8 |
| ViT-B16-In21k-augreg | 84.5 | 34.5 | 19.9 | 21.0 | 21.2 | **13.0** | 94.6 | 40.6 | 30.7 | 18.8 | 27.0 | 22.3 | 21.6 | 13.0 | 14.5 | 19.1 | 19.3 |
| ViT-B16-In21k-orig | 81.8 | 33.9 | 21.3 | 21.5 | 23.2 | 20.7 | 62.7 | 24.2 | 22.6 | 20.9 | 40.5 | 25.9 | 24.4 | 21.2 | **18.8** | 25.8 | 24.6 |
| ViT-B16-In21k-miil | 84.3 | 36.3 | 23.3 | 22.2 | 26.8 | 26.2 | 97.4 | 28.9 | 23.2 | 21.7 | 53.5 | 23.7 | 29.5 | 34.4 | **19.0** | 32.6 | 26.3 |
| ViT-L16-In21k-augreg | 85.8 | 29.3 | 16.0 | 16.1 | 17.4 | 11.0 | 93.4 | 35.6 | 24.7 | 15.7 | 33.1 | 18.7 | 16.9 | 10.7 | 11.8 | 15.8 | 15.8 |
| ViT-L16-In21k-orig | 81.5 | 32.7 | 22.1 | 22.0 | 23.6 | 21.0 | 44.9 | 22.8 | 22.0 | 21.8 | 40.3 | 25.8 | 24.3 | 20.6 | **18.3** | 24.3 | 23.6 |
| ViT-S16-augreg | 78.8 | 55.8 | 45.4 | 47.5 | 47.6 | 56.6 | 97.1 | 61.3 | 58.2 | 47.7 | 52.5 | 50.3 | 53.8 | 35.6 | **35.5** | 36.3 | 36.0 |
| ViT-B16-augreg | 79.2 | 55.3 | 47.2 | 43.3 | 48.6 | 53.0 | 88.2 | 53.8 | 52.5 | 48.2 | 54.3 | 50.5 | 52.7 | 36.4 | **34.6** | 35.8 | 34.7 |
| ViT-B16-CLIP-L2b-In12k | 86.2 | 32.9 | 31.6 | 28.5 | 29.3 | 22.3 | 99.7 | 20.8 | 19.0 | 26.0 | 34.0 | 22.9 | 25.7 | 28.7 | **17.1** | 23.9 | 21.1 |
| ViT-L14-CLIP-L2b-In12k | 88.2 | 23.2 | 18.8 | 18.1 | 19.3 | 14.5 | 98.0 | 17.5 | 15.1 | 19.1 | 29.6 | 16.4 | 17.7 | 16.8 | **13.4** | 17.2 | 16.3 |
| ViT-H14-CLIP-L2b-In12k | 88.6 | 23.7 | 19.6 | 19.0 | 20.3 | 14.8 | 98.3 | 18.3 | 15.3 | 20.1 | 51.3 | 16.3 | 18.7 | 15.6 | **12.8** | 16.9 | 15.9 |
| ViT-so400M-SigLip | 89.4 | 36.8 | 41.1 | 30.7 | 34.0 | 17.7 | 92.4 | 15.4 | **14.3** | 31.9 | 64.6 | 18.7 | 17.3 | 16.8 | 15.1 | 16.3 | 16.4 |
| **Average** | 84.4 | 44.0 | 45.7 | 42.6 | 40.4 | 28.1 | 89.1 | 35.1 | 32.1 | 36.7 | 50.3 | 32.3 | 35.4 | 29.1 | **23.0** | 30.5 | 29.3 |

*Table 10.* AUC on OpenOOD, Green indicates that normalized method is better than its unnormalized counterpart, **bold** indicates the best method, and underlined indicates second best method. Maha++ improves over Maha on average by 2.1% in AUC over all models. Similarly, rMaha++ is 0.6% better in AUC than rMaha. In total, Maha++ improves the SOTA compared to the previously strongest methods rMaha by 1.4%, which is significant. The highest AUC is achieved by Maha++ for the EVA02-L14-M38m-In21k highlighted in blue.

| Model | Val Acc | MSP | E | E+R | ML | ViM | AshS | KNN | NNG | NEC | GMN | GEN | fDBD | Maha | Maha++ | rMaha | rMaha++ |
|---|---|---|---|---|---|---|---|---|---|---|---|---|---|---|---|---|---|
| ConvNeXt-B-In21k | 86.3 | 88.8 | 86.8 | 88.9 | 88.4 | 93.3 | 52.1 | 90.1 | 92.6 | 91.6 | 88.1 | 91.5 | 90.3 | 91.9 | **93.7** | 92.0 | 92.5 |
| ConvNeXt-B | 84.4 | 80.9 | 61.3 | 71.2 | 74.7 | 85.4 | 19.8 | 84.4 | 86.8 | 76.4 | 81.6 | 83.2 | 84.7 | 87.1 | **88.7** | 87.9 | 88.7 |
| ConvNeXtV2-T-In21k | 85.1 | 87.4 | 88.1 | 88.3 | 88.3 | 93.3 | 40.3 | 88.6 | 91.0 | 91.3 | 89.1 | 90.0 | 86.4 | 91.8 | 92.7 | 91.2 | 91.5 |
| ConvNeXtV2-B-In21k | 87.6 | 89.9 | 90.4 | 91.0 | 90.4 | 94.8 | 45.2 | 92.0 | 93.6 | 93.4 | 91.1 | 92.3 | 91.4 | 94.0 | **94.9** | 93.7 | 94.0 |
| ConvNeXtV2-L-In21k | 88.2 | 90.6 | 91.1 | 91.9 | 91.1 | 93.8 | 45.4 | 92.1 | 93.8 | 94.0 | 92.7 | 92.8 | 91.7 | 93.7 | **95.3** | 93.8 | 94.2 |
| ConvNeXtV2-T | 83.5 | 83.5 | 78.1 | 82.2 | 81.7 | 86.7 | 21.6 | 81.2 | 85.3 | 83.5 | 82.2 | 86.6 | 84.9 | 87.0 | 88.9 | 88.2 | **89.1** |
| ConvNeXtV2-B | 85.5 | 82.8 | 73.0 | 78.7 | 79.0 | 86.7 | 15.8 | 86.3 | 88.3 | 80.1 | 83.4 | 86.7 | 86.8 | 89.0 | **90.5** | 89.6 | 90.3 |
| ConvNeXtV2-L | 86.1 | 83.0 | 73.0 | 79.9 | 78.9 | 84.5 | 17.3 | 87.2 | 88.8 | 78.9 | 85.8 | 87.3 | 88.0 | 89.8 | **90.6** | 90.3 | 90.6 |
| DeiT3-S16-In21k | 84.8 | 80.3 | 78.7 | 81.2 | 79.5 | 87.1 | 19.5 | 86.0 | 87.4 | 81.1 | 87.9 | 85.2 | 85.5 | 87.9 | **89.5** | 88.1 | 89.2 |
| DeiT3-B16-In21k | 86.7 | 82.4 | 74.8 | 82.2 | 78.4 | 90.6 | 18.4 | 89.7 | 90.4 | 82.7 | 89.8 | 87.4 | 88.7 | 91.0 | 92.3 | 91.1 | 92.1 |
| DeiT3-L16-In21k | 87.7 | 84.3 | 81.5 | 86.6 | 82.7 | 92.0 | 20.5 | 91.1 | 91.4 | 88.3 | 90.9 | 89.3 | 89.4 | 92.0 | **93.0** | 91.9 | 92.8 |
| DeiT3-S16 | 83.4 | 84.2 | 83.8 | 83.0 | 84.7 | 88.2 | 61.9 | 83.0 | 86.5 | 84.9 | 85.4 | 87.8 | 86.5 | 87.9 | 88.9 | 88.6 | **89.3** |
| DeiT3-B16 | 85.1 | 83.5 | 69.5 | 63.3 | 79.0 | 87.6 | 22.4 | 83.1 | 79.1 | 79.8 | 85.1 | 87.6 | 86.1 | 88.1 | 88.9 | 88.9 | **89.6** |
| DeiT3-L16 | 85.8 | 82.9 | 76.6 | 66.4 | 80.5 | 87.0 | 65.3 | 84.6 | 80.4 | 80.6 | 86.3 | 87.8 | 86.7 | 89.0 | 89.9 | 89.7 | **90.4** |
| EVA02-B14-In21k | 88.7 | 91.0 | 90.4 | 91.3 | 90.9 | 95.0 | 54.0 | 92.2 | 93.6 | 93.4 | 92.5 | 93.3 | 92.6 | 94.0 | 94.9 | 93.6 | 94.1 |
| EVA02-L14-M38m-In21k | 90.1 | 92.4 | 91.7 | 92.2 | 92.3 | 95.9 | 48.0 | 93.6 | 94.6 | 94.4 | 92.0 | 94.3 | 94.1 | 95.5 | **95.9** | 95.1 | 95.3 |
| EVA02-T14 | 80.6 | 81.3 | 79.1 | 78.9 | 80.8 | 87.2 | 43.1 | 81.7 | 85.0 | 85.0 | 86.4 | 85.0 | 80.0 | 86.6 | **87.2** | 86.3 | 86.7 |
| EVA02-S14 | 85.7 | 84.4 | 79.7 | 80.1 | 82.1 | 91.6 | 22.7 | 87.2 | 89.5 | 86.4 | 90.5 | 87.8 | 85.4 | 90.8 | 91.2 | 90.4 | 90.7 |
| EffNetV2-S | 83.9 | 83.2 | 74.2 | 83.2 | 79.3 | 86.7 | 20.8 | 86.8 | 88.0 | 82.0 | 82.1 | 87.2 | 86.6 | 88.6 | 90.0 | 89.7 | **90.4** |
| EffNetV2-L | 85.7 | 83.9 | 73.5 | 83.2 | 80.6 | 85.6 | 17.8 | 86.9 | 87.8 | 82.2 | 86.3 | 87.8 | 86.6 | 89.6 | 90.8 | 90.7 | **91.3** |
| EffNetV2-M | 85.2 | 83.7 | 74.4 | 83.3 | 80.6 | 86.3 | 17.8 | 86.3 | 87.6 | 82.4 | 84.6 | 88.0 | 86.2 | 88.9 | 90.5 | 90.2 | **91.0** |
| Mixer-B16-In21k | 76.6 | 80.4 | 78.9 | 78.7 | 79.8 | 82.5 | 48.6 | 79.3 | 79.3 | 79.9 | 84.0 | 82.2 | 81.3 | 83.1 | 86.4 | 85.1 | 86.4 |
| SwinV2-B-In21k | 87.1 | 88.0 | 87.0 | 90.4 | 88.3 | 92.1 | 47.5 | 88.8 | 91.8 | 91.9 | 87.0 | 91.4 | 90.1 | 90.6 | 92.9 | 90.8 | 91.9 |
| SwinV2-L-In21k | 87.5 | 88.7 | 86.8 | 90.8 | 88.1 | 91.9 | 38.4 | 89.8 | 92.2 | 91.8 | 88.9 | 91.6 | 90.8 | 91.1 | **93.7** | 91.5 | 92.7 |
| SwinV2-S | 84.2 | 82.3 | 74.8 | 81.4 | 78.9 | 87.0 | 14.8 | 84.3 | 86.8 | 81.5 | 85.7 | 86.4 | 85.7 | 87.9 | **90.0** | 88.3 | 90.0 |
| SwinV2-B | 84.6 | 82.4 | 75.9 | 82.8 | 79.4 | 86.1 | 21.2 | 85.7 | 87.2 | 82.0 | 86.7 | 87.2 | 86.1 | 88.8 | **90.2** | 89.0 | 90.1 |
| ResNet101 | 81.9 | 79.9 | 68.2 | 23.1 | 75.8 | 83.7 | 56.4 | 82.8 | 85.5 | 76.7 | 74.9 | 83.0 | 79.2 | 88.0 | **89.5** | 87.0 | 85.2 |
| ResNet152 | 82.3 | 80.4 | 67.6 | 21.0 | 75.8 | 84.2 | 56.0 | 82.7 | 85.9 | 77.6 | 77.7 | 83.6 | 80.6 | 88.5 | **90.0** | 87.7 | 86.0 |
| ResNet50 | 80.9 | 77.5 | 52.6 | 27.6 | 73.3 | 82.0 | 60.8 | 79.7 | 81.5 | 73.3 | 65.1 | 81.9 | 77.4 | 86.5 | **87.5** | 84.6 | 83.1 |
| ResNet50-supcon | 78.7 | 85.2 | 87.7 | 88.7 | 87.6 | 82.3 | 88.6 | 86.4 | 88.6 | 87.8 | 78.9 | 86.7 | 87.1 | 52.9 | **89.2** | 76.5 | 85.6 |
| ViT-T16-In21k-augreg | 75.5 | 79.9 | 85.5 | **86.9** | 85.1 | 86.5 | 45.7 | 75.9 | 80.0 | 86.4 | 84.1 | 83.9 | 85.5 | 84.3 | 86.5 | 83.8 | 84.0 |
| ViT-S16-In21k-augreg | 81.4 | 84.4 | 90.4 | 89.9 | 90.0 | 91.6 | 66.5 | 85.3 | 88.2 | 90.8 | 88.8 | 89.1 | 89.2 | 90.4 | 91.5 | 89.1 | 89.4 |
| ViT-B16-In21k-augreg2 | 85.1 | 84.4 | 84.6 | 87.3 | 85.4 | 85.9 | 24.6 | 86.8 | 89.1 | 87.6 | 86.0 | 88.6 | 84.5 | 86.7 | **90.3** | 88.6 | 90.0 |
| ViT-B16-In21k-augreg | 84.5 | 87.5 | 91.9 | 91.3 | 91.7 | 93.6 | 54.8 | 85.6 | 89.1 | 92.4 | 90.1 | 91.6 | 91.4 | 93.4 | 92.5 | 92.0 | 91.9 |
| ViT-B16-In21k-orig | 81.8 | 88.0 | 93.2 | 93.1 | 92.7 | 93.5 | 77.1 | 90.4 | 91.9 | 93.3 | 87.7 | 91.1 | 92.0 | 92.7 | **93.5** | 91.6 | 91.8 |
| ViT-B16-In21k-miil | 84.3 | 87.7 | 90.6 | 91.0 | 90.1 | 91.8 | 32.1 | 88.4 | 91.1 | 91.7 | 86.6 | 91.0 | 89.1 | 89.6 | 92.6 | 90.2 | 91.2 |
| ViT-L16-In21k-augreg | 85.8 | 89.6 | 93.0 | 94.1 | 92.9 | 94.6 | 58.1 | 87.8 | 90.6 | 93.5 | 89.9 | 93.0 | 93.1 | **94.9** | 94.0 | 93.9 | 93.7 |
| ViT-L16-In21k-orig | 81.5 | 89.5 | 93.3 | 93.3 | 93.0 | 93.4 | 86.2 | 92.0 | 92.9 | 93.5 | 87.3 | 91.9 | 92.5 | 92.8 | **93.8** | 92.2 | 92.4 |
| ViT-S16-augreg | 78.8 | 82.3 | 84.9 | 84.6 | 84.8 | 80.4 | 46.7 | 77.9 | 81.8 | 84.6 | 83.9 | 85.1 | 83.5 | 86.9 | 86.8 | 87.4 | **87.4** |
| ViT-B16-augreg | 79.2 | 82.7 | 85.8 | 86.4 | 85.7 | 84.1 | 52.9 | 81.2 | 84.0 | 85.6 | 84.5 | 85.6 | 84.8 | 87.4 | 87.6 | 88.0 | **88.1** |
| ViT-B16-CLIP-L2b-In12k | 86.2 | 87.9 | 85.8 | 87.8 | 87.1 | 92.3 | 19.5 | 90.4 | 92.5 | 89.6 | 91.2 | 90.8 | 90.6 | 90.5 | 93.3 | 91.2 | 92.2 |
| ViT-L14-CLIP-L2b-In12k | 88.2 | 90.5 | 89.9 | 90.6 | 90.4 | 94.6 | 38.2 | 92.2 | 94.1 | 90.9 | 92.3 | 92.6 | 92.7 | 93.6 | **94.8** | 93.7 | 94.0 |
| ViT-H14-CLIP-L2b-In12k | 88.6 | 90.6 | 89.4 | 90.1 | 90.2 | 94.2 | 27.6 | 92.0 | 93.6 | 90.8 | 88.9 | 92.7 | 92.1 | 93.7 | **94.7** | 93.7 | 94.0 |
| ViT-so400M-SigLip | 89.4 | 87.6 | 79.7 | 85.8 | 83.8 | 92.6 | 25.7 | 91.8 | 93.1 | 85.7 | 88.8 | 91.6 | 92.0 | 93.2 | **93.8** | 93.4 | 93.6 |
| **Average** | 84.4 | 85.0 | 81.5 | 81.0 | 84.4 | 89.1 | 40.4 | 86.6 | 88.5 | 86.2 | 86.2 | 88.4 | 87.5 | 89.1 | **91.2** | 89.8 | 90.4 |

*Table 11.* AUC on NINCO datasets, Green indicates that normalized method is better than its unnormalized counterpart, **bold** indicates the best method, and underlined indicates second best method. Maha++ improves over Maha on average by 2.6% in AUC over all models. Similarly, rMaha++ is 1.0% better in AUC than rMaha. In total, Maha++ improves the SOTA compared to the previously strongest methods rMaha by 1.0%, which is significant. The highest AUC is achieved by Maha++ for the EVA02-L14-M38m-In21k highlighted in blue.

| Model | Val Acc | MSP | E | E+R | ML | ViM | AshS | KNN | NNG | NEC | GMN | GEN | fDBD | Maha | Maha++ | rMaha | rMaha++ |
|---|---|---|---|---|---|---|---|---|---|---|---|---|---|---|---|---|---|
| ConvNeXt-B-In21k | 86.3 | 88.2 | 85.7 | 87.8 | 87.6 | 92.5 | 45.4 | 88.0 | 91.5 | 90.7 | 87.7 | 90.9 | 89.5 | 91.2 | **94.3** | 92.3 | 93.5 |
| ConvNeXt-B | 84.4 | 81.7 | 64.8 | 73.2 | 76.7 | 83.0 | 25.1 | 81.6 | 85.2 | 78.0 | 81.1 | 83.3 | 83.0 | 85.8 | 88.5 | 87.2 | **88.8** |
| ConvNeXtV2-T-In21k | 85.1 | 86.7 | 87.2 | 87.4 | 87.5 | 92.7 | 39.7 | 86.8 | 90.2 | 90.6 | 88.8 | 89.4 | 85.9 | 91.7 | **92.9** | 91.4 | 91.9 |
| ConvNeXtV2-B-In21k | 87.6 | 89.4 | 89.1 | 89.9 | 89.3 | 94.8 | 43.8 | 91.1 | 93.5 | 92.7 | 91.3 | 92.0 | 91.8 | 94.5 | **95.6** | 94.5 | 95.0 |
| ConvNeXtV2-L-In21k | 88.2 | 90.2 | 89.3 | 90.7 | 89.7 | 93.9 | 39.8 | 91.6 | 93.9 | 93.4 | 92.6 | 92.3 | 92.8 | 94.1 | **96.2** | 94.7 | 95.5 |
| ConvNeXtV2-T | 83.5 | 82.9 | 76.2 | 80.3 | 80.5 | 83.5 | 26.0 | 78.0 | 83.1 | 82.0 | 81.0 | 85.6 | 82.5 | 85.4 | 88.3 | 87.5 | **89.0** |
| ConvNeXtV2-B | 85.5 | 82.6 | 73.2 | 78.2 | 79.2 | 83.6 | 20.0 | 83.7 | 86.6 | 79.8 | 82.9 | 86.1 | 85.2 | 87.9 | 90.1 | 89.0 | **90.3** |
| ConvNeXtV2-L | 86.1 | 82.2 | 71.7 | 78.6 | 78.0 | 80.8 | 19.1 | 84.8 | 87.1 | 77.5 | 84.2 | 86.4 | 86.1 | 88.7 | 90.2 | 89.9 | **90.6** |
| DeiT3-S16-In21k | 84.8 | 77.9 | 74.8 | 77.2 | 76.2 | 85.4 | 23.2 | 83.4 | 85.4 | 77.6 | 87.6 | 82.7 | 84.2 | 87.3 | **89.2** | 87.6 | 89.0 |
| DeiT3-B16-In21k | 86.7 | 81.5 | 74.5 | 80.5 | 77.5 | 89.3 | 21.3 | 87.6 | 89.2 | 81.3 | 89.6 | 86.6 | 87.5 | 90.2 | 91.8 | 90.6 | **91.9** |
| DeiT3-L16-In21k | 87.7 | 83.9 | 81.9 | 86.3 | 82.7 | 90.9 | 25.3 | 89.6 | 90.8 | 87.4 | 90.6 | 89.1 | 88.7 | 91.6 | 92.8 | 91.9 | **92.9** |
| DeiT3-S16 | 83.4 | 82.9 | 81.4 | 81.2 | 83.1 | 86.3 | 64.6 | 81.1 | 85.0 | 83.0 | 84.4 | 86.4 | 85.1 | 87.5 | 88.6 | 88.4 | **89.3** |
| DeiT3-B16 | 85.1 | 82.2 | 66.6 | 62.4 | 76.7 | 85.1 | 28.4 | 80.5 | 77.1 | 77.3 | 83.9 | 86.4 | 84.0 | 87.2 | 88.3 | 88.3 | **89.2** |
| DeiT3-L16 | 85.8 | 81.2 | 75.5 | 68.8 | 78.6 | 84.7 | 62.3 | 81.1 | 78.8 | 78.7 | 84.8 | 86.0 | 84.9 | 88.2 | 89.1 | 89.2 | **90.1** |
| EVA02-B14-In21k | 88.7 | 90.6 | 89.7 | 91.0 | 90.3 | 94.7 | 50.1 | 91.1 | 93.2 | 92.7 | 92.6 | 93.2 | 93.0 | 94.0 | **95.1** | 93.9 | 94.6 |
| EVA02-L14-M38m-In21k | 90.1 | 92.6 | 91.1 | 91.8 | 92.1 | 96.1 | 40.9 | 93.2 | 94.6 | 94.2 | 92.0 | 94.7 | 94.8 | 96.0 | **96.4** | 95.9 | 96.0 |
| EVA02-T14 | 80.6 | 79.2 | 75.3 | 75.0 | 77.7 | 83.6 | 41.1 | 78.3 | 81.9 | 81.5 | 83.7 | 82.3 | 78.0 | 84.2 | **84.8** | 84.3 | 84.7 |
| EVA02-S14 | 85.7 | 81.8 | 76.1 | 76.5 | 78.8 | 88.6 | 27.0 | 84.3 | 87.1 | 82.8 | 89.6 | 85.0 | 82.8 | 89.2 | **89.6** | 89.3 | 89.6 |
| EffNetV2-S | 83.9 | 81.1 | 71.1 | 77.6 | 76.3 | 81.7 | 20.7 | 83.5 | 84.9 | 78.3 | 80.6 | 84.6 | 83.7 | 85.5 | 87.1 | 88.1 | **89.4** |
| EffNetV2-L | 85.7 | 82.6 | 73.3 | 80.3 | 79.6 | 80.2 | 20.9 | 84.0 | 85.4 | 79.7 | 85.3 | 86.4 | 84.0 | 87.4 | 89.0 | 89.5 | **90.5** |
| EffNetV2-M | 85.2 | 82.3 | 71.8 | 79.5 | 78.6 | 81.0 | 19.3 | 84.0 | 85.4 | 79.4 | 83.9 | 86.6 | 84.1 | 87.2 | 89.1 | 89.5 | **90.6** |
| Mixer-B16-In21k | 76.6 | 79.3 | 78.1 | 78.1 | 78.8 | 80.8 | 49.8 | 76.2 | 78.6 | 78.9 | 82.3 | 80.9 | 79.1 | 80.5 | 84.6 | 83.8 | **85.2** |
| SwinV2-B-In21k | 87.1 | 87.4 | 86.0 | 89.2 | 87.5 | 90.8 | 44.1 | 86.1 | 90.5 | 91.2 | 86.7 | 90.8 | 88.6 | 89.4 | **92.9** | 90.4 | 92.5 |
| SwinV2-L-In21k | 87.5 | 88.2 | 85.9 | 89.9 | 87.4 | 90.8 | 36.9 | 87.8 | 91.5 | 91.4 | 87.7 | 91.2 | 89.8 | 90.0 | **94.1** | 91.2 | 93.5 |
| SwinV2-S | 84.2 | 80.9 | 74.8 | 79.7 | 78.1 | 83.9 | 17.6 | 80.7 | 83.8 | 80.3 | 84.5 | 84.7 | 82.9 | 86.0 | 88.7 | 86.6 | **88.9** |
| SwinV2-B | 84.6 | 80.8 | 74.9 | 80.4 | 78.0 | 81.4 | 28.9 | 82.3 | 84.2 | 79.7 | 84.5 | 85.2 | 83.1 | 86.8 | 88.5 | 87.3 | **88.7** |
| ResNet101 | 81.9 | 78.7 | 67.3 | 21.1 | 74.6 | 76.9 | 47.5 | 77.6 | 82.6 | 74.7 | 74.1 | 80.9 | 73.9 | 85.3 | **89.1** | 87.8 | 88.1 |
| ResNet152 | 82.3 | 79.1 | 67.2 | 20.0 | 74.6 | 77.3 | 48.7 | 77.6 | 83.1 | 75.6 | 76.4 | 81.4 | 75.7 | 85.9 | **89.5** | 87.9 | 88.6 |
| ResNet50 | 80.9 | 76.8 | 55.0 | 23.7 | 73.0 | 73.5 | 52.6 | 75.1 | 78.8 | 72.3 | 66.2 | 79.7 | 71.7 | 82.7 | 86.5 | 86.1 | **86.6** |
| ResNet50-supcon | 78.7 | 84.9 | 87.2 | 87.1 | 87.2 | 79.2 | 86.0 | 84.3 | 86.8 | 87.3 | 80.3 | 86.9 | 85.0 | 48.2 | **88.1** | 78.7 | 87.0 |
| ViT-T16-In21k-augreg | 75.5 | 78.1 | 81.0 | 82.1 | 81.3 | 82.2 | 54.4 | 73.3 | 76.2 | 82.3 | 81.0 | 80.8 | 83.0 | **84.0** | 83.7 | 83.0 | 82.8 |
| ViT-S16-In21k-augreg | 81.4 | 83.4 | 88.6 | 88.4 | 88.5 | 89.7 | 61.1 | 82.0 | 85.6 | 89.2 | 86.7 | 87.7 | 88.4 | 90.7 | **90.8** | 89.5 | 89.4 |
| ViT-B16-In21k-augreg2 | 85.1 | 83.2 | 82.7 | 85.2 | 83.9 |  | 28.7 | 84.7 | 87.6 | 86.1 | 85.3 | 87.8 | 82.8 | 86.1 | 90.7 |  | **90.8** |
| ViT-B16-In21k-augreg | 84.5 | 86.3 | 91.1 | 90.3 | 91.0 | 92.4 | 56.2 | 82.8 | 87.5 | 91.5 | 89.3 | 91.1 | 90.3 | **94.1** | 93.2 | 93.2 | 93.0 |
| ViT-B16-In21k-orig | 81.8 | 87.2 | 91.8 | 91.8 | 91.5 | 92.6 | 71.0 | 88.3 | 90.1 | 92.1 | 87.3 | 90.3 | 91.2 | 93.1 | **93.8** | 92.7 | 92.9 |
| ViT-B16-In21k-miil | 84.3 | 86.5 | 88.1 | 88.6 | 88.0 | 91.2 | 31.3 | 87.2 | 89.7 | 90.0 | 86.5 | 89.8 | 88.2 | 89.9 | **93.1** | 91.1 | 92.2 |
| ViT-L16-In21k-augreg | 85.8 | 89.5 | 92.7 | 94.9 | 92.7 | 93.9 | 48.1 | 83.9 | 88.8 | 93.2 | 88.6 | 93.2 | 92.4 | **95.3** | 94.4 | 95.1 | 94.8 |
| ViT-L16-In21k-orig | 81.5 | 89.3 | 91.7 | 91.8 | 91.6 | 91.9 | 81.7 | 89.7 | 91.3 | 92.1 | 86.1 | 91.1 | 91.6 | 92.4 | **93.5** | 92.9 | 93.1 |
| ViT-S16-augreg | 78.8 | 80.8 | 81.8 | 81.5 | 82.2 | 74.8 | 44.4 | 73.8 | 78.2 | 81.9 | 81.4 | 83.0 | 81.0 | 85.3 | 85.1 | **86.6** | 86.6 |
| ViT-B16-augreg | 79.2 | 81.0 | 83.3 | 83.8 | 83.4 | 81.0 | 52.4 | 77.8 | 81.1 | 83.3 | 82.0 | 83.9 | 82.5 | 85.7 | 85.7 | **86.9** | **86.9** |
| ViT-B16-CLIP-L2b-In12k | 86.2 | 86.6 | 82.9 | 85.5 | 84.9 | 91.1 | 21.5 | 88.4 | 91.1 | 86.9 | 90.6 | 89.6 | 90.1 | 89.4 | **92.9** | 90.6 | 92.3 |
| ViT-L14-CLIP-L2b-In12k | 88.2 | 90.0 | 87.9 | 89.0 | 89.0 | 94.4 | 33.4 | 91.2 | 93.3 | 89.6 | 92.6 | 92.2 | 93.4 | 93.9 | **95.2** | 94.2 | 94.8 |
| ViT-H14-CLIP-L2b-In12k | 88.6 | 89.7 | 87.2 | 88.2 | 88.5 | 93.9 | 27.6 | 91.0 | 92.4 | 89.2 | 89.8 | 91.8 | 92.7 | 94.2 | **95.3** | 94.3 | 94.8 |
| ViT-so400M-SigLip | 89.4 | 87.3 | 79.5 | 85.1 | 83.2 | 92.1 | 26.1 | 91.8 | 93.3 | 84.7 | 88.3 | 91.7 | 92.4 | 93.3 | 94.6 | 94.2 | **94.8** |
| **Average** | 84.4 | 84.1 | 80.2 | 79.3 | 83.1 | 86.6 | 39.9 | 84.1 | 86.7 | 84.6 | 85.4 | 87.3 | 85.9 | 88.1 | **90.7** | 89.7 | 90.7 |

*Table 12.* FPR on NINCO for cosine-based methods, Green indicates that the normalized method is better than its unnormalized counterpart, **bold** indicates the best method, and underlined indicates the second best method. *Mahalanobis++* consistently outperforms other cosine-based methods. In only 2 out of 44 models, another method (once NNguide and once Cosine) is better than Maha++.

| Model | Accuracy | Maha++ | Cosine | KNN (Sun et al., 2022) | NNguide (Park et al., 2023a) | SSC (Techapanurak et al., 2020) |
|---|---|---|---|---|---|---|
| ConvNeXt-B | 84.434 | **50.5** | 60.6 | 70.1 | 62.2 | 69.3 |
| ConvNeXt-B-In21k | 86.270 | **28.8** | 42.2 | 51.6 | 41.2 | 51.3 |
| ConvNeXtV2-B | 85.474 | **44.7** | 57.1 | 67.3 | 60.3 | 66.6 |
| ConvNeXtV2-B-In21k | 87.642 | **22.4** | 31.1 | 40.9 | 31.7 | 40.4 |
| ConvNeXtV2-L | 86.120 | **43.0** | 53.8 | 62.4 | 56.9 | 59.7 |
| ConvNeXtV2-L-In21k | 88.196 | **18.4** | 27.0 | 38.7 | 29.9 | 39.2 |
| ConvNeXtV2-T | 83.462 | **52.3** | 69.4 | 82.3 | 73.9 | 72.8 |
| ConvNeXtV2-T-In21k | 85.104 | **32.8** | 45.8 | 54.1 | 45.0 | 55.1 |
| DeiT3-B16 | 85.074 | **57.2** | 67.1 | 74.5 | 80.1 | 65.7 |
| DeiT3-B16-In21k | 86.744 | **38.8** | 46.9 | 52.6 | 46.4 | 53.2 |
| DeiT3-L16 | 85.812 | **50.4** | 62.2 | 67.2 | 77.9 | 68.7 |
| DeiT3-L16-In21k | 87.722 | **33.9** | 38.9 | 43.8 | 37.8 | 46.3 |
| DeiT3-S16 | 83.434 | **53.5** | 67.6 | 75.6 | 57.8 | 63.6 |
| DeiT3-S16-In21k | 84.826 | **50.8** | 58.4 | 62.9 | 59.3 | 65.2 |
| EVA02-B14-In21k | 88.694 | **23.8** | 29.1 | 37.6 | 30.0 | 34.4 |
| EVA02-L14-M38m-In21k | 90.054 | **18.6** | 22.7 | 30.3 | 26.1 | 28.8 |
| EVA02-S14 | 85.720 | **48.0** | 53.6 | 60.0 | 54.0 | 63.8 |
| EVA02-T14 | 80.630 | **64.0** | 69.8 | 74.5 | 71.0 | 75.5 |
| Mixer-B16-In21k | 76.598 | **65.4** | 78.2 | 85.8 | 83.7 | 79.5 |
| ResNet101 | 81.890 | **50.4** | 61.5 | 74.9 | 66.4 | 87.2 |
| ResNet152 | 82.286 | **46.5** | 62.1 | 72.0 | 61.6 | 85.8 |
| ResNet50 | 80.856 | **61.0** | 64.0 | 83.7 | 75.0 | 88.2 |
| ResNet50-supcon | 78.686 | 59.6 | 58.9 | 65.8 | **58.4** | 74.1 |
| SwinV2-B-In21k | 87.096 | **31.3** | 45.9 | 57.2 | 42.7 | 53.7 |
| SwinV2-B | 84.604 | **52.2** | 63.4 | 69.4 | 65.2 | 71.6 |
| SwinV2-L-In21k | 87.468 | **28.3** | 42.8 | 55.1 | 41.7 | 53.6 |
| SwinV2-S | 84.220 | **49.8** | 65.2 | 73.1 | 66.8 | 75.2 |
| EffNetV2-L | 85.664 | **47.8** | 57.0 | 62.5 | 60.1 | 62.4 |
| EffNetV2-M | 85.204 | **50.0** | 58.0 | 63.1 | 60.6 | 64.4 |
| EffNetV2-S | 83.896 | 59.9 | **58.8** | 60.9 | 59.6 | 68.7 |
| ViT-B16-In21k-augreg2 | 85.096 | **45.9** | 56.4 | 64.0 | 57.0 | 64.8 |
| ViT-B16-augreg | 79.152 | **61.3** | 73.1 | 77.6 | 75.9 | 75.7 |
| ViT-B16-In21k-augreg | 84.528 | **35.7** | 53.4 | 67.7 | 59.0 | 54.0 |
| ViT-B16-In21k-orig | 81.790 | **31.6** | 46.0 | 52.7 | 47.6 | 45.2 |
| ViT-B16-In21k-miil | 84.268 | **35.4** | 49.7 | 59.6 | 51.7 | 62.1 |
| ViT-B16-CLIP-L2b-In12k | 86.172 | **35.8** | 43.5 | 49.4 | 42.3 | 48.8 |
| ViT-H14-CLIP-L2b-In12k | 88.588 | **23.7** | 31.5 | 41.7 | 27.4 | 36.5 |
| ViT-L14-CLIP-L2b-In12k | 88.178 | **25.4** | 30.9 | 39.5 | 25.4 | 33.3 |
| ViT-L16-In21k-augreg | 85.840 | **28.9** | 50.0 | 68.6 | 58.9 | 48.2 |
| ViT-L16-In21k-orig | 81.508 | **32.4** | 39.2 | 45.8 | 40.7 | 42.0 |
| ViT-S16-augreg | 78.842 | **63.1** | 75.8 | 82.1 | 80.0 | 78.1 |
| ViT-S16-In21k-augreg | 81.388 | **44.6** | 60.3 | 70.9 | 64.0 | 61.5 |
| ViT-so400M-SigLip | 89.406 | **27.4** | 29.0 | 36.3 | 30.0 | 35.6 |
| ViT-T16-In21k-augreg | 75.466 | **63.2** | 77.2 | 81.7 | 81.9 | 74.2 |

*Table 13.* AUROC for CIFAR10, Green indicates that the normalized method is better than its unnormalized counterpart, **bold** indicates the best method, and underlined indicates the second best method. Maha++ is clearly the best method. Only for the WRN28-10 Maha is better (but not significantly). Maha++ improves in all cases over the previously beset methods ViM. We highlight the best AUC achieved by Maha++ for the ViT-B16-21k-1k in blue.

| Model | Ash | Dice | Ebo | KlM | KNN | ML | MSP | O-Max | React | She | NNguide | T-Scal | ViM | Neco | rMD | rMD++ | MD | MD++ |
|---|---|---|---|---|---|---|---|---|---|---|---|---|---|---|---|---|---|---|
| SwinV2-S-1k | 69.96 | 92.85 | 95.61 | 98.04 | 99.25 | 95.83 | 96.60 | 97.02 | 96.83 | 96.88 | 67.51 | 96.61 | 99.53 | 98.86 | 98.83 | 98.79 | 99.50 | **99.57** |
| ViT-B16-21k-1k | 82.75 | 99.33 | 99.42 | 96.98 | 99.64 | 99.41 | 98.88 | 97.66 | 99.45 | 98.99 | 87.30 | 99.06 | 99.67 | 99.56 | 99.03 | 99.04 | 99.60 | **99.71** |
| RN18 | 87.15 | 89.60 | 91.09 | 79.62 | 91.58 | 90.97 | 89.93 | 89.04 | 90.78 | 87.62 | 63.57 | 90.32 | 91.12 | 90.67 | 89.92 | 90.06 | 86.87 | **91.69** |
| RN34 | 78.29 | 84.84 | 87.26 | 82.75 | 92.15 | 87.20 | 88.11 | 87.38 | 87.50 | 81.40 | 55.07 | 88.07 | 92.50 | 86.39 | 90.34 | 90.49 | 91.53 | **93.61** |
| RNxt29-32 | 78.33 | 71.90 | 88.45 | 83.19 | 90.46 | 88.20 | 87.98 | 85.65 | 85.27 | 87.90 | 29.57 | 87.97 | 91.36 | 89.62 | 89.84 | 88.69 | 90.70 | **91.56** |
| Average | 79.29 | 87.70 | 92.37 | 88.11 | 94.62 | 92.32 | 92.30 | 91.35 | 91.97 | 90.56 | 60.60 | 92.41 | 94.84 | 93.02 | 93.59 | 93.41 | 93.64 | **95.23** |
| RN50-SC | — | — | — | — | 96.76 | — | — | — | — | — | — | — | — | — | 94.46 | 94.30 | 59.00 | **96.80** |
| RN34-SC | — | — | — | — | 96.15 | — | — | — | — | — | — | — | — | — | 94.72 | 94.24 | 64.21 | **96.77** |

*Table 14.* FPR for CIFAR10, `Green` indicates that the normalized method is better than its unnormalized counterpart, **bold** indicates the best method, and underlined indicates the second best method. Maha++ is the best method on average. We highlight the best FPR achieved by Maha++ for the ViT-B16-21k-1k in blue.

| Model | Ash | Dice | Ebo | KlM | KNN | ML | MSP | O-Max | React | She | NNguide | T-Scal | ViM | Neco | rMD | rMD++ | MD | MD++ |
|---|---|---|---|---|---|---|---|---|---|---|---|---|---|---|---|---|---|---|
| SwinV2-S-1k | 93.43 | 19.51 | 8.82 | 6.02 | 4.03 | 8.07 | 6.74 | 5.97 | 7.21 | 12.08 | 63.18 | 6.73 | 2.17 | 3.66 | 3.42 | 3.18 | 2.35 | **2.16** |
| ViT-B16-21k-1k | 60.41 | 2.42 | 1.93 | 6.23 | 1.75 | 1.97 | 3.27 | 3.15 | 1.99 | 4.48 | 41.88 | 2.91 | 1.29 | 1.57 | 2.59 | 2.66 | 1.66 | **1.24** |
| RN18 | 47.52 | 41.18 | **39.22** | 56.48 | 45.55 | 40.28 | 56.42 | 76.47 | 40.22 | 45.89 | 77.65 | 52.58 | 51.99 | 41.10 | 52.51 | 54.09 | 69.46 | 46.15 |
| RN34 | 46.12 | 44.20 | **38.05** | 52.89 | 46.24 | 39.14 | 51.99 | 78.47 | 42.36 | 45.81 | 76.79 | 48.98 | 48.15 | 41.93 | 52.67 | 53.67 | 54.45 | 38.36 |
| RNxt29-32 | 97.43 | 63.57 | 47.36 | 56.54 | 55.91 | 50.41 | 53.15 | 89.85 | 56.26 | 38.13 | 99.97 | 53.36 | 36.13 | 51.32 | 58.07 | 61.31 | 41.17 | **34.64** |
| Average | 68.98 | 34.18 | 27.08 | 35.63 | 30.70 | 27.97 | 34.31 | 50.78 | 29.61 | 29.28 | 71.89 | 32.91 | 27.95 | 27.92 | 33.85 | 34.98 | 33.82 | **24.51** |
| RN50-SC | — | — | — | — | 19.48 | — | — | — | — | — | — | — | — | — | 33.23 | 35.26 | 81.77 | **18.59** |
| RN34-SC | — | — | — | — | 22.47 | — | — | — | — | — | — | — | — | — | 30.52 | 32.89 | 78.65 | **17.55** |

*Table 15.* AUROC for CIFAR100, `Green` indicates that the normalized method is better than its unnormalized counterpart, **bold** indicates the best method, and underlined indicates the second best method. Maha++ is clearly the best method. Only for the RNxt29-32 She is slightly better. Maha++ improves in all cases over the previously best methods ViM, Maha and KNN. We highlight the best AUC achieved by Maha++ for the ViT-B32-21k in blue.

| Model | Ash | Dice | Ebo | KlM | KNN | ML | MSP | O-Max | React | She | NNguide | T-Scal | ViM | Neco | rMD | rMD++ | MD | MD++ |
|---|---|---|---|---|---|---|---|---|---|---|---|---|---|---|---|---|---|---|
| SwinV2-S-1k | 48.67 | 63.60 | 84.72 | 82.52 | 90.06 | 85.20 | 85.68 | 85.82 | 87.53 | 89.66 | 71.36 | 85.93 | 91.34 | 90.38 | 89.77 | 89.30 | 90.29 | **92.99** |
| Deit3-S-21k | 49.99 | 44.78 | 85.69 | 81.59 | 88.06 | 86.18 | 86.21 | 84.52 | 88.88 | 87.47 | 55.19 | 86.43 | 90.41 | 89.86 | 87.44 | 87.85 | 88.30 | **90.54** |
| ConvN-T-21k | 63.80 | 53.50 | 77.76 | 80.48 | 86.60 | 78.51 | 79.09 | 82.60 | 80.17 | 82.94 | 62.98 | 79.22 | 87.67 | 81.31 | 85.00 | 84.89 | 87.95 | **89.55** |
| ViT-B32-21k | 59.23 | 88.31 | 90.28 | 89.13 | 94.87 | 89.99 | 85.36 | 88.00 | 88.59 | 94.10 | 87.17 | 86.73 | 94.62 | 90.70 | 92.37 | 92.82 | 95.59 | **96.84** |
| ViT-S16-21k | 65.78 | 84.35 | 89.85 | 84.23 | 93.97 | 89.44 | 83.87 | 88.09 | 88.45 | 92.32 | 80.08 | 85.38 | 95.91 | 90.80 | 93.09 | 93.32 | 95.63 | **96.81** |
| RN18 | 74.20 | 79.77 | 80.31 | 74.11 | 81.22 | 80.31 | 79.70 | 68.22 | 80.27 | 79.18 | 81.06 | 80.02 | 78.50 | 80.66 | 81.27 | 80.91 | 78.46 | **81.71** |
| RN34 | 65.82 | 78.86 | 79.88 | 75.63 | 81.51 | 79.76 | 79.13 | 73.14 | 80.48 | 77.15 | 74.13 | 79.56 | 82.13 | 80.61 | 81.22 | 80.94 | 82.03 | **82.16** |
| RNxt29-32 | 79.46 | 82.01 | 78.58 | 70.79 | 80.89 | 78.47 | 78.37 | 66.11 | 78.36 | **82.59** | 73.21 | 78.22 | 75.33 | 79.68 | 76.87 | 77.06 | 76.18 | 82.48 |
| Average | 63.37 | 71.90 | 83.38 | 79.81 | 87.15 | 83.48 | 82.18 | 79.56 | 84.09 | 85.68 | 73.15 | 82.69 | 86.99 | 85.50 | 85.88 | 85.89 | 86.80 | **89.14** |
| RN34-SC | — | — | — | — | 83.76 | — | — | — | — | — | — | — | — | — | 76.80 | 80.03 | 53.30 | **84.83** |
| RN50-SC | — | — | — | — | 82.41 | — | — | — | — | — | — | — | — | — | 77.90 | 79.67 | 59.01 | **82.44** |

*Table 16.* FPR for CIFAR100, `Green` indicates that the normalized method is better than its unnormalized counterpart, **bold** indicates the best method, and underlined indicates the second best method. Maha++ is improving in all cases over Maha and is on average the best method. We highlight the best FPR achieved by Maha++ for the ViT-S16-21k in blue.

| Model | Ash | Dice | Ebo | KlM | KNN | ML | MSP | O-Max | React | She | NNguide | T-Scal | ViM | Neco | rMD | rMD++ | MD | MD++ |
|---|---|---|---|---|---|---|---|---|---|---|---|---|---|---|---|---|---|---|
| SwinV2-S-1k | 92.66 | 75.98 | 40.95 | 49.65 | 36.27 | 40.96 | 47.28 | 67.04 | 39.54 | 39.64 | 80.29 | 45.58 | 34.02 | 33.59 | 41.40 | 47.14 | 40.10 | **26.01** |
| Deit3-S-21k | 94.47 | 96.34 | 41.61 | 47.86 | 36.81 | 42.37 | 48.92 | 66.00 | 40.46 | 40.93 | 96.35 | 47.15 | 39.99 | 37.12 | 41.02 | 41.36 | 41.99 | **31.72** |
| ConvN-T-21k | 92.11 | 89.10 | 57.67 | 65.50 | 51.16 | 57.44 | 60.60 | 66.86 | 58.23 | 53.76 | 91.25 | 60.04 | 51.18 | 53.92 | 62.79 | 61.66 | 52.48 | **42.69** |
| ViT-B32-21k | 93.98 | 46.59 | 30.51 | 43.24 | 26.49 | 31.28 | 48.02 | 53.68 | 32.53 | 33.25 | 64.86 | 40.74 | 27.14 | 28.61 | 33.80 | 31.03 | 26.28 | **18.94** |
| ViT-S16-21k | 80.45 | 56.38 | 36.06 | 50.09 | 31.91 | 37.63 | 52.17 | 57.38 | 36.48 | 38.89 | 77.85 | 46.68 | 24.90 | 33.24 | 34.10 | 32.83 | 25.51 | **18.58** |
| RN18 | 78.98 | 80.53 | 80.19 | 78.85 | 76.61 | 79.87 | 80.59 | 97.36 | 80.18 | 80.46 | **68.16** | 80.25 | 79.61 | 79.89 | 76.14 | 77.49 | 79.48 | 72.92 |
| RN34 | 78.27 | 78.31 | 75.19 | 78.08 | 74.44 | 75.33 | 76.93 | 94.07 | 74.51 | 78.76 | 75.07 | 76.20 | 77.17 | **74.25** | 75.82 | 76.22 | 76.63 | 74.51 |
| RNxt29-32 | 72.59 | **67.03** | 82.22 | 87.56 | 73.17 | 82.30 | 82.31 | 96.32 | 81.87 | 69.42 | 81.89 | 82.60 | 76.40 | 80.54 | 86.58 | 84.39 | 77.67 | 67.71 |
| Average | 85.44 | 73.78 | 55.55 | 62.60 | 50.86 | 55.90 | 62.10 | 74.84 | 55.47 | 54.39 | 79.47 | 59.90 | 51.30 | 52.65 | 56.46 | 56.51 | 52.52 | **44.13** |
| RN34-SC | — | — | — | — | 66.87 | — | — | — | — | — | — | — | — | — | 90.02 | 74.37 | 93.76 | **63.51** |
| RN50-SC | — | — | — | — | **66.69** | — | — | — | — | — | — | — | — | — | 83.53 | 78.15 | 82.38 | 67.95 |

*Table 17.* **Normalization improves robustness against noise distributions.** We report the number of failed unit tests (noise distributions with FPR values $\geq 10\%$) from (Bitterwolf et al., 2023). Normalization improves the brittleness of Mahalanobis-based detectors.

| model | Maha | Maha++ |
|---|---|---|
| ConvNeXt-B | 16 | **15** |
| ConvNeXt-B-In21k | 4 | **0** |
| ConvNeXtV2-B | 14 | **6** |
| ConvNeXtV2-B-In21k | 5 | **0** |
| ConvNeXtV2-L | 13 | **4** |
| ConvNeXtV2-L-In21k | 2 | **0** |
| ConvNeXtV2-T | 17 | **9** |
| ConvNeXtV2-T-In21k | 6 | **0** |
| DeiT3-B16 | **14** | 15 |
| DeiT3-B16-In21k | 6 | **3** |
| DeiT3-L16 | **8** | **8** |
| DeiT3-L16-In21k | 1 | **0** |
| DeiT3-S16 | 15 | **10** |
| DeiT3-S16-In21k | 17 | **11** |
| EVA02-B14-In21k | 3 | **0** |
| EVA02-L14-M38m-In21k | **0** | **0** |
| EVA02-S14 | 8 | **0** |
| EVA02-T14 | 11 | **0** |
| Mixer-B16-In21k | 17 | **10** |
| ResNet101 | **0** | 1 |
| ResNet152 | **0** | **0** |
| ResNet50 | **0** | 1 |
| ResNet50-supcon | 17 | **0** |
| SwinV2-B-In21k | 10 | **0** |
| SwinV2-B | 12 | 6 |
| SwinV2-S | 15 | 4 |
| EffNetV2-L | 13 | **7** |
| EffNetV2-M | 13 | **4** |
| EffNetV2-S | 11 | **3** |
| ViT-B16-224-In21k-augreg2 | 16 | **7** |
| ViT-B16-224-augreg | 11 | **4** |
| ViT-B16-224-In21k-orig | 2 | **0** |
| ViT-B16-224-In21k-miil | 17 | **0** |
| ViT-B16-CLIP-L2b-In12k | 14 | **0** |
| ViT-H14-CLIP-L2b-In12k | 4 | **0** |
| ViT-L14-CLIP-L2b-In12k | 7 | **0** |
| ViT-L16-224-In21k-orig | 5 | **0** |
| ViT-S16-224-augreg | 2 | **1** |
| ViT-so400M-SigLip | 8 | **0** |

*Table 18.* **Comparison to SSD+.** SSD+ consists of a) training with contrastive loss (implicitly normalizing the features), b) estimating cluster means in the normalized feature space via k-means, c) centering the train features with the closest class mean and estimating a shared covariance matrix, and d) using the Mahalanobis distance at inference time for OOD detection. SSD+ is therefore not readily applicable as post-hoc OOD detection method. To highlight the benefits of post-hoc methods, we report the performance of SSD+ with a ResNet-50, which was trained for 700 epochs with supervised contrastive loss, and compare it to a ConvNext model and an EVA model with varied pretraining schemes. The latter models outperform SSD+ clearly, underlining the importance of post-hoc methods for OOD detection.

| Model | FPR (%) |
|---|---|
| SSD+ w. 100 clusters | 66.0 |
| SSD+ w. 500 clusters | 65.7 |
| SSD+ w. 1000 clusters | 67.8 |
| CnvNxtV2-L + Maha++ | 18.4 |
| EVA02-L14 + Maha++ | 18.6 |

# F. Models

Table 19. Imagenet model checkpoints.

| Modelname | Checkpoint | source |
|---|---|---|
| ViT-B16-In21k-augreg | vit_base_patch16_224.augreg_in21k_ft_in1k | timm / huggingface |
| ViT-L16-In21k-augreg | vit_large_patch16_224.augreg_in21k_ft_in1k | timm / huggingface |
| ViT-T16-In21k-augreg | vit_tiny_patch16_224.augreg_in21k_ft_in1k | timm / huggingface |
| ViT-S16-In21k-augreg | vit_small_patch16_224.augreg_in21k_ft_in1k | timm / huggingface |
| ViT-B16-augreg | vit_base_patch16_224.augreg_in1k | timm / huggingface |
| ViT-S16-augreg | vit_small_patch16_224.augreg_in1k | timm / huggingface |
| ViT-so400M-SigLip | vit_so400m_patch14_siglip_378.webli_ft_in1k | timm / huggingface |
| ViT-H14-CLIP-L2b-In12k | vit_huge_patch14_clip_336.laion2b_ft_in12k_in1k | timm / huggingface |
| ViT-L14-CLIP-L2b-In12k | vit_large_patch14_clip_336.laion2b_ft_in12k_in1k | timm / huggingface |
| ViT-B16-In21k-orig | vit_base_patch16_224.orig_in21k_ft_in1k | timm / huggingface |
| ViT-L16-In21k-orig | vit_large_patch32_384.orig_in21k_ft_in1k | timm / huggingface |
| ViT-B16-In21k-miil | vit_base_patch16_224_miil.in21k_ft_in1k | timm / huggingface |
| ViT-B16-In21k-augreg2 | vit_base_patch16_224.augreg2_in21k_ft_in1k | timm / huggingface |
| ViT-B16-CLIP-L2b-In12k | vit_base_patch16_clip_224.laion2b_ft_in12k_in1k | timm / huggingface |
| EVA02-L14-M38m-In21k | eva02_large_patch14_448.mim_m38m_ft_in22k_in1k | timm / huggingface |
| EVA02-B14-In21k | eva02_base_patch14_448.mim_in22k_ft_in22k_in1k | timm / huggingface |
| EVA02-S14 | eva02_small_patch14_336.mim_in22k_ft_in1k | timm / huggingface |
| EVA02-T14 | eva02_tiny_patch14_336.mim_in22k_ft_in1k | timm / huggingface |
| DeiT3-B16 | deit3_base_patch16_224 | timm / huggingface |
| DeiT3-B16-In21k | deit3_base_patch16_224_in21ft1k | timm / huggingface |
| DeiT3-L16-In21k | deit3_large_patch16_384.fb_in22k_ft_in1k | timm / huggingface |
| DeiT3-B16-In21k | deit3_base_patch16_384.fb_in22k_ft_in1k | timm / huggingface |
| DeiT3-L16 | deit3_large_patch16_384.fb_in1k | timm / huggingface |
| DeiT3-B16 | deit3_base_patch16_384.fb_in1k | timm / huggingface |
| DeiT3-S16-In21k | deit3_small_patch16_384.fb_in22k_ft_in1k | timm / huggingface |
| DeiT3-S16 | deit3_small_patch16_384.fb_in1k | timm / huggingface |
| SwinV2-S | swinv2_small_window16_256.ms_in1k | timm / huggingface |
| SwinV2-B | swinv2_base_window16_256.ms_in1k | timm / huggingface |
| SwinV2-L-In21k | swinv2_large_window12to24_192to384.ms_in22k_ft_in1k | timm / huggingface |
| SwinV2-B-In21k | swinv2_base_window12to24_192to384.ms_in22k_ft_in1k | timm / huggingface |
| ResNet50 | resnet50.tv2_in1k | timm / huggingface |
| ResNet101 | resnet101.tv2_in1k | timm / huggingface |
| ResNet152 | resnet152.tv2_in1k | timm / huggingface |
| ResNet50-supcon | rn50supcon | github.com/roomo7time/nnguide/ |
| ConvNeXt-B | convnext_base.fb_in1k | timm / huggingface |
| ConvNeXt-B-In21k | convnext_base.fb_in22k_ft_in1k | timm / huggingface |
| ConvNeXtV2-L-In21k | convnextv2_large.fcmae_ft_in22k_in1k_384 | timm / huggingface |
| ConvNeXtV2-B-In21k | convnextv2_base.fcmae_ft_in22k_in1k_384 | timm / huggingface |
| ConvNeXtV2-T-In21k | convnextv2_tiny.fcmae_ft_in22k_in1k_384 | timm / huggingface |
| ConvNeXtV2-T | convnextv2_tiny.fcmae_ft_in1k | timm / huggingface |
| ConvNeXtV2-B | convnextv2_base.fcmae_ft_in1k | timm / huggingface |
| ConvNeXtV2-L | convnextv2_large.fcmae_ft_in1k | timm / huggingface |
| Mixer-B16-In21k | mixer_b16_224.goog_in21k_ft_in1k | timm / huggingface |
| EffNetV2-M | tf_efficientnetv2_m.in1k | timm / huggingface |
| EffNetV2-S | tf_efficientnetv2_s.in1k | timm / huggingface |
| EffNetV2-L | tf_efficientnetv2_l.in1k | timm / huggingface |

*Table 20.* Cifar model checkpoints.

| | |
|---|---|
| SwinV2-S-1k | ft from timm model |
| Deit3-S-21k | ft from timm model |
| ConvN-T-21k | ft from timm model |
| ViT-B32-21k | https://github.com/google-research/big_vision |
| ViT-S16-21k | https://github.com/google-research/big_vision |
| RN18 | https://huggingface.co/edadaltocg/ |
| RN34 | https://huggingface.co/edadaltocg/ |
| RN34-SC | https://huggingface.co/edadaltocg/ |
| RN50-SC | https://huggingface.co/edadaltocg/ |
| RNxt29-32 | self trained |

# G. Methods

We describe OOD detection methods evaluated in our work. Let a neural network $n_\theta(x) = g(\phi(x))$ decompose into a feature extractor $\phi$ and linear layer $g(\phi_i) = \mathbf{W}^T \phi_i + \mathbf{b}$. For input $x$, $\phi(x)$ denotes the feature embedding, and $g(\phi(x))$ produces logits $\mathbf{o}$, which can be transformed to a probability vector $\mathbf{p}$ via the softmax function.

**MSP** (Hendrycks & Gimpel, 2017): Classifer confidence, i.e. max-softmax-probability

$$s = \max_c(p_c)$$

**Max-Logit** (ML or MLS) (Hendrycks et al., 2022): Max-Logit returns the largest entry of the logit-vector $\mathbf{o}$, i.e.

$$s = \max_c(o_c)$$

**Energy** (E) (Liu et al., 2020): Energy or log-sum-exp of logits:

$$s = \log \sum_c^C \exp(o_c)$$

**KL-Matching** (KLM) (Hendrycks et al., 2022): KL-Matching computes the average probability vector $\mathbf{d}_c$ for each of the $C$ classes. For a test input, the KL-distances of all $\mathbf{d}_c$ vectors to its probability vector $\mathbf{p}$ are computed, and the OOD-score is the negative of the smallest of those distances:

$$s = -\min_c \mathrm{KL}[\mathbf{p}||\mathbf{d}_c]$$

In the original paper by (Hendrycks et al., 2022), the average for $\mathbf{d}_c$ is computed over an additional validation set. Since none of the other methods leverages extra data and we are interested in fair comparison, we deploy KL-Matching like in (Wang et al., 2022; Yang et al., 2022), where the average is computed over the train set.

**KNN** (KNN) (Sun et al., 2022): Computes the k-nearest neighbour in the normalized feature-space: The feature vector of a test input is normalized to $\mathbf{z} = \phi/||\phi||_2$ and the pairwise distances $r_i(\mathbf{z}) = ||\mathbf{z} - \mathbf{z}_i||_2$ to the normalized features $\mathcal{Z} = \{\mathbf{z}_1, ..., \mathbf{z}_N\}$ of all samples of the training set are computed. The distances $r_i(\mathbf{z})$ are then sorted according to their magnitude and the $K^{\text{th}}$ smallest distance, denoted $r^K(\mathbf{z})$ is used as negative OOD-score:

$$s = -r^K(\mathbf{z})$$

Like suggested in (Sun et al., 2022), we use $K = 1000$.

**ReAct** (E+R) (Sun et al., 2021): The authors propose to perform a truncation of the feature vector, $\bar{\phi} = \min(\phi, r)$, where the $\min$ operation is to be understood element-wise and $r$ is the truncation threshold. The truncated features can then be

converted to so-called rectified logits via $\bar{\mathbf{o}} = g(\bar{\phi}) = \mathbf{W}^T \bar{\phi} + \mathbf{b}$. While the rectified logits can now be used with a variety of existing detection methods, we follow (Sun et al., 2021) and use the rectified Energy as OOD-score:

$$s = \log \sum_c^C \exp\left(\bar{o}_c\right)$$

As suggested in (Wang et al., 2022), we set the threshold $r$ such that 1% of the activations from the train set would be truncated.

**Virtual Logit Matching** (ViM) (Wang et al., 2022): The idea behind ViM is that meaningful features are thought to lie in a low-dimensional manifold, called the principal space $P$, whereas features from OOD-samples should also lie in $P^\perp$, the space orthogonal to $P$. $P$ is the $D$-dimensional subspace spanned by the eigenvectors with the largest $D$ eigenvalues of the matrix $\mathbf{F}^T \mathbf{F}$, where $\mathbf{F}$ is the matrix of all train features offsetted by $\mathbf{u} = -(\mathbf{W}^\mathbf{T})^+ \mathbf{b}$ (+ denotes the Moore-Penrose inverse). A sample with feature vector $\phi$ is then also offset to $\tilde{\mathbf{h}} = \phi - \mathbf{u}$ and can be decomposed into $\tilde{\mathbf{h}} = \tilde{\mathbf{h}}^P + \tilde{\mathbf{h}}^{P^\perp}$, and $\tilde{\mathbf{h}}^{P^\perp}$ is referred to as the *Residual* of $\phi$. ViM leverages the Residual and converts it to a virtual logit $o_0 = \alpha ||\tilde{\mathbf{h}}^{P^\perp}||_2$, where

$$\alpha = \frac{\sum_{i=1}^N \max_c o_i^c}{\sum_{i=1}^N ||\phi_i^{P^\perp}||_2}$$

is designed to match the scale of the virtual logit to the scale of the real train logits. The virtual logit is then appended to the original logits of the test sample, i.e. to $\mathbf{o}$, and a new probability vector is computed via the softmax function. The probability corresponding to the virtual logit is then the final OOD-score:

$$s = -\frac{\exp\left(o_0\right)}{\sum_{c=1}^C \exp\left(o_c\right) + \exp\left(o_0\right)}$$

Like suggested in (Wang et al., 2022), we use $D = 1000$ if the dimensionality of the feature space $d$ is $d \geq 2048$, $D = 512$ if $2048 \geq d \geq 768$, and $D = d/2$ rounded to integers otherwise.

**Cosine** (Cos) (Techapanurak et al., 2020; Galil et al., 2023): This method computes the maximum cosine-similarity between the features of a test-sample and embedding vectors $\tilde{\mathbf{u}}_c$ (sometimes also called concept-vector):

$$s = \max_c \frac{\tilde{\mathbf{u}}_c^T \phi}{||\tilde{\mathbf{u}}_c^T||_2 ||\phi||_2} \tag{10}$$

**Ash** (Ash) (Djurisic et al., 2023): Ash applies activation shaping at inference time by pruning acitvations below a certain threshold, and then binarizing (Ash-b) or scaling (Ash-s) the remaining activations, which are then processed as usually in the network. As suggested by the authors, we apply ash to the pre-logit feature activations.

**Softmax-scaled-Cosine** (SSC) (Tack et al., 2020): Normalize the rows of the weight matrix $\mathbf{w}_i$ and the features, and compute the cosine between the two:

$$\cos\theta_i \equiv \frac{\mathbf{w}_i \cdot \phi}{||\mathbf{w}_i|| ||\phi||}$$

Then scale by a scalar $t$ and apply the softmax, to finally use the max-softmax as OOD score:

$$s = \max_i \left(\mathrm{softmax}(t * \theta)_i\right)$$

In Tack et al. (2020) the scalr $s$ is learned, for our post-hoc setup we set $s = 1$.

**NeCo** (Nec) (Ammar et al., 2024): Compute the covariance matrix of the feature space, and project to the $d$ eigenvectors with largest eigenvalues with the corresponding projection matrix $P$. The difference in norm of the projected features and the original features is then scaled with the max-logit and serves as OOD score.

$$s = \frac{||P\phi(x)||}{||\phi(x)||} * \max_i o_i = \sqrt{\frac{\phi(x)^\top P P^\top \phi(x)}{\phi(x)^\top \phi(x)}} * \max_i o_i$$

Like suggested by the authors, we standardize data and select $d$ such that 90% of the train variance are explained.

**Gaussian Mixture Model** (GMM or GMN) (Mukhoti et al., 2023): Estimate a Gaussian mixture model on the train features $\phi(x)$, and use the log-probabilities as OOD score. We use GMN for Gaussian mixture model with normalized features, and GMM for Gaussian mixture model with regular features.

**NNguide** (NNG) (Park et al., 2023a): "Guide" the energy score by a nearest-neighbor score:

$$s = s_{Energy} * s_{KNN}$$

where $s_{KNN}$ is a KNN score in the normalized feature space, estimated on a subset of the train features. Like suggested by the authors, we use $1\%$ of the train features and $K = 10$ neighbors for ImageNet experiments. We also tried $K = 1000$, as increasing $K$ showed promising results in an ablation by the authors (Figure 4 in the paper), but found that it performs worse on average than $K = 10$.

**Relative Mahalanobis distance** (rMaha) (Ren et al., 2021): A modification of the Mahalanobis distance method, thought to improve near-OOD detection, is to additionally fit a global Gaussian distribution to the train set without taking class-information into account:

$$\hat{\mu}_{\text{global}} = \frac{1}{N} \sum_i \phi_i, \quad \hat{\Sigma}_{\text{global}} = \frac{1}{N} \sum_i (\phi_i - \hat{\mu}_{\text{global}})(\phi_i - \hat{\mu}_{\text{global}})^T$$

The OOD-score is then defined as the difference between the original Mahalanobis distance and the Mahalanobis distance w.r.t. the global Gaussian distribution:

$$s = -\min_c \left( (\phi - \hat{\mu}_c)\hat{\Sigma}^{-1}(\phi - \hat{\mu}_c)^T - (\phi - \hat{\mu}_{\text{global}})\hat{\Sigma}_{\text{global}}^{-1}(\phi - \hat{\mu}_{\text{global}})^T \right)$$

