# OpenReview forum: "Mahalanobis++: Improving OOD Detection via Feature Normalization"
_ICML.cc/2025/Conference — ICML 2025 poster_

### Official Review · Reviewer_qhSY · 2025-03-06

**Overall Recommendation:** 4

**Summary:**

The paper proposes a simple fix to the Post-Hoc OOD detection technique based on the Mahalanobis distance computed on the feature space of the neural network of interest. This simple fix consists of normalizing the features by their $l_2$ norm before computing the distance. The authors emphasize how the samples violate the assumptions underlying Mahalanobis distance in the feature space:
- Assumption 1: the class wise features follow a multivariate normal distribution
- Assumption 2: class conditional covariance matrices are the same
They do so by analyzing the magnitude of the feature norm, emphasizing how the fix can alleviate this problem. Experiments on a comprehensive benchmark of various models empirically demonstrate the effectiveness of this method.

**Claims And Evidence:**

The problem with the feature norm is clearly illustrated with experiments based on Lemma 3.1, expected squared variance deviation, and QQ plots.

However, I have some concerns with the link between the fix and the assumptions. The fix intends to alleviate the difference between the feature norms of samples from different classes, but I do not see how it makes the features satisfy assumptions 1 and 2. Specifically, nothing ensures that after the fix, which is just a normalization, the features follow a normal gaussian, and that their covariance matrices are equal.
- QQ plots are here to show that the obtained features are closer to normal gaussian, but they might still be non gaussian.
- I do not see the relation between assumption 2 and the expected squared variance deviation, which I think is not a standard metric. Two very different covariance matrices could have a deviation of zero with this metric. Why not conducting statistical tests, or using standard probability distribution divergence measures that can be estimated?

**Essential References Not Discussed:**

All essential references are discussed to the best of my knowledge

**Experimental Designs Or Analyses:**

yes

**Methods And Evaluation Criteria:**

- The experiments to emphasize the problem with feature norm are thorough and theoretically grounded
- The evaluation benchmark is extensive.

**Other Comments Or Suggestions:**

- In the proof of Lemma 3.1, $X$ is not introduced (the lemma is about $\Phi(X)$)
- Eq. 5 "trace" please use the same notation as lemma 3.1 ("tr")

**Other Strengths And Weaknesses:**

### Strenght

- The problem with Assumptions 1 and 2 is clearly emphasized
- The method is simple
- It consistently improves the performance of Mahalanobis method

### Weaknesses

- What is called a "fix" might not be an actual "fix" but just a tool to make Mahalanobis method better
- Concerns with expected squared variance deviation

**Questions For Authors:**

- Could you plot the distribution of normalized features in a plot similar to Figure 3?

**Relation To Broader Scientific Literature:**

The contributions are closely related to (Lee et al. 2018b) and (Ren et al, 2021) which are appropriately discussed.

**Theoretical Claims:**

I checked the proof of Lemma 3.1 which is Ok.
I skimmed through Appendix C (proof of expected squared variance deviation) but did not check thoroughly.

---

> ### Author Rebuttal · Authors · 2025-03-31
>
> We thank the reviewer for the thoughtful comments and appreciate the positive feedback. Below we address the reviewers remarks:
> - __“The fix intends to alleviate the difference between the feature norms of samples from different classes”__
>
>     We would like to clarify that different feature norms for samples from different classes are not a problem per se. For instance, if the mean vectors of different classes were of different magnitude (which is typically not the case), the Gaussian assumption could still be satisfied. However, our analysis shows that the observed feature norm distribution and the one that we would expect under the Gaussian model are significantly different, for instance showing very heavy tails. We take this as indication that the Gaussian assumption is violated, and substantiate this with further analysis (QQ plots, etc).
> - __“nothing ensures that after the fix, which is just a normalization, the features follow a normal gaussian, and that their covariance matrices are equal.”__ and __“QQ plots are here to show that the obtained features are closer to normal gaussian, but they might still be non gaussian.”__
>
>     We agree that we cannot guarantee that the features follow a normal distribution. In fact, we believe that there is no reason to believe that the features follow any particular distribution. However, we provide strong empirical evidence that _modelling_ the feature distribution with a normal distribution with shared covariance is _more appropriate after normalization_. In particular, 1) the QQ plots are less skewed, 2) the shared covariance assumption is better satisfied, and 3) the feature norm does not act as a confounder for OOD detection anymore.
> - __"What is called a 'fix' might not be an actual 'fix'"__
>
>     In addition to the above, we are happy to rephrase from "fix" to e.g. "remedy"
> - __“Could you plot the distribution of normalized features in a plot similar to Figure 3?”__
>
>     The feature norms of the normalized features would show as a straight line at 1 with no deviation. Please let us know in case this does not clarify the question.
> - We thank the reviewer for the remarks about the trace notation and for noting that $\Phi(X)$ has not been properly introduced. We will adjust the notation and clarify that $\Phi(X)$ is a random variable representing the feature distribution for input $X$.
> - __”two very different covariance matrices could have a deviation of zero with this metric. “ (Eq. 5)__
>
>     We respectfully disagree with the reviewer. In particular, we can write $$\mathbb{E}_u \left[ \left( \frac{u^T \hat{\Sigma}_i u}{u^T \hat{\Sigma} u} - 1 \right)^2 \right]=\mathbb{E}_u \left[ \left( \frac{u^T (\hat{\Sigma}_i-\hat{\Sigma}) u}{u^T \hat{\Sigma} u}\right)^2 \right]$$ Since $\Sigma$ is pd, $u^T {\Sigma} u>0$, and the only way for the expectation to be zero is that $u^T (\hat{\Sigma}_i-\hat{\Sigma}) u=0$ for all $u$, wich is only the case when $\Sigma=\Sigma_i$.
> - __"expected squared variance deviation ... is not a standard metric"__
>
>     We agree that this metric is not commonly evaluated, but we argue that it is the right one to look at. In particular, the Mahalanobis distance performs a whitening by the variances: Deviations in a certain direction are measured _relative_ to the sample variance in this direction. Small absolute deviations can thus result in large distances when they are along a direction of small variance. We therefore need a measure that can capture _relative_ deviations instead of absolute deviations, since absolute deviations would be dominated by directions of large variance. Our proposed measure computes the _relative deviation_ of the variance of $\Sigma_i$ from $\Sigma$ in _every direction u_ and averages this deviation over all directions. This is a natural way to assess whether $\Sigma_i$ and $\Sigma$ are similar in all possible directions in the feature space. A similar measure that is commonly used to compare covariance matrices is the Riemannian metric (see e.g. [1,2]) $d(\Sigma_1, \Sigma_2) := \sqrt{\mathrm{tr}\left(\ln^2\left({\Sigma_1^{-0.5}}\Sigma_2{\Sigma_1^{-0.5}}\right)\right)}$. It is also possible to compute an appropriate measure with divergences like the KL divergence: $KL_{\text{normal}}(I_n,\Sigma_1^{-0.5}\Sigma_2\Sigma_1^{-0.5})$. We evaluate both, confirming that normalization aligns the covariance structure in a meaningful way (lower is better):
>
> ||Riemann|Riemann|KL| KL|
> |:-------------|------------------------:|:----------------------|----------------------------:|:--------------------------|
> ||unnormalized |normalized|unnormalized|normalized|
> | mean|98.2 | **88.2**| 1090.6|**982.0**|
> | median|93.6 | **84.7** | 1011.0  |**908.3** |
>
> We are happy to discuss any of the points further!
>
>
> [1] Förstner & Moonen. (2000). A Metric for Covariance Matrices. 10.1007/978-3-662-05296-9_31.
>
> [2] Pennec, Fillard, & Ayache. A Riemannian Framework for Tensor Computing. Int J Comput Vision 66, 41–66 (2006).

---

> > ### Comment · Reviewer_qhSY · 2025-04-04
> >
> > I appreciate the author's response and would like to increase my rating.

---

> > > ### Author Response · Authors · 2025-04-04
> > >
> > > We are glad that the reviewer appreciates our rebuttal response and would like to thank them for raising the score!

---

### Official Review · Reviewer_ruxa · 2025-03-10

**Overall Recommendation:** 3

**Summary:**

This submission focuses on the OOD detection task and it proposes a simple yet effective method to improving the Mahalanobis distance approach.

## update after rebuttal

The authors rebuttal has largely addressed my concerns and I thus maintain my positive rate.

**Claims And Evidence:**

While in a mixture of theoretical and emphrical way, the reviewer believees that the submission is supported by clear evidences.

**Essential References Not Discussed:**

N.A.

**Experimental Designs Or Analyses:**

Yes, the reviewer has gone through the experimental settings in the main paper and finds that they largely make sense.

**Methods And Evaluation Criteria:**

Yes, the reviewer believes that the proposed method makes sense.

**Other Comments Or Suggestions:**

N.A.

**Other Strengths And Weaknesses:**

I really appreciate the authors effort in perfoming extensive experimental analysis and I believe that they can to a large scale strength this submission. Below, I still have several queries over this submission that I hope can be addressed to further improve the quality.

1. I hope the related works section can be better organlized and the differences between the proposed method and existing methods to be better elaborated. For example, when the method names a subsection called Mahalanobis distance, it actually wants to review Mahalanobis-distance-based existing methods from my understanding. Thus, it is important for this to be clarified. Meanwhile, it is appreciated if the difference between the proposed method and existing similar methods to be better elaborated.

2. When the authors present Lemma 3.1, if I am not wrong, it is only the property of Gaussian distributed features but not sufficient condition. If this is the case, I appreciate the authors to make this more clear there to avoid reader's misunderstanding.

3. The authors claim that "we expect this to be negligible due to the large dataset size". I first appreciate more explanation or elaborate on this negligibility. Meanwhile, the authors seem to require the size to be very large (>10^6). What if in some cases that this is not the case? If the negligibility still holds?

4. Finally, if I am not wrong, the key motivation seems to be concentrating the feature norm. I am thus a bit curious here, what if we not only normalize like Eq. 6 but concentrate the feature even further? What will happen? Meanwhile, while I admit its naturaness, is there any specific reason for the authors to choose to perform concentration via normalization?

I still have these queries yet I remain positive on this submission. I thus vote for weak accept now.

**Questions For Authors:**

(see above in other weaknesses.)

**Relation To Broader Scientific Literature:**

The reviewer believes that Gaussian distribution assumption can be very common in different scientific literatures. From this perspective, it is very interesting for a method to appear if it does can help mitigating the violation of this assumption in a relatively simple way.

**Theoretical Claims:**

The reviewer hasn't carefully checked the proof of Lemma 3.1. Yet, Lemma 3.1 is at least not contradicting with the reviewer's intuition.

---

> ### Author Rebuttal · Authors · 2025-03-31
>
> We thank the reviewer for their positive feedback, and for appreciating our work. We address the remarks below:
> 1.  __"organize related work section"__ and __"elaborate difference to existing similar methods"__
>
>     We will extend the discussion about related work, and emphasize the differences to previous work that used feature normalization [3,4] or the Mahalanobis distance, or both [1,2]. Most importantly, other works have investigated _train-time_ methods that involve normalization. Either implicitly through contrastive losses (CIDER [1], SSD[2]), or explicitly to improve OOD detection [3,4] It is then natural to also apply normalization at inference time. For instance, CIDER applies KNN, and SSD performs k-means and then Mahalanobis. Those methods thus normalize their features for OOD detection _because_ they also normalize during training. This is orthogonal to our work: The standard Mahalanobis method for OOD detection is a _post-hoc_ method, where adjusting the pretraining scheme is not feasible. We show that in this setting, the Gaussian assumption underlying this method is often severely violated, and that normalizing the features better aligns with this assumption, consistently improving OOD detection across architectures and pretraining techniques. We will clarify this distinction and expand the discussion of other approaches in the paper (see the answer to reviwer jfEM for a more thorough discussion and quant. comparisons to SSD). If there is a specific reference the reviewer would like us to discuss, please let us know.
>
> 2. __Lemma 3.1, not sufficient condition__
>
>     We will clarify that a concentrated feature norm is not a sufficient, but a necessary condition for a Gaussian distribution. Lemma 3.1 only shows that - under the assumption of a Gaussian distribution in feature space - we expect some concentration of the feature norm. To illustrate this, we sample from class-specific Gaussian distributions with the estimated means and shared covariance matrix (Figure 3-left), noting that in practice (Figure 3-right) the feature norms deviate strongly from the Gaussian model (e.g. via heavy tails). This suggests severe violations of the Gaussian assumption, which we substantiate by QQ plots and the variance alignment analysis. Our remedy - normalization - aligns the features better with the premise of normally distributed data with shared covariance matrix.
>
> 3. __"elaborate on neligibility" (in QQ plot analysis)__
>
>     In QQ-plots, we compare empirical quantiles against a theoretical standard normal distribution. Since normalized and unnormalized features have different variances, their QQ-plots would have different slopes, making direct comparison difficult. To align comparisons, we divide both samples by their empirical standard deviation—this ensures both are evaluated against the same reference slope (black line in Figure 4). Dividing by the empirical variance technically transforms a normal distribution to a Student's *t*-distribution with *n*-1 degrees of freedom.  As the reviewer pointed out correctly, this matters for small $n$. However, the *t*-distribution converges to a Gaussian as $n\to\infty$, and for $n>30$, the difference is typically negligible [5]. We use all ImageNet train features ($n>10^{6}$) in our QQ plots, making the *t*-distribution practically Gaussian, allowing for the analysis we performed in the paper. We would like to stress that all of this is only a technicality in the analysis of the features via QQ plots, and irrelevant for Maha++ as an OOD detection method.
>
> 5. __concentration of feature norm is "key motivation"__
>
>     Our key motivation is not to concentrate the feature norm. Instead, feature norm concentration is a necessary condition IF the features were indeed normally distributed. As we find, the feature norms are, however, not concentrated, but for instance show extremely heavy tails. We take this as an indication that the Gaussian assumption is violated, and further validate it via QQ plots and our variance analysis. Regarding the reviewers question about __concentrating even further__: We are not sure we understand what the reviewer means by this. One could, in principle, normalize by a different norm (e.g. $\ell_1$ or $\ell_\infty$), but this would change the direction of the features. We therefore opted for $\ell_2$ normalization. Does this answer the question?
>
> We are happy to clarify any of the points further!
>
> [1] Ming et al. How to exploit hyperspherical embeddings for out-of-distribution detection? ICLR2023
>
> [2] Sehwag et al. Ssd: A unified framework for self-supervised outlier detection, ICLR 2021
>
> [3] Regmi et al. T2fnorm: Train-time feature normalization for ood detection in image classificatio, CVPR 2024 workshop
>
> [4] Haas et al. Linking neural collapse and l2 normalization with improved out-of-distribution detection in deep neural network, TMLR 2023
>
> [5] https://www.jmp.com/en/statistics-knowledge-portal/t-test/t-distribution

---

### Official Review · Reviewer_vddp · 2025-03-14

**Overall Recommendation:** 4

**Summary:**

The paper revisits the Mahalanobis distance for out-of-distribution detection. It first examines how the assumptions underlying the Mahalanobis distance for OOD detection are violated by a variety of models. It then proposes a maximally simple but effective remedy by applying l2-normalization to the pre-logit features. The evaluation shows that this outperforms previous works by a significant margin.

**Claims And Evidence:**

The claims made by the paper are supported by clear and convincing evidence. The paper demonstrates that, empirically, feature distributions of some models do not fit the assumptions made by prior Mahalanobis distance-based OOD detection. Figure 5 further shows that, for SwinV2-B models, the feature norm is strongly correlated with the Mahalanobis distance, while beeing a bad OOD predictor, which in turn leads to suboptimal OOD detection performance. In contrast, applying l2-normalization as proposed reduces correlation between feature norms and Mahalanobis distance, which allows drawing a better decision boundary. The findings are furthermore validated by the quantitative evaluation of the proposed method on a wide variety of pre-trained models.

**Essential References Not Discussed:**

None.

**Experimental Designs Or Analyses:**

The main experimental design is focused on evaluating the OOD false positive rate at a fixed true positive rate of 95% across different models and datasets, which is in line with prior work. The experimental analysis demonstrates that models suffer from violations of Mahalanobis based OOD detection with varying degree, but that most models benefit somewhat from l2-normalization as proposed.

**Methods And Evaluation Criteria:**

The method is well motivated by pointing out how the assumptions in prior Mahalanobis based OOD detection methods can be violated by some models. The evaluation metrics (false-positive rate at true positive rate of 95% in particular) and benchmark datasets make sense and are in line with prior work on OOD detection. I appreciate that the evaluation is performed on a wide variety of model types, architectures and sizes.

**Other Comments Or Suggestions:**

1. ImageNet reference renders as "(University, 2015)"

**Other Strengths And Weaknesses:**

None.

**Questions For Authors:**

None.

**Relation To Broader Scientific Literature:**

While the purely methodological innovation of this paper is minimal, its value lies in identifying and empirically demonstrating violations of key assumptions of Mahalanobis based OOD detection in practice, proposing a maximally simple remedy, and providing thorough evaluation of this remedy on a wide variety of models.

**Theoretical Claims:**

The main theoretical claim can be found in equation 5 and is elaborated upon in the appendix, which I did only check superficially.

---

> ### Author Rebuttal · Authors · 2025-03-31
>
> We thank the reviewer for carefully reading and evaluating our paper, and we are glad that the reviewer finds that our claims are __“supported by clear and convincing evidence”__, that our method is __“well motivated”__, that that they appreciate the __“wide variety of model types, architectures and sizes”__ in our __“thorough evaluation"__. We agree with the reviewer that the results about augreg ViTs stand out, and think that investigating the underlying reasons for the behaviour of those models (i.e. the why the augreg training scheme results in the favourable structure of the feature space) is an interesting direction for future research. We thank the reviewer for pointing out the incorrect ImageNet reference, which we will fix. For the rebuttal, we have included a more thorough discussion and comparison to SSD (see response to reviewer jfEM), an evaluation of a DinoV2 model (also in response to reviewer jfEM) and more variance deviation measures (see response to reviewer qhSY). If there is anything else the reviewer would like to see addressed, we would be happy to discuss this.

---

### Official Review · Reviewer_jfEM · 2025-03-16

**Overall Recommendation:** 3

**Summary:**

This paper presents an holistic empirical analysis illustrating the current violation of the gaussian distribution of the representations of most vision backbones. From this constatation, the paper introduces a variation of the Mahalanobis distance for OOD detection called Mahalanobis++. Extensive experiments on multiple recent OOD benchmarks and various bacbones are proposed to assess the good behavior of the proposed approach.

## Update after rebuttal

I am satisfied with the rebuttal and will thus keep my positive rating

**Claims And Evidence:**

The principal claim concerns the violation of the class-wise unimodal Gaussian hypothesis of the representations. This is a reasonable claim as the Mahalanobis method does rely on strong relaxations for computational reasons. Moreover, this claim is supported by strong empirical evidence in this paper, see Fig. 3, 4, 5, and Table 1.

**Essential References Not Discussed:**

* Good performances of Mahalanobis distance for OOD detection on normalized features have already been explored in SSD [1]. In the related work section, authors state that "Adapting them to ImageNet-scale setups as post-hoc OOD detectors has so far not been successful". Same in the end of the method section: "While $\ell_2$-normalization has been used with non-parametric methods like KNN (Sun et al., 2022; Park et al., 2023a) or cosine similarity (Techapanurak et al., 2020), it is - to the best of our knowledge - not used with the Mahalanobis score". This is a bit of an overstatement as Mahalnobis is a strong and cheap baseline even on large scale datasets and SSD has been successfully experimented on ImageNet-1k.Thus, a broader discussion and comparison with SSD is missing in the current paper.

[1] Sehwag, Vikash, Mung Chiang, and Prateek Mittal. “SSD: A Unified Framework for Self-Supervised Outlier Detection,” ICLR 2021

**Experimental Designs Or Analyses:**

The evaluation protocol is well designed. However, as many backbones pretrained with a contrastive loss also normalize the representations, comparison of Mahalanobis with SSD [1] or on other DINO-like backbones would give important insight on the method and the importance of normalization for OOD detection.


[1] Sehwag, Vikash, Mung Chiang, and Prateek Mittal. “SSD: A Unified Framework for Self-Supervised Outlier Detection,” ICLR 2021

**Methods And Evaluation Criteria:**

Evaluation criteria and benchmarks are standard for OOD detections.

Reporting only FPR95 is not a standard practice as this metric is not robust to small changes in the decision function and is particularly sensitive to class imbalance. FPR@95 highlights performance at a specific critical threshold but is typically complemented by AUC, ensuring a more holistic evaluation.  I see that AUC scores in the supplementary are still in favor of the proposed approach.

**Other Comments Or Suggestions:**

NA

**Other Strengths And Weaknesses:**

The paper is very well written and supported with extensive evaluations.

**Questions For Authors:**

Despite bringing appreciated insights into the distance-based OOD literature, the related work section misses clear positioning *e.g.* which challenges are unaddressed by normalized approaches such as [2,3] or CIDER [4]? What makes the proposed method better suited for OOD detection?


[2] Regmi, S., Panthi, B., Dotel, S., Gyawali, P. K., Stoyanov, D., and Bhattarai, B. T2fnorm: Train-time feature normalization for ood detection in image classification, CVPR Workshop 2024
[3] Haas, J., Yolland, W., and Rabus, B. T. Exploring simple, high quality out-of-distribution detection with l2 normalization, TMLR, 2024.
[4] Ming, Y., Sun, Y., Dia, O., and Li, Y. How to exploit hyperspherical embeddings for out-of-distribution detection? Neurips 2023

**Relation To Broader Scientific Literature:**

This paper shares the violation of the Gaussian distribution with multiple other distance-based papers. Other approaches proposed a pre-training strategy to mitigate this limitation.

The proposed extension to Mahalanobis is particularly incremental. However, it is well illustrated both by empirical statistical evaluation of the feature dispersion and extensive evaluations.

**Theoretical Claims:**

Lemma 3.1. does not support the indicated conclusion. First, features should be concentrated around $\sqrt{\text{tr}(\Sigma) - ||\mu||^2_2}$. Moreover, the higher the dimension, the looser is the upper-bound.

---

> ### Author Rebuttal · Authors · 2025-03-31
>
> We thank the reviewer for their valuable comments and address the remarks below:
> - __"the features should be concentrated around $\sqrt{\mathrm{tr}(\Sigma)-\|{\mu}\|^2_2}$" (in Lemma 3.1)__
>
>     We thank the reviewer for checking our proof, but we strongly believe that the term $\sqrt{\mathrm{tr}(\Sigma)+\|{\mu}\|^2_2}$ is correct. The term stated by the reviewer can even become complex if the norm of the mean is large enough. As the variance goes to zero, the norm of the random variable should be concentrated around $\|\mu\|_2$, which is exactly what our term states. We are happy to answer any questions on a particular step of the proof to resolve any potential confusion.
> - __"the higher the dimension, the looser is the upper-bound" (in Lemma 3.1)__
>
>     This is expected, as the squared $\ell_2$-norm grows with dimension $d$. However, the deviation per dimension decreases: $$\Pr\left(\frac{1}{d}\left|\|\Phi(X)\|^2_2 - \left(\mathrm{tr}(\Sigma) + \|\mu\|^2_2\right)\right| \geq \epsilon\right) \leq \frac{\mathrm{Var}\left(\|\Phi(X)\|_2^2\right)}{d^2 \epsilon^2}$$ The right side decreases with $d$ as the variance grows linearly in $d$. Lemma 3.1 shows that under the Gaussian assumption, we should see some concentration of the feature norms. To illustrate this, we simulate it in Figure 3 (left) by sampling from class-specific Gaussian distributions with the estimated means and shared covariance matrix, noting that the actual feature norms (Fig. 3 right) deviate strongly from the Gaussian model (e.g. via heavy tails). This suggests severe violations of the Gaussian assumption, which we substantiate by QQ plots and the variance alignment analysis.
> - __"the sentence _'Adapting them to IN-scale setups ... has so far not been successful'_ ... is a bit of an overstatement ... as Mahalanobis ... has been successful on IN-1k"__
>
>     We agree, this statement only refers to Gaussian mixture models (GMMs), and not to the Mahalanobis distance. We will clarify this in the paper and explain the difference between GMMs and the Mahalanobis distance.
> - __"the sentence _'l2-normalization...is ... not used with the Mahalanobis score'_ ... is a bit of an overstatement ... as SSD ... has been successful on IN-1k"__
>
>     We agree that Mahalanobis has been applied to normalized features in other works like SSD[1] and CIDER[2], and we should have chosen our statement more carefully. However, these are _train-time_ methods where normalization is implicitly part of their contrastive loss. Those methods thus normalize their features for OOD detection _because_ they also normalize during training. This is orthogonal to our work: The standard Mahalanobis method for OOD detection is a _post-hoc_ method, where adjusting the pretraining scheme is not feasible. We show that in this setting, the Gaussian assumption underlying this method is often severely violated, and that normalizing the features better aligns with this assumption, consistently improving OOD detection across architectures and pretraining techniques. We will clarify this distinction and expand the discussion of [1-4] in the paper (see below for SSD).
> - __"a broader discussion and comparison with SSD"__ and __"which challenges are unaddressed by normalized approaches"__
>
>   SSD involves three steps:
>     1) Training with a supervised (SSD+) or unsupervised (SSD) contrastive loss (implicitly normalizing features),
>     2) Cluster estimation via k-means in the normalized feature space,
>     3) Mahalanobis-based OOD detection using cluster labels instead of class labels.
>
>     This setting differs fundamentally from ours, as SSD, like [1,3,4], is a _train-time_ method. Methods like [1-4] cannot be directly applied to the pretrained checkpoints we evaluate. To demonstrate the advantages of post-hoc approaches, we evaluate SSD+ on NINCO using the ResNet50 from [5] (trained for 700 epochs). SSD+ is clearly outperformed by our top models, with FPR >3× higher. Those are obtained from various pretraining schemes, and retraining models with SSD or [1,3,4] on this scale is typically not feasible.
> | model | FPR|
> |-------------|-------|
> |SSD+ w. 100 clusters|66.0% |
> |SSD+ w. 500 clusters|65.7% |
> |SSD+ w. 1000 clusters|67.8% |
> |CnvNxtV2-L + Maha++ | 18.4%|
> |EVA02-L14 + Maha++| 18.6%|
> - __comparison with DINO__
>
>     We report the FPR of a DinoV2-S model (ft on IN1k) on NINCO. Maha++ outperforms Maha clearly.
>
> | Maha|Maha++|
> |------|------|
> | 77.3%| 53.4%|
>
> [1]Ming et al. How to exploit hyperspherical embeddings for out-of-distribution detection? ICLR2023
>
> [2]Sehwag et al. Ssd: A and unified framework for self-supervised outlier detection, ICLR 2021
>
> [3]Regmi et al. T2fnorm: Train-time feature normalization for ood detection in image classificatio, CVPR 2024 workshop
>
> [4]Haas et al T. Linking neural collapse and l2 normalization with improved out-of-distribution detection in deep neural network, TMLR 2023
>
> [5]Sun et al. Out-of-distribution detection with deep nearest neighbors. ICML 2022

---

### Decision · Program_Chairs · 2025-05-01

**Decision:**

Accept (poster)

**Comment:**

After review, the paper received four positive evaluations. Following the authors' rebuttal, one reviewer increased the rating, while the other three maintained their original scores. All reviewers acknowledged the paper's contributions and expressed satisfaction with its technical merits.

The AC concurs with the reviewers' assessments and recommends acceptance.